# Residential green space, air pollution, and related metabolites in association with depression among cancer survivors

Jianhui Zhao [1,2,10], Jingyu Ye[1,10], Erxu Xue[1,2,3,10], Liying Xu[1], Jing Sun [1], Siyun Zhou[1], Tengfei Li[1,4], Haoze Cao[1,4], Zhongquan Sun[4], Weilin Wang [4], Yazhou He[5], Yuan Ding [4,6,7] ✉ & Xue Li [1,8,9] ✉

The association of natural environmental exposure and air pollution with depression incidence among cancer survivors, as well as the potential role of plasma metabolomics, remains unclear. Here, we analyze 21,507 cancer survivors from the UK Biobank over a median follow-up of 12.39 years and find that individuals exposed to higher levels of green space and natural environment (tertile 3 vs. tertile 1) within a 1000-m buffer have 15.8% (95% CI: 4.0%-26.1%) and 18.2% (95% CI: 7.0%-28.1%) lower risks of depression, respectively. The strongest protective association is observed among breast cancer survivors. In contrast, higher exposures to nitrogen dioxide and nitrogen oxides are associated with an increased risk of depression. Meanwhile, plasma metabolic signatures associated with green space and natural environment may partially mediate these associations. These findings highlight that residential green space, natural environment, and lower air pollution levels may reduce depression risk among cancer survivors, possibly via metabolic pathways.

Although the global incidence and prevalence of cancer have steadily increased in recent decades, advances in early diagnosis, treatment, clinical management, and post-diagnostic lifestyle modifications have contributed to a decline in cancer mortality[1–3]. The pooled prevalence of depression among cancer patients ranges from 8% to 24%[4], increasing with disease recurrence and progression[5,6]. Depression in cancer patients is often underdiagnosed and undertreated, contributing to disease progression, increased mortality and other adverse outcomes[7,8]. The substantial burden of depression in cancer patients underscores the critical need to identify modifiable factors associated with mental health, which could serve as potential targets for future interventions and reveal novel therapeutic strategies.

In recent years, there has been growing recognition of the impact of the environment on individual health. Residential green space, blue space, and the natural environment, which are unique and potentially modifiable exposure constructs, are increasingly recognized for their potential protective effects against depression, anxiety, and other mental health issues[9,10]. Although a meta-analysis supports improving green space exposure to prevent depression in the general population[11], most studies are cross-sectional and have not specifically investigated cancer survivors, a psychologically vulnerable group. Furthermore, to date, prospective evidence examining the relationship between green and blue space and the natural environment exposure and incident depressive in long-term cancer survivors

[1]Department of Big Data in Health Science, The Second Affiliated Hospital, School of Public Health, Zhejiang University School of Medicine, Hangzhou, China. [2]Clinical and Translational Epidemiology Unit, Massachusetts General Hospital and Harvard Medical School, Boston, MA, USA. [3]The D. H. Chen School of Universal Health, Sir Run Run Shaw Hospital, Zhejiang University School of Medicine, Hangzhou, China. [4]Department of Hepatobiliary and Pancreatic Surgery, The Second Affiliated Hospital, Zhejiang University School of Medicine, Hangzhou, China. [5]Department of Oncology, West China School of Public Health and West China Fourth Hospital, Sichuan University, Chengdu, China. [6]Key Laboratory of Precision Diagnosis and Treatment for Hepatobiliary and Pancreatic Tumor of Zhejiang Province, Hangzhou, China. [7]Cancer Center, Zhejiang University, Hangzhou, China. [8]Zhejiang Key Laboratory of Intelligent Preventive Medicine, Hangzhou, China. [9]Centre for Global Health, Usher Institute, University of Edinburgh, Edinburgh, UK. [10]These authors contributed equally: Jianhui Zhao, Jingyu Ye, Erxu Xue. ✉e-mail: dingyuan@zju.edu.cn; xueli157@zju.edu.cn

remains limited. Given the stress-sensitive nature of long-term cancer survivors, it seems reasonable to hypothesize that exposure to green and blue space, as well as the natural environment, might be associated with a lower risk of mental health issues among cancer survivors. Ambient air pollution, including nitrogen dioxide ($NO_2$), nitrogen oxides ($NO_x$), particulate matter with aerodynamic diameter ≤10 μm ($PM_{10}$), and fine particulate matter ($PM_{2.5}$), is a major global health concern and has been associated with an increased risk of depression following both short- and long-term exposure[12,13]. However, its impact on depression among cancer survivors remains unclear. Furthermore, a recent study has identified several new metabolic effectors associated with depression, revealing dysregulation of multiple metabolites in the condition[14]. Metabolomics enables comprehensive analysis of metabolites in biological samples, providing valuable insights into the dynamic metabolic processes influenced by green space, blue space, natural environment, and air pollutants. However, evidence remains limited regarding the specific metabolites associated with these environmental exposures and whether disrupted circulating metabolism serves as a key intermediary linking such exposures to the development of depression.

In this work, we conduct a prospective investigation into the associations of green space, blue space, natural environment and air pollution with the risk of depression among cancer survivors. We further identify metabolic signatures associated with environmental exposures, prospectively examine their associations with depression risk among cancer survivors, and explore the potential mediating effects of these metabolic signatures on the associations between environmental exposures and depression (Fig. 1). Overall, our results highlight the favorable effects of green space and the natural environment, as well as the harmful impacts of air pollution on depression among cancer survivors, and elucidate the potential mechanisms driven by metabolites.

## Results

### Baseline characteristics

This study utilized data from the UK Biobank, and the baseline characteristics of the included participants are presented in Table 1. Among the 21,507 eligible cancer survivors, 13,414 (percentage = 62.4%) were female, and 19,853 (92.3%) were White, with a median age of 62.0 years. Compared to participants without depression, those diagnosed with depression were younger, more likely to be female, had a lower educational level and household income, a higher rate of employment, a higher prevalence of obesity, lower physical activity, higher rates of current smoking, and reported poorer dietary habits. The mean levels of residential green space, blue space, and natural environment at the 300 m buffer were 35.79%, 0.92%, and 27.02%, respectively; at the 1000 m buffer, the corresponding values were 45.71%, 1.27%, and 41.81% (Supplementary Table 1). The correlation matrix of environmental exposures was presented in Supplementary Table 2, where green space, blue space, and natural environment were generally negatively correlated with air pollutants.

### Environmental exposures and depression risk among cancer survivors

During an average follow-up period of 12.39 years, a total of 1427 depression cases were identified. Figure 2 illustrated the associations between nature exposures at 300 m and 1000 m buffers and the risk of depression among cancer survivors (Supplementary Tables 3 and 4). Compared with cancer survivors exposed to the lowest tertile of green space, blue space, and natural environment at 300 m buffer, those in the highest tertile showed a significantly reduced risk of depression in Model 1. However, the protective associations of green space and natural environment were not statistically significant in Model 3. Notably, compared with cancer survivors in the lowest tertile of blue space exposure at 300 m buffer, those in the highest tertile had a significantly lower risk of depression in Model 3 (hazard ratio [HR] = 0.867, 95% confidence interval [CI] = 0.763–0.986), with a significant trend ($P_{trend}$ = 0.031). Compared with cancer survivors exposed to the lowest tertile of green space at 1000 m buffer, those in the second and third tertiles showed a significantly reduced risk of depression in Model 3 ($HR_{tertile\ 2}$ = 0.878, 95% CI = 0.774–0.997, $P$ = 0.044; $HR_{tertile\ 3}$ = 0.842, 95% CI = 0.739–0.960, $P$ = 0.010). Similarly, compared with those in the lowest tertile of natural environment exposure at 1000 m buffer, cancer survivors in tertile 2 and tertile 3 exhibited a significantly reduced risk of depression ($HR_{tertile\ 2}$ = 0.814, 95% CI = 0.717–0.923, $P$ = 0.001; $HR_{tertile\ 3}$ = 0.818, 95% CI = 0.719–0.930, $P$ = 0.002). Exposure to higher levels of green space and natural environment was significantly associated with a reduced risk of depression in a dose-response manner ($P_{trend}$ = 0.010 and 0.002 for green space and natural environment, respectively). Additionally, each 5% increment in green space and natural environment at 1000 m buffer was associated with a 1.7% reduction in depression risk among cancer survivors ($P$ = 0.006 for green space; $P$ = 0.002 for natural environment). No significant association was observed between blue space at 1000 m buffer and depression risk among cancer survivors. Restricted cubic spline (RCS) analysis indicated significant linear exposure–response relationships between residential green space and natural environment at 1000 m buffer and incident depression ($P_{overall}$ = 0.009, $P_{non-linearity}$ = 0.171 for green space; $P_{overall}$ = 0.006, $P_{non-linearity}$ = 0.234 for natural environment) (Supplementary Fig. 1).

### Air pollutants and depression risk among cancer survivors

In Models 1 and 2, higher levels of air pollutants ($NO_2$, $NO_x$, and $PM_{2.5}$) were associated with an increased risk of depression among cancer survivors (Table 2). In the fully adjusted Model 3, cancer survivors exposed to the highest tertile of $NO_2$ had a 14.0% higher risk of depression compared to those in the lowest tertile (HR = 1.140, 95% CI = 1.001–1.299, $P$ = 0.048). Similarly, exposure to the highest tertile of $NO_x$ was associated with a 14.3% higher risk of depression (HR = 1.143, 95% CI = 1.004–1.303, $P$ = 0.039). Subsequently, an air pollution score (APS) was calculated based on weighted concentrations of $NO_2$, $NO_x$, and $PM_{2.5}$ (for details, see the Methods section). Results from Model 3 indicated tertile 3 of the APS was associated with a 15.2% higher risk of depression (HR = 1.152, 95% CI = 1.011–1.312; $P$ = 0.033) compared to the lowest tertile. Positive dose-response associations between $NO_2$, $NO_x$, and the APS and depression risk were observed when tertiles were modeled as a continuous variable ($P_{trend}$ = 0.045, 0.039, and 0.029 for $NO_2$, $NO_x$, and APS, respectively).

### Joint and mediation analyses

Joint analyses of air pollutants and green space, blue space, and natural environment were conducted (Figs. 3 and 4). Results showed that, compared with participants in Tertile 1 of green space at 300 m buffer and Tertile 3 of $NO_2$, $NO_x$, $PM_{2.5}$, and the APS, those in Tertile 3 of green space and Tertile 1 of air pollutants had a lower risk of depression by 21.1% (HR = 0.789, 95% CI = 0.670–0.929, $P$ = 0.005), 15.3% (HR = 0.847, 95% CI = 0.718–1.001, $P$ = 0.051), 17.8% (HR = 0.833, 95% CI = 0.708–0.980, $P$ = 0.027), and 18.1% (HR = 0.819, 95% CI = 0.696–0.962, $P$ = 0.015), respectively. Compared with participants in Tertile 1 of natural environment at 300 m buffer and Tertile 3 of $NO_2$, $NO_x$, $PM_{2.5}$, and the APS, those in Tertile 3 of natural environment and Tertile 1 of air pollutants had a lower risk of depression by 16.0% (HR = 0.840, 95% CI = 0.714–0.989, $P$ = 0.036), 14.1% (HR = 0.859, 95% CI = 0.731–1.009, $P$ = 0.064), 14.8% (HR = 0.852, 95% CI = 0.727–0.999, $P$ = 0.049), and 15.2% (HR = 0.848, 95% CI = 0.723–0.993, $P$ = 0.041), respectively. Furthermore, compared with participants in Tertile 1 of green space at 1000 m buffer and Tertile 3 of $NO_2$, $NO_x$, $PM_{2.5}$, and the APS, those in Tertile 3 of green space and Tertile 1 of air pollutants had a significantly lower risk of depression by 17.9% (HR = 0.821, 95% CI = 0.705–0.956, $P$ = 0.011), 16.4% (HR = 0.836, 95% CI = 0.712–0.982,

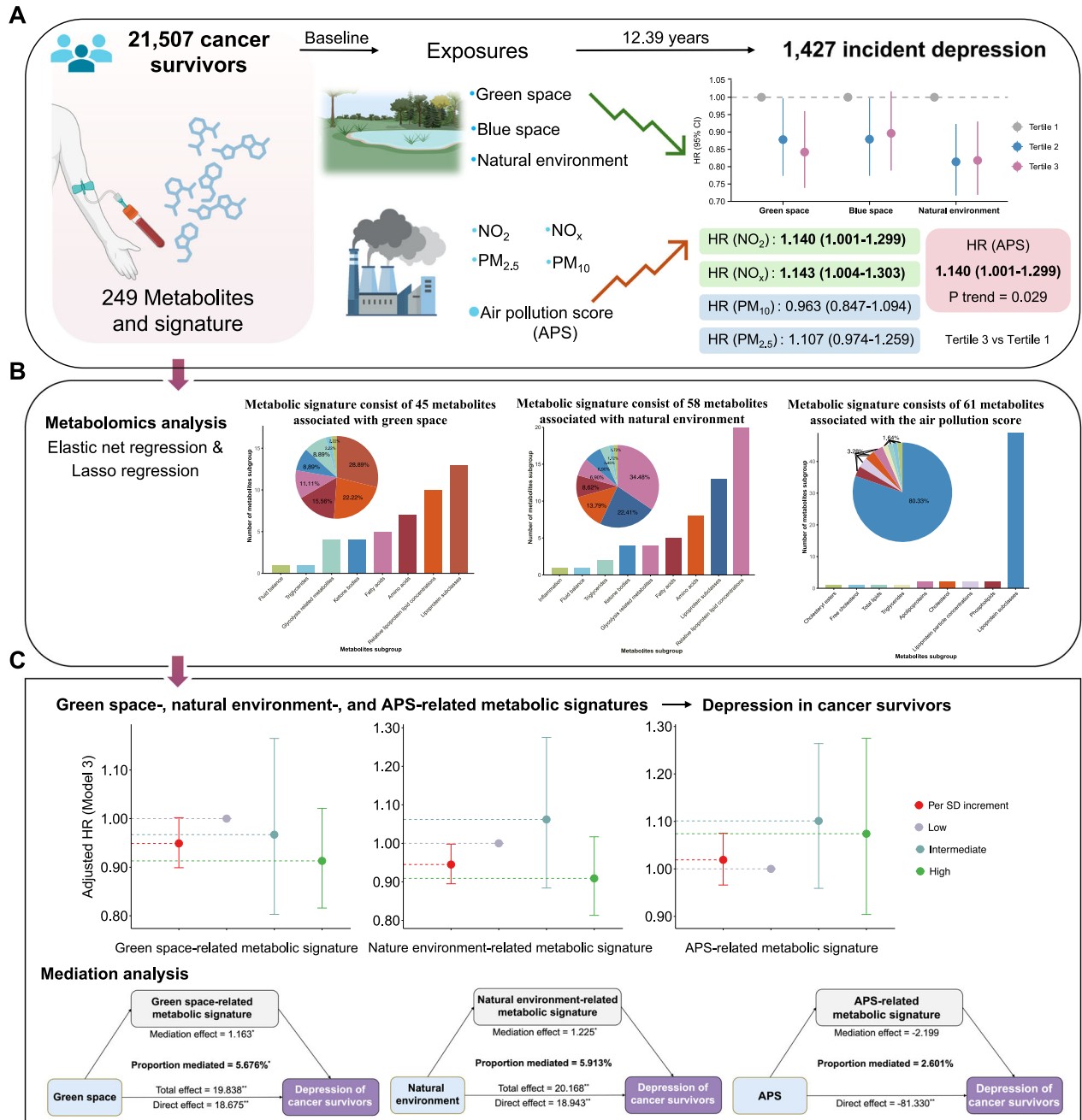

**Fig. 1 | Overview of the study design. A** Associations of green space, blue space, natural environment exposure, and ambient air pollution with depression risk among cancer survivors. A weighted APS is calculated to capture the joint exposure to various air pollutants by summing up concentrations of 3 ambient air pollutants ($PM_{2.5}$, $NO_2$ and $NO_x$), weighted by the multivariable-adjusted (Model 3) risk estimates ($\beta$ coefficients). **B** The development of metabolic signature score model. Using the 249 metabolites, we develop metabolic signatures associated with green space, natural environment and APS by employing both elastic net and LASSO regression to identify relevant metabolites. **C** Associations of individual metabolic signatures with incident depression among cancer survivors, and mediation analysis. Mediation analysis was conducted to identify the potential mediating effects of the metabolic signatures on the associations between green space, the natural environment, and APS and the risk of depression among cancer survivors. APS air pollution score, CI confidence interval, HR hazard ratio, LASSO Least Absolute Shrinkage and Selection Operator, $NO_2$ nitrogen dioxide, $NO_x$ nitrogen oxides, $PM_{10}$ particulate matter with aerodynamic diameter ≤10 μm, $PM_{2.5}$ particulate matter with aerodynamic diameter ≤2.5 μm.

$P = 0.029$), 16.1% (HR = 0.839, 95% CI = 0.714−0.986, $P = 0.033$), and 16.4% (HR = 0.836, 95% CI = 0.716−0.977, $P = 0.024$), respectively. Similarly, compared with participants in Tertile 1 of natural environment at 1000 m buffer and Tertile 3 of air pollutants, those in Tertile 3 of natural environment and Tertile 1 of $NO_2$, $NO_x$, $PM_{2.5}$, and the APS exhibited a reduced depression risk by 16.3% (HR = 0.837, 95% CI = 0.720-0.973, $P = 0.021$), 17.7% (HR = 0.823, 95% CI = 0.703−0.964,

$P = 0.016$), 15.3% (HR = 0.847, 95% CI = 0.724−0.991, $P = 0.038$), and 15.2% (HR = 0.834, 95% CI = 0.716−0.972, $P = 0.020$), respectively. Interestingly, similar results were observed in the joint analyses of blue space at 300 m and 1000 m buffers with air pollution, particularly under exposure at 300 m buffer.

Mediation analysis results indicated that while green space, blue space, and natural environment had significant direct effects on the

**Table 1 | Baseline characteristics of cancer survivors in this study**

| Variables | Total | Individuals without depression | Individuals with depression | P value |
|---|---|---|---|---|
| Number of participants | 21,507 | 20,080 | 1427 | |
| Age (year)[a] | 62.00 [56.00, 66.00] | 62.00 [56.00, 66.00] | 61.00 [55.00, 65.00] | <0.001 |
| Sex, n (%) | | | | |
| Female | 13,414 (62.4) | 12,382 (61.7) | 1032 (72.3) | <0.001 |
| Male | 8093 (37.6) | 7698 (38.3) | 395 (27.7) | |
| Ethnicity, n (%) | | | | |
| Non-white | 1654 (7.7) | 1553 (7.7) | 101 (7.1) | 0.397 |
| White | 19,853 (92.3) | 18,527 (92.3) | 1326 (92.9) | |
| Educational level, n (%) | | | | |
| University or college | 6386 (29.7) | 6058 (30.2) | 328 (23.0) | <0.001 |
| Other | 15,121 (70.3) | 14,022 (69.8) | 1099 (77.0) | |
| BMI, n (%) | | | | |
| Normal | 7168 (33.3) | 6773 (33.7) | 395 (27.7) | <0.001 |
| Overweight | 8930 (41.5) | 8375 (41.7) | 555 (38.9) | |
| Obese | 5409 (25.1) | 4932 (24.6) | 477 (33.4) | |
| Employment status, n (%) | | | | |
| Currently unemployed | 8929 (41.5) | 8438 (42.0) | 491 (34.4) | <0.001 |
| Currently employed | 12,578 (58.5) | 11,642 (58.0) | 936 (65.6) | |
| Household income, n (%) | | | | |
| Low | 6599 (30.7) | 5979 (29.8) | 620 (43.4) | <0.001 |
| Middle | 11,133 (51.8) | 10,469 (52.1) | 664 (46.5) | |
| High | 3775 (17.6) | 3632 (18.1) | 143 (10.0) | |
| Physical activity (MET), n (%)[b] | | | | |
| Low | 10,930 (50.8) | 10,157 (50.6) | 773 (54.2) | 0.010 |
| High | 10,577 (49.2) | 9923 (49.4) | 654 (45.8) | |
| Smoking status, n (%) | | | | |
| Not current | 19,656 (91.4) | 18,444 (91.9) | 1212 (84.9) | <0.001 |
| Current | 1851 (8.6) | 1636 (8.1) | 215 (15.1) | |
| Drinking status, n (%) | | | | |
| Not current | 1837 (8.5) | 1659 (8.3) | 178 (12.5) | <0.001 |
| Current | 19,670 (91.5) | 18,421 (91.7) | 1249 (87.5) | |
| Diet, n (%) | | | | |
| Healthy | 7881 (36.6) | 7380 (36.8) | 501 (35.1) | 0.224 |
| Not healthy | 13,626 (63.4) | 12,700 (63.2) | 926 (64.9) | |
| Antidepressant use, n (%) | | | | |
| No | 19,886 (92.5) | 19,098 (95.1) | 788 (55.2) | <0.001 |
| Yes | 1621 (7.5) | 982 (4.9) | 639 (44.8) | |

Continuous variables were compared using the Mann–Whitney U test, and categorical variables were compared using the chi-square test.
*BMI* body mass index, *MET* metabolic equivalent of task.
All *P* values were based on two-sided tests.
[a]Age is presented as median (25th, 75th percentile).
[b]Physical activity: MET scores (range 0–21, positively correlated with weekly physical activity) below and above the cohort median represent low and high physical activity, respectively.

risk of depression among cancer survivors, no significant mediation effects or proportion mediated were observed for $NO_2$, $NO_x$, $PM_{2.5}$, $PM_{10}$ or APS in the associations between green space, blue space, natural environment and depression risk (Supplementary Figs. 2 and 3 with detailed displays in Supplementary Tables 5 and 6).

**Metabolites associated with the green space, natural environment, and air pollution**
We conducted elastic net and LASSO regression analyses on 249 metabolites (Supplementary data 1) to consistently identify the metabolic signatures of green space, natural environment, and APS (Figs. 5A and 6A). The results showed a combination of 45 metabolites associated with green space, 58 metabolites with natural environment, and 61 metabolites associated with APS, respectively (Supplementary

Figs. 4 and 5). The associations between these metabolites and depression among cancer survivors were presented in Supplementary Figs. 6–8. The green space-, natural environment-, and APS-related metabolic signatures were further calculated based on the coefficients from LASSO regression. Linear regression analysis revealed that the green space- and natural environment-related metabolic signatures were positively correlated with green space and natural environment, respectively, while both were negatively correlated with air pollutants (Supplementary Tables 7 and 8). The APS-related metabolic signature was positively correlated with APS ($\beta_{per\ 5\%\ increment} = 0.031$, 95% CI = 0.023–0.039) and negatively associated with both green space ($\beta_{per\ 5\%\ increment} = -0.012$, 95% CI = [−0.015, −0.009]) and natural environment ($\beta_{per\ 5\%\ increment} = -0.011$, 95% CI = [−0.013, −0.008]) (Supplementary Table 9).

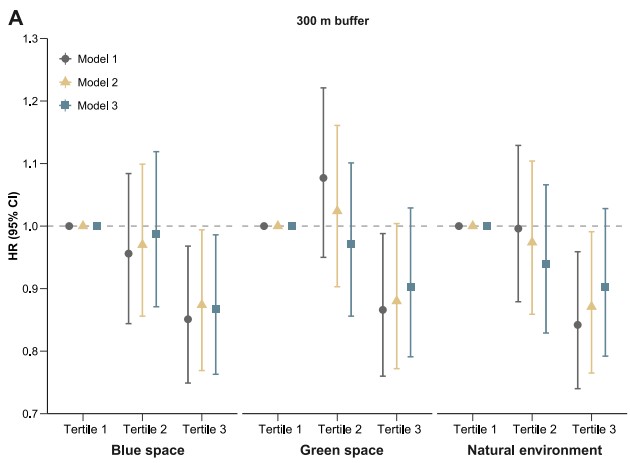

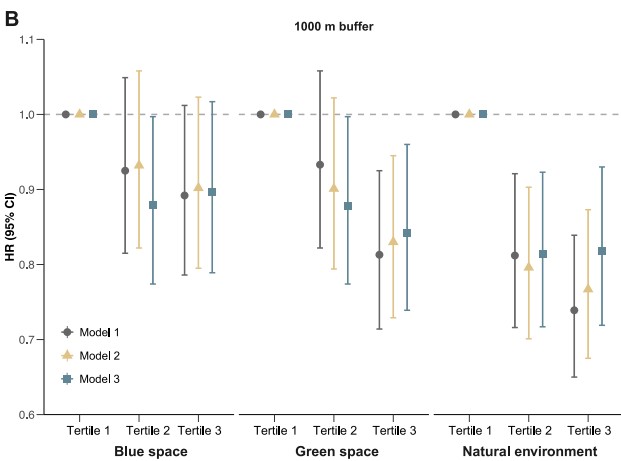

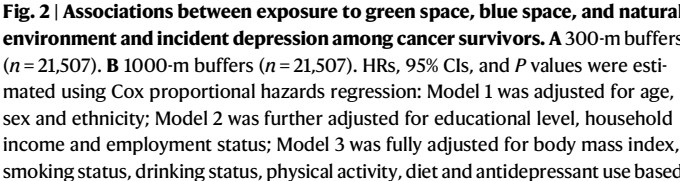

**Fig. 2 | Associations between exposure to green space, blue space, and natural environment and incident depression among cancer survivors. A** 300-m buffers ($n$ = 21,507). **B** 1000-m buffers ($n$ = 21,507). HRs, 95% CIs, and $P$ values were estimated using Cox proportional hazards regression: Model 1 was adjusted for age, sex and ethnicity; Model 2 was further adjusted for educational level, household income and employment status; Model 3 was fully adjusted for body mass index, smoking status, drinking status, physical activity, diet and antidepressant use based on Model 2. Tertile 1 was used as the reference group in Cox proportional hazards regression models. The shapes denote the point estimates of the HRs, while the vertical lines represent the corresponding 95% CIs. Detailed HRs (95 % CIs) and exact adjusted $P$ values are provided in Supplementary Tables 3 and 4. All $P$ values were two-sided. CI confidence interval, HR hazard ratio. Source data are provided as a Source Data file.

## Metabolic signatures and depression risk among cancer survivors

In Model 1, green space- and natural environment-related metabolic signatures were associated with a decreased risk of incident depression among cancer survivors, with HRs of 0.918 (95% CI = 0.871–0.969, $P$ = 0.002) and 0.923 (95% CI = 0.875–0.974, $P$ = 0.004) per standard deviation (SD) increment, respectively (Fig. 5B). Conversely, APS-related metabolic signature was associated with an increased risk of incident depression, with an HR of 1.074 (95% CI = 1.018–1.134, $P$ = 0.009) per SD increment (Fig. 6B). In Model 3, only the protective effect of natural environment-related metabolic signature remained significant (HR = 0.945, 95% CI = 0.895–0.998; $P$ = 0.042). When metabolic signatures were categorized into three groups (Low, Intermediate, high exposure), we found that metabolic signatures associated with green space and natural environment were generally inversely related to depression risk, whereas these linked to the APS were positively related; however, these associations did not reach statistical significance in the fully adjusted model.

## Mediating roles of metabolic signatures

Mediation analysis indicated that the green space- and natural environment-related metabolic signatures mediated the protective associations between green space, natural environment, and depression incidence among cancer survivors, with mediation effects of 1.163 (95% CI = 0.045–2.472, $P$ = 0.040) and 1.225 (95% CI = 0.081–2.561, $P$ = 0.036), respectively (Fig. 5C). However, only the mediation proportion of the green space-related metabolic signature was statistically significant, with a mediation proportion of 5.676% ($P$ = 0.042). For the APS-related metabolic signature, the mediation effect on the association between APS and depression among cancer survivors was not statistically significant, as neither the mediation effect nor the mediation proportion reached significance (Fig. 6C).

## Subgroups and sensitivity analyses

Stratified analyses indicated that the protective association between natural environment at 300 m buffer and depression risk was stronger among females compared to males ($P_{interaction}$ = 0.037). No other variables were found to modify the associations between green space, blue space, and natural environment at 300 m and 1000 m buffers and depression risk (Supplementary Figs. 9 and 10). Furthermore, stratified analyses indicated that the association between $NO_2$ and increased depression risk was more pronounced among cancer survivors with an unhealthy diet ($HR_{tertile\ 3\ vs.\ tertile\ 1}$ = 1.256, 95% CI = 1.065–1.481) compared to those with a healthy diet ($P_{interaction}$ = 0.039) (Supplementary Fig. 11). No other variables were found to modify the associations between air pollutants and depression risk.

In analyses stratified by cancer type, the results for breast cancer survivors—the largest subgroup—revealed a stronger protective association of green space and the natural environment with depression compared with the overall cancer survivor population (Supplementary Tables 10 and 11). For melanoma skin cancer patients, exposure to natural environment at 1000 m buffer was associated with a slightly reduced risk of depression ($HR_{tertile\ 3\ vs.\ tertile\ 1}$ = 0.780, 95% CI = 0.610–0.998) (Supplementary Tables 12 and 13). For non-melanoma skin cancer patients, exposure to blue space at 1000 m buffer was also associated with a slightly reduced risk of depression ($HR_{tertile\ 3\ vs.\ tertile\ 1}$ = 0.549, 95% CI = 0.319–0.946) (Supplementary Tables 14 and 15). Interestingly, among lung cancer survivors, high exposure to blue space at 1000 m buffer was significantly and inversely associated with depression risk (Supplementary Tables 16 and 17). Corresponding analyses for prostate and colorectal cancer survivors are presented in Supplementary Tables 18–21. We conducted a series of sensitivity analyses to enhance the robustness of our study (Supplementary Tables 22–29 and Supplementary data 2), including: (1) further adjustment for air pollution, cancer type, sleep pattern, and daily sun exposure (for skin cancer only); (2) exclusion of anti-depressant use from the covariates ($N$ = 21,507); (3) exclusion of depression cases within one year ($N$ = 21,437) or three years ($N$ = 21,279) post-cancer; (4) exclusion of deaths within ten years post-cancer ($N$ = 20,522); and (5) restriction to participants with more than ten years at their current residence prior to baseline ($N$ = 15,162). The results remained consistent with the main findings.

## Discussion

In this prospective cohort study based on a large-scale population from the UK Biobank, we found that higher levels of green space and natural environment at 1000 m buffer were significantly associated with a lower risk of depression among cancer survivors. In contrast, exposure to air pollutants ($NO_2$, $NO_x$, $PM_{2.5}$, and the APS) was associated with an increased risk of depression. Joint exposure to low levels

**Table 2 | Association between air pollutants and air pollution score and depression risk among cancer survivors**

| Air pollutants | Model 1 | | Model 2 | | Model 3 | |
|---|---|---|---|---|---|---|
| | HR (95% CI) | P value | HR (95% CI) | P value | HR (95% CI) | P value |
| $NO_2$ | | | | | | |
| Tertile 1 | Ref | | Ref | | Ref | |
| Tertile 2 | 1.132 (0.993–1.290) | 0.064 | 1.045 (0.916–1.192) | 0.512 | 1.026 (0.900–1.171) | 0.698 |
| Tertile 3 | 1.299 (1.142–1.477) | <0.001 | 1.207 (1.060–1.374) | 0.005 | 1.140 (1.001–1.299) | 0.048 |
| P trend | | <0.001 | | 0.004 | | 0.045 |
| Per 5% increment | 1.014 (1.007–1.021) | <0.001 | 1.010 (1.003–1.017) | 0.003 | 1.008 (1.001–1.015) | 0.028 |
| $NO_x$ | | | | | | |
| Tertile 1 | Ref | | Ref | | Ref | |
| Tertile 2 | 1.135 (0.994–1.295) | 0.061 | 1.060 (0.928–1.210) | 0.393 | 1.026 (0.898–1.172) | 0.704 |
| Tertile 3 | 1.334 (1.173–1.516) | <0.001 | 1.218 (1.070–1.386) | 0.003 | 1.143 (1.004–1.303) | 0.044 |
| P trend | | <0.001 | | 0.002 | | 0.039 |
| Per 5% increment | 1.007 (1.004–1.010) | <0.001 | 1.005 (1.002–1.008) | 0.003 | 1.003 (1.000–1.007) | 0.035 |
| $PM_{10}$ | | | | | | |
| Tertile 1 | Ref | | Ref | | Ref | |
| Tertile 2 | 1.014 (0.892–1.154) | 0.827 | 0.970 (0.853–1.104) | 0.649 | 0.892 (0.783–1.016) | 0.084 |
| Tertile 3 | 1.124 (0.990–1.276) | 0.070 | 1.083 (0.953–1.229) | 0.221 | 0.963 (0.847–1.094) | 0.560 |
| P trend | | 0.069 | | 0.214 | | 0.604 |
| Per 5% increment | 1.027 (1.000–1.055) | 0.053 | 1.014 (0.987–1.042) | 0.311 | 0.997 (0.970–1.025) | 0.833 |
| $PM_{2.5}$ | | | | | | |
| Tertile 1 | Ref | | Ref | | Ref | |
| Tertile 2 | 1.067 (0.935–1.217) | 0.335 | 0.998 (0.874–1.139) | 0.974 | 0.958 (0.839–1.094) | 0.529 |
| Tertile 3 | 1.298 (1.143–1.473) | <0.001 | 1.177 (1.036–1.338) | 0.012 | 1.107 (0.974–1.259) | 0.120 |
| P trend | | <0.001 | | 0.010 | | 0.099 |
| Per 5% increment | 1.134 (1.082–1.189) | <0.001 | 1.089 (1.038–1.143) | 0.001 | 1.050 (1.000–1.101) | 0.048 |
| Air pollution score | | | | | | |
| Tertile 1 | Ref | | Ref | | Ref | |
| Tertile 2 | 1.123 (0.984–1.282) | 0.086 | 1.050 (0.919–1.199) | 0.476 | 1.017 (0.890–1.162) | 0.803 |
| Tertile 3 | 1.342 (1.181–1.526) | <0.001 | 1.229 (1.080–1.398) | 0.002 | 1.152 (1.011–1.312) | 0.033 |
| P trend | | <0.001 | | 0.001 | | 0.029 |
| Per 5% increment | 1.073 (1.043–1.105) | <0.001 | 1.051 (1.020–1.083) | 0.001 | 1.035 (1.004–1.067) | 0.028 |

HRs, 95% CIs, and P values were estimated using Cox proportional hazards regression models (Models 1–3) (n = 21,507). Model 1 was adjusted for age, sex and ethnicity; Model 2 was further adjusted for educational level, household income and employment status; Model 3 was fully adjusted for body mass index, smoking status, drinking status, physical activity, diet and antidepressant use based on Model 2.

$NO_2$ nitrogen dioxide, $NO_x$ nitrogen oxides, $PM_{10}$ particulate matter with aerodynamic diameter ≤10 μm, $PM_{2.5}$ particulate matter with aerodynamic diameter <2.5 μm, Ref reference.
All P values were two-sided.

of air pollution and high levels of green space or natural environment amplified the protective effect against depression. Furthermore, we identified a comprehensive array of metabolites associated with green space, the natural environment, and APS, which were synthesized into corresponding metabolic signatures. The natural environment-related metabolic signature was significantly associated with a decreased risk of depression among cancer survivors, and the green space-related metabolic signature was found to mediate 5.676% of the association between green space exposure and incident depression in this population.

Previously, most studies have examined the relationship between levels of residential green space and cancer incidence or mortality. For instance, a U.S. cohort study involving cancer patients and survivors found that higher levels of residential green space were associated with a 6% reduction in cancer mortality risk[15]. Meanwhile, a study by Sakhvidi et al. conducted over 27 years on 19,408 individuals in France revealed an increased risk of all-site cancers with increased green space and proximity to agricultural lands and forests, while suggesting a potential protective role of green space for breast cancer[16]. Recently, one study established in China found that a higher level of residential

green space was associated with a lower risk of depression and anxiety disorders[17]. However, this study is cross-sectional and assessed depression and anxiety through self-reported questionnaire surveys, posing limitations in causal inference and outcome determination. Our study found that prolonged exposure to green space and the natural environment at 1000 m buffer was associated with a reduced risk of developing depression in a large-scale cohort. Notably, the exposure-response curves generally indicated that the impact of surrounding greenery on depression tended to be more significant with higher levels of exposure. This underscores the necessity for continued greenery and urban planning to safeguard the mental well-being of vulnerable populations such as cancer survivors. Interestingly, in analyses of survivors of specific cancer types, we found that among breast cancer survivors, green space and natural environment exhibited a stronger protective effect against depression than in the total cancer survivors and survivors of other cancer types. Previous studies have shown that urban green space is associated with a reduced risk of breast cancer[18], but their impact on the risk of incident depression among breast cancer patients has not yet been explored. Our findings suggest that targeted environmental interventions, such as increasing

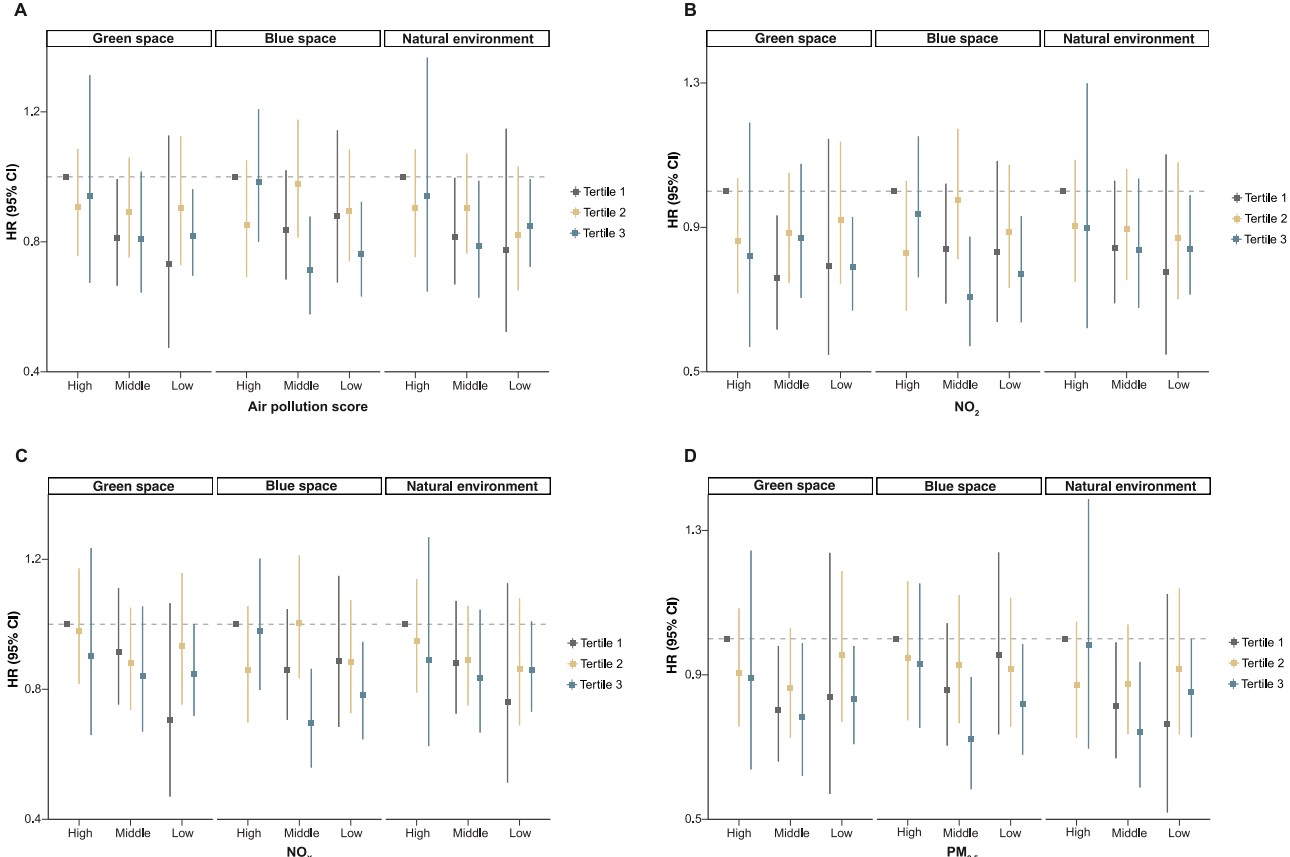

**Fig. 3 | Joint analysis of air pollution and green space, blue space, and natural environment at 300 m buffer on depression risk among cancer survivors. A** Air pollution score ($n = 21,507$). **B** $NO_2$ ($n = 21,507$). **C** $NO_x$ ($n = 21,507$). **D** $PM_{2.5}$ ($n = 21,507$). HRs, 95% CIs, and $P$ values were estimated using Cox proportional hazards regression with adjustment for covariates in Model 3 (age, sex and ethnicity, educational level, household income and employment status, body mass index, smoking status, drinking status, physical activity, diet and antidepressant use). Gray, yellow, and blue correspond to Tertile 1, Tertile 2, and Tertile 3 of green space, blue space, and natural environment exposure, respectively. The $x$-axis represents categories (high, medium, low) of air pollution (including the air pollution score, $NO_2$, $NO_x$, and $PM_{2.5}$), where "high", "medium", and "low" correspond to Tertile 3, Tertile 2, and Tertile 1 of each air pollution measure. Squares denote the point estimates of the HRs, while the vertical lines represent the corresponding 95% CIs. Exact $P$ values are provided in the Source Data file, and all $P$ values were two-sided. CI confidence interval, HR hazard ratio, $NO_2$ nitrogen dioxide, $NO_x$ nitrogen oxides, $PM_{2.5}$ particulate matter with aerodynamic diameter ≤2.5 μm. Source data are provided as a Source Data file.

access to green space, could be incorporated into supportive care strategies for breast cancer survivors to reduce the risk of depression. Exposure to green space and natural environment has been linked to stress reduction, improved mood, and enhanced overall well-being[19,20], mechanisms that may be particularly beneficial for this specific population.

Furthermore, our results indicated that high levels (tertile 3) of $NO_2$ and $NO_x$ were associated with a 14.0% and 14.3% increased risk of depression among cancer survivors, respectively, while higher APS was linked to a 15.2% increased risk. Each 5% increment in $PM_{2.5}$ was also found to be associated with an increased risk of depression among cancer survivors. A previous study among Korean adult cancer survivors reported that exposure to $PM_{10}$ was associated with an increased risk of depressive symptoms and suicidal ideation, whereas no significant association was observed for $NO_2$[21]. Notably, although this study applied propensity score matching between cancer survivors and the general population, its cross-sectional design limited the ability to infer causal relationships between air pollution and depression. In addition, the lack of data on $NO_x$ and $PM_{2.5}$ further hindered a comprehensive assessment of air pollutant exposures. In our study, we did not observe a significant association between $PM_{10}$ exposure and depression risk among cancer survivors. Compared with $PM_{2.5}$, $NO_2$ and $NO_x$, the effect of $PM_{10}$ on depression appeared to be weaker, potentially requiring a longer exposure duration to trigger its impact[22].

Our study further confirmed the detrimental effects of air pollutants on depression risk among cancer survivors, highlighting that controlling air pollution may offer additional mental health benefits, especially for vulnerable groups. Moreover, our findings indicate that combined exposure to multiple air pollutants may exacerbate the risk of depression among cancer survivors, highlighting the need for integrated environmental governance. The cumulative effects of air pollution on mental health underscore that regulatory efforts should not focus on individual pollutants in isolation but consider the broader environmental mixture. Governments and public health authorities could prioritize interventions that reduce overall air pollution levels, improve urban planning, and enhance access to clean and green spaces, particularly in areas with high concentrations of vulnerable populations such as cancer survivors. These policy measures could help mitigate the mental health burden associated with environmental exposures and contribute to broader strategies for improving population well-being.

Another key finding was that high levels of green, blue, and natural spaces, in combination with low levels of air pollution, conferred protective effects against depression risk among cancer survivors. Interestingly, despite the negative correlation between air pollution and green space or natural environment, mediation analysis suggested that air pollutants did not significantly mediate these associations, indicating their independent contributions to depression risk in this

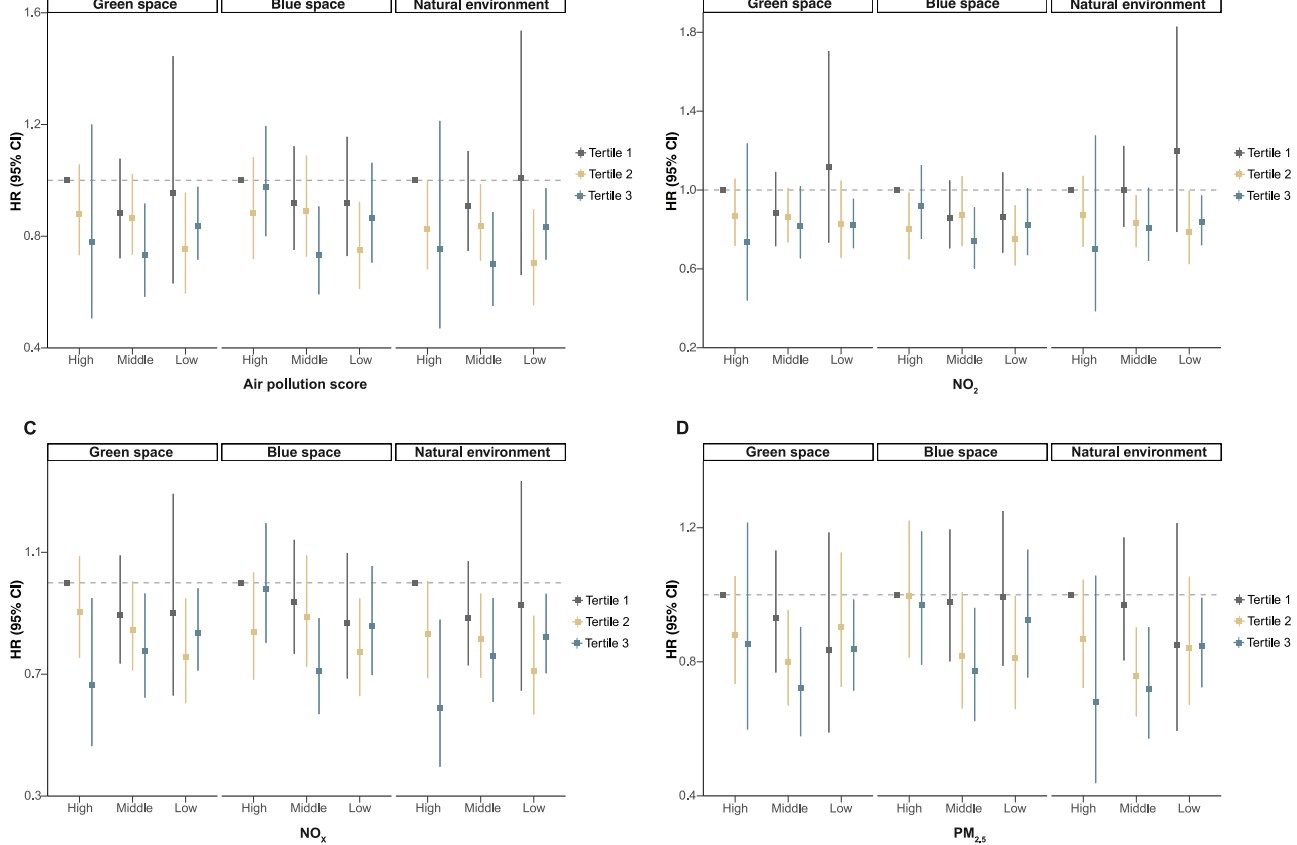

**Fig. 4 | Joint analysis of air pollutants and green space, blue space, and natural environment within 1000 m buffer on depression risk among cancer survivors.** **A** Air pollution score ($n = 21{,}507$). **B** NO$_2$ ($n = 21{,}507$). **C** NO$_x$ ($n = 21{,}507$). **D** PM$_{2.5}$ ($n = 21{,}507$). HRs, 95% CIs, and $P$ values were estimated using Cox proportional hazards regression ($n = 21{,}507$) with adjustment for covariates in Model 3 (age, sex and ethnicity, educational level, household income and employment status, body mass index, smoking status, drinking status, physical activity, diet and antidepressant use). Gray, yellow, and blue correspond to Tertile 1, Tertile 2, and Tertile 3 of green space, blue space, and natural environment exposure, respectively. The

$x$-axis represents categories (high, medium, low) of air pollution (including the air pollution score, NO$_2$, NO$_x$, and PM$_{2.5}$), where "high", "medium", and "low" correspond to Tertile 3, Tertile 2, and Tertile 1 of each air pollution measure. Squares denote the point estimates of the HRs, while the vertical lines represent the corresponding 95% CIs. Exact $P$ values are provided in the Source Data file, and all $P$ values were two-sided. CI confidence interval, HR hazard ratio, NO$_2$ nitrogen dioxide, NO$_x$ nitrogen oxides, PM$_{2.5}$ particulate matter with aerodynamic diameter $\leq 2.5\,\mu m$. Source data are provided as a Source Data file.

population. A recent review estimated that increasing residential greenness by one SD would reduce air pollution by only 0.8%, supporting our finding that its capacity to mitigate air pollution is limited[23]. For cancer survivors, urban green space provides social interaction, stress relief, and opportunities for socializing[24]. These spaces' key traits, such as cleanliness, esthetics, nature, interaction, safety, and freedom, are highly beneficial for relieving stress, preventing depression, and reducing anxiety. Establishing a positive connection with nature benefits lifelong mental health by strengthening social bonds and promoting well-being in accessible green space[25]. Previous studies have indicated that women are more susceptible to experiencing common mental disorders such as depression and anxiety compared to men[26]. Additionally, as one of the most burdensome cancers among women, breast cancer may pose significant psychological challenges for long-term survivors[27]. In our study, breast cancer accounted for the largest number of cases ($n = 7365$) among cancer survivors with a survival time of more than five years. In the analysis of breast cancer survivors, we observed that long-term exposure to green space was associated with a 21.9% to 23.9% reduction in the risk of depression.

The pathophysiology of depression is complex, characterized by significant molecular alterations and dysregulation across multiple pathways. Metabolomics, which captures simultaneously hundreds of

molecular changes, offers a comprehensive approach to evaluating the pathophysiology of depression[28]. Our study identified a comprehensive panel of metabolites associated with green space, natural environment, and the APS, which were subsequently integrated into metabolic signatures. The results showed that metabolic signatures related to the natural environment and green space were associated with reduced depression risk, while the APS-related metabolic signature was linked to an increased risk among cancer survivors. Metabolic signatures related to the natural environment and green space act as potential mediators in their associations with depression among cancer survivors. Specifically, we found that three amino acids-leucine, glutamine, and tyrosine-were consistently positively associated with green space and the natural environment. Previous studies have shown that leucine supplementation can alleviate social avoidance and depressive-like behaviors in mice[29]. The glutamate–glutamine cycle and glutamatergic neurotransmission play critical roles in normal brain function, highlighting the importance of maintaining glutamine homeostasis in the brain. A deficiency of neuronal glutamine in the medial prefrontal cortex of rodents has been linked to depressive behaviors and mild cognitive impairment[30]. Furthermore, Mendelian randomization evidence supports a potential causal relationship between downregulation of glutamine and major depressive disorder, anhedonia, fatigue, and low mood[31]. For tyrosine, previous studies

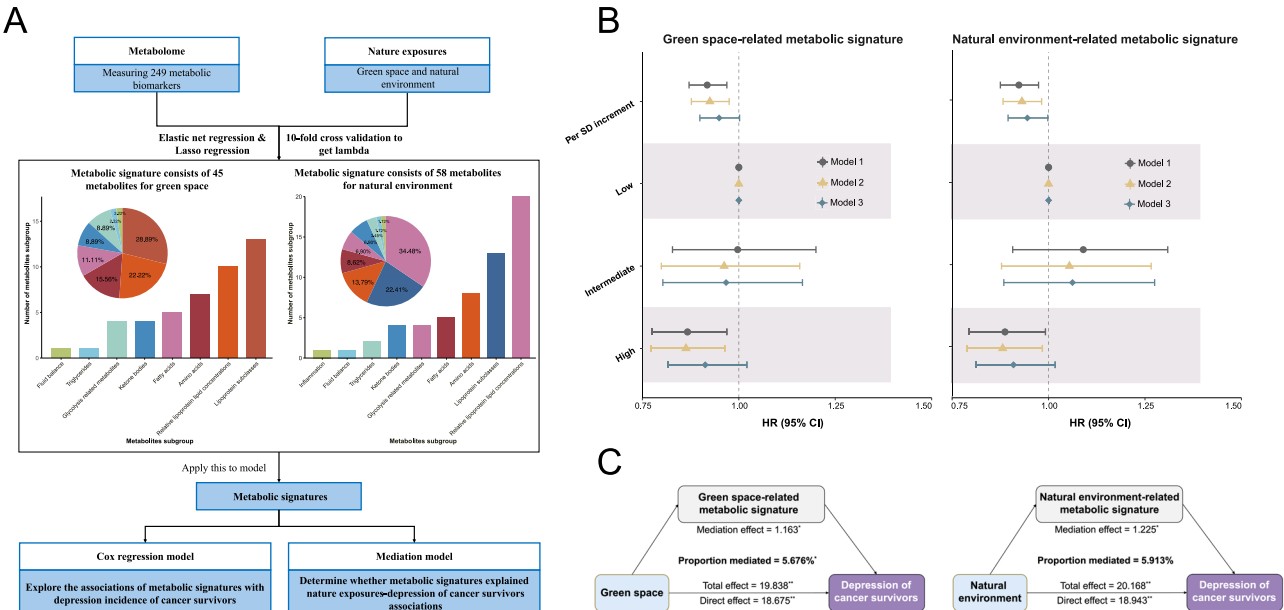

**Fig. 5 | Associations of green space and natural environment–related metabolic signatures with depression risk among cancer survivors and their mediating effects. A** Development of green space- and natural environment-related metabolic signature scores. Elastic net and LASSO penalized regression models with ten-fold cross-validation were employed for variable selection to identify relevant metabolites. Metabolic signatures related to green space and the natural environment were derived using a weighted approach, with weights defined by coefficients from LASSO penalized regression used for variable selection. **B** Associations of green space- and natural environment-related metabolic signatures with risk of incident depression among cancer survivors (*n* = 21,507). HRs, 95% CIs, and *P* values were estimated using Cox proportional hazards regression with adjustment for covariates in Model 1–3. Model 1 was adjusted for age, sex and ethnicity; Model 2 was further adjusted for educational level, household income and employment status; Model 3 was fully adjusted for body mass index, smoking status, drinking status, physical activity, diet and antidepressant use based on Model 2. Metabolic signatures were categorized into quintiles: low (lowest quintile), moderate (quintiles 2–4), and high (highest quintile). Symbols: HRs; Vertical lines: 95% CI. **C** The estimated mediation proportions of metabolic signatures on the associations of green space and the natural environment with incident depression among cancer survivors. Mediation analysis was performed using an accelerated failure time model with covariates adjusted as in Model 3. Exact *P* values are provided in the Source Data file, and all *P* values were two-sided. CI confidence interval, SD standard deviation. Source data are provided as a Source Data file.

have reported significantly reduced levels in the blood and urine of patients with depression[32], with concentrations showing a significant negative correlation with Hamilton Depression Rating Scale (HAMD) scores[33]. Additionally, 3-Hydroxybutyrate, a ketone body that was consistently positively associated with green space and the natural environment, has been shown to significantly reduce anxiety-like behaviors in rats when administered through long-term dietary supplementation[34]. Interestingly, for large Very Low-Density Lipoprotein (VLDL), we found that cholesteryl esters in chylomicrons and extremely large VLDL were significantly negatively associated with exposure to natural environment and green space, whereas the phospholipids-to-total lipids ratio in large VLDL showed a significant positive association. VLDL cholesterol has previously been identified as part of the metabolomic signature of depression, and positively associated with depressive symptoms[35]. In contrast, phospholipids, as structural core components of VLDL-particularly phosphatidylcholines and phosphatidylethanolamines-are essential constituents of cell membranes in the central nervous system, playing critical roles in maintaining neurotransmitter systems, brain function, and emotional regulation[36]. These findings further underscore the importance of incorporating metabolic health into the understanding of the benefits of nature exposure.

It is worth mentioning that, as in most epidemiological studies, the use of single-time-point measurements to determine participants' circulating metabolite levels has raised concerns about whether this approach can reasonably reflect stable metabolite levels. Several epidemiological studies have demonstrated that certain metabolites exhibit high or low within-individual reproducibility[37–41]. Specifically, a recent study evaluated the within-individual reproducibility of plasma metabolites in 61 Black breast cancer survivors from the Women's

Health Circle follow-up study[42]. The study found that the within-individual reproducibility of plasma metabolites in breast cancer survivors over the course of one year was generally acceptable, suggesting that single-time-point measurements could be useful in assessing the associations between metabolites and breast cancer outcomes. We should acknowledge that changes in cancer metabolite levels are inevitable, as the metabolic reprogramming of tumor cells plays a critical role in cancer initiation, proliferation, and progression[43]. However, by restricting our study population to long-term cancer survivors, whose disease status is relatively stable, we were able to mitigate this potential source of bias. Emerging research highlights the exciting role played by the unique microenvironment formed by the tumor microenvironment in shaping the tumor's metabolic landscape[44,45]. Further research is needed on the metabolic stability/reproducibility in cancer patients.

Several limitations of this study should be acknowledged. First, participants in the UK Biobank were recruited on a voluntary basis, potentially resulting in differences in characteristics compared to the general population, thus introducing selection bias. Second, while the spatial distribution of land use characteristics, including green space and blue space, is relatively stable in the UK over short periods, estimating exposure solely based on participants' baseline residence may introduce bias. Although exposure was assessed at fixed residential addresses without accounting for participants' activity patterns or residential mobility, sensitivity analyses restricted to individuals who had lived at their current address for more than 10 years suggested a consistently robust direction of the associations. However, it should be noted that the temporal gap between the air pollution estimates and the cohort recruitment period may have introduced potential exposure misclassification due to the use of fixed-year models. Third, using

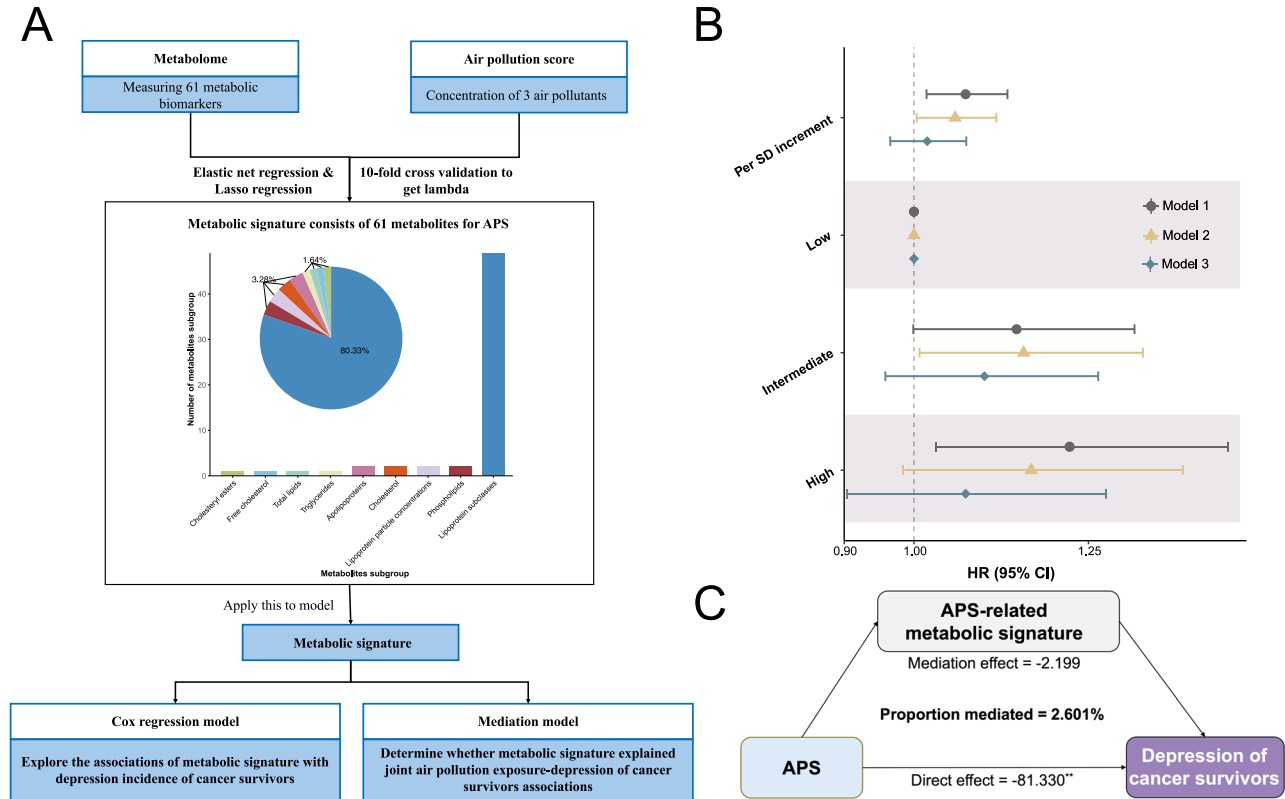

**Fig. 6 | Associations of air pollution–related metabolic signature with depression risk among cancer survivors and their mediating effects.**
**A** Development of APS-related metabolic signature score. Elastic net and LASSO penalized regression with ten-fold cross-validation were used to identify relevant metabolites, and an APS-related metabolic signature was derived using a weighted approach based on LASSO regression coefficients. **B** Associations of APS-related metabolic signature with risk of incident depression among cancer survivors ($n = 21,507$). HRs, 95% CIs, and $P$ values were estimated using Cox proportional hazards regression with adjustment for covariates in Model 1–3. Model 1 was adjusted for age, sex and ethnicity; Model 2 was further adjusted for educational level, household income and employment status; Model 3 was fully adjusted for

body mass index, smoking status, drinking status, physical activity, diet and anti-depressant use based on Model 2. Metabolic signatures were categorized into quintiles: low (lowest quintile), moderate (quintiles 2–4), and high (highest quintile). The shapes denote the point estimates of the HRs, while the horizontal lines represent the corresponding 95% CIs. **C** The estimated mediation proportions of metabolic signature on the association of APS with incident depression among cancer survivors. Mediation analysis was performed using an accelerated failure time model with covariates adjusted as in Model 3. Exact $P$ values are provided in the Source Data file, and all $P$ values were two-sided. APS air pollution score, CI confidence interval, LASSO Least Absolute Shrinkage and Selection Operator, SD standard deviation. Source data are provided as a Source Data file.

diagnosis rather than precise symptom onset to define depressive disorder may introduce bias into the estimates. Fourth, individual nature exposures may be influenced by extraneous factors such as time spent outdoors and workplace exposure duration. Notably, considering that patients with skin cancer may interact with green or blue space differently due to concerns about sun exposure, we further adjusted for daily sun exposure duration, which still suggested a consistently positive direction of the associations. Fifth, due to data limitations in the UK Biobank, our analysis was restricted to 300 m and 1000 m buffers. Future studies employing alternative buffer sizes for residential green space and natural environment (e.g., 500 m) may provide further insights. Sixth, various environmental and meteorological factors, such as wind speed, rainfall, and seasonal variations, may influence the dispersion and accumulation of air pollutants. Therefore, the observed mediating role of air pollution should be interpreted with caution. Seventh, the reliance on baseline metabolic biomarker measurements limits the ability to explore dynamic changes in metabolites and the stability of metabolomic profiles over time, which may affect the validity of mediation analyses and their implications for depression risk among long-term cancer survivors. Finally, despite extensive confounder adjustment, residual confounding may persist. For instance, cancer stage-a key prognostic factor-could not be assessed due to its unavailability in the UK Biobank.

In conclusion, exposure to green space and the natural environment was associated with a decreased risk of depression among cancer survivors, while exposure to air pollutants ($NO_2$, $NO_x$, $PM_{2.5}$, and the APS) was associated with an increased risk. Notably, the combination of lower air pollution and higher levels of green space or natural environment appeared to be associated with a greater reduction in depression risk. Furthermore, our findings suggest that the beneficial effects of residential green space and the natural environment on depression may be partially mediated through alterations in the metabolomic signatures. These results provide valuable insights into strategies for safeguarding mental health in cancer survivors and other vulnerable populations.

## Methods

### Study design and participants

The UK Biobank study was approved by the North West Multi-Centre Research Ethics Committee, and all participants provided written informed consent (https://www.ukbiobank.ac.uk/learn-more-about-uk-biobank/about-us/ethics). The study protocol is available online (https://www.ukbiobank.ac.uk). Data were derived from the UK Biobank, which is an ongoing prospective cohort study. Initially, more than 500,000 participants (aged 37–73 years) were recruited during baseline (2006–2010) from 22 research centers across the UK

(England, Wales and Scotland). After obtaining electronic consent for the use of de-identified data, every participant completed a self-completed touch-screen questionnaire, a computer-assisted interview. Participants also consented to a range of physical measures, as well as sampling assays and genotyping.

In this study, the diagnosis information of cancer (excluding non-melanoma skin cancer) was provided by the Medical Research Information Service of the National Health Service (NHS) Information Centre (residents in England and Wales) and the Information Services Division of NHS Scotland (residents in Scotland) (see details at https://biobank.ndph.ox.ac.uk/showcase/showcase/docs/CancerLinkage.pdf) using the International Statistical Classification of Diseases and Related Health Problems 9th (ICD-9) (140–208, except 173 [i.e., non-melanoma skin cancer]) and 10th edition (ICD-10) (C00-C96, except C44 [i.e., non-melanoma skin cancer]). The detailed participant inclusion and exclusion process was illustrated in Supplementary Fig. 12. The cohort initially included 93,677 participants with cancer. We excluded individuals diagnosed with cancer after baseline ($n = 65,195$), those with less than five years of cancer survivorship ($n = 714$)[3,46], participants with missing data on green space, blue space, and natural environment exposures ($n = 3393$), those with a diagnosis of depression prior to baseline ($n = 303$), and those diagnosed with other psychiatric disorders ($n = 2565$). After these exclusions, a total of 21,507 cancer survivors were included in the final analysis. Supplementary Figs. 13 and 14 show the distribution of cancer types stratified by survival time and diagnosis-to-enrollment interval.

## Assessment of exposure

Environmental exposures assessed in this study include percentage coverage of the natural environment, green space, and blue space. The percentages of residential green space and blue space, classified as "greenspace" and "water", respectively, were calculated as proportions of the total area of all land-use types. Residential green space and blue space were calculated using land use data from the 2005 Generalized Land Use Database (GLUD) for England at 2001 Census Output Areas (COA) level and were consistent with previous studies[47–49]. Data on natural environment were also linked using the 2007 Land Cover Map (LCM) data of the Centre for Ecology and Hydrology (CHE)[47], which has been used in previous studies as well[50]. Percentages of residential green space, blue space and natural environment (home location buffer classed as "green space", "blue space", and "natural environment") at 300 m and 1000 m buffers were calculated (Supplementary Fig. 15). The LCM includes 23 land cover classes; in our study, Classes 1–21 were classified as natural environment, while Classes 22 and 23 (buildings and gardens) were excluded[10]. Thus, privately owned lands like residential gardens were not considered part of the natural environment.

Four air pollutants, $NO_x$, $NO_2$, $PM_{10}$, and $PM_{2.5}$, were calculated by the UK Biobank by using land-use regression (LUR) models developed by the ESCAPE (European Study of Cohorts for Air Pollution Effects) project[51,52]. Following the UK Biobank recommendations, air pollutant concentrations from the year 2010 were uniformly applied in our analysis. To capture the joint exposure to various air pollutants, we created a weighted APS by summing up concentrations of 3 ambient air pollutants ($PM_{2.5}$, $NO_2$, and $NO_x$), weighted by the multivariable-adjusted (model 3) risk estimates ($\beta$ coefficients)[53]. The $\beta$ coefficients were derived from the final model, with each air pollutant included as a single independent variable, one at a time. The equation was: air pollution score = $(\beta_{PM2.5} \times PM_{2.5} + \beta_{NO2} \times NO_2 + \beta_{NOx} \times NO_x) \times (3/sum$ of the $\beta$ coefficients). In addition to being primary exposures, air pollutants were also treated as potential mediators to investigate their mediating role in the associations between green space, blue space, and natural environment with depression risk among cancer survivors. Furthermore, they were included as additional covariates beyond

those in Model 3 for sensitivity analyses. Further details are provided in the "Statistical analysis" section.

## Ascertainment of outcome

Follow-up in the UK Biobank was carried out through linkage to the health administrative datasets of the National Health Service (NHS). In this study, incident depression cases were identified using hospital inpatient records from the UK Biobank (Data-Field 41270), based on the International Classification of Diseases, 10th Revision (ICD-10) codes F32 (including F32.0, F32.1, F32.2, F32.3, F32.8, and F32.9) and F33 (including F33.0, F33.1, F33.2, F33.3, F33.4, F33.8, and F33.9). Time-to-event was measured from the date of recruitment to the first occurrence of a depression diagnosis, death, loss to follow-up, or the end of follow-up (October 31, 2022 for England; August 31, 2022 for Scotland; and March 31, 2022 for Wales), whichever came first.

## Nuclear magnetic resonance (NMR) metabolomics measurement

Baseline plasma samples, obtained at the time of participant enrollment in the UK Biobank (2006–2010), were utilized for high-throughput NMR metabolomic profiling. These samples were analyzed by Nightingale Health, which measured 251 metabolic biomarkers in plasma from a randomly selected subset of approximately 280,000 participants from the UK Biobank. In contrast to the timing of sample collection, metabolomic profiling was performed in two distinct phases—Phase 1 (June 2019 to April 2020) and Phase 2 (April 2020 to June 2022)—utilizing state-of-the-art spectrometric techniques in Finland. The process involved comprehensive profiling procedures and rigorous quality control measures, as previously described[54,55]. The biomarkers investigated span a range of metabolic pathways, including lipoprotein lipids across 14 subclasses, fatty acid compositions, fatty acids, and low-molecular-weight metabolites. In our study, we considered 249 available metabolic biomarkers. Glucose-lactate and spectrometer-corrected alanine were excluded due to their availability only in the phase 2 release, which resulted in a substantial discrepancy in sample size compared to the other 249 biomarkers. The UK Biobank and Nightingale Health applied rigorous quality control procedures and developed statistical approaches to monitor measurement consistency, exclude unreliable biomarkers, and correct for technical variation, thereby ensuring high reproducibility and minimizing batch effect–related biases in NMR metabolomics data[56], with specific details provided in the Supplementary Method.

## Assessment of covariates

Covariates included a range of demographic, lifestyle and medication-related factors: age (<65 years; ≥65 years), sex (female; male), ethnicity (White; non-White), educational level (university or college; other), employment status (currently employed; unemployed), household income (high: ≥£52,000; middle: £18,000–£51,999; low: <£18,000), body mass index (BMI) (normal, <25.0 kg/m²; overweight, 25.0–29.9 kg/m²; obese, ≥30.0 kg/m²), smoking status (current; not current), drinking status (current; not current), physical activity (low; high), diet (healthy; not healthy) and antidepressant use (yes; no). Diet was assessed using a healthy diet score ranging from 0 to 5, with one point awarded for each of the following favorable dietary behaviors: consuming ≥4 tablespoons of vegetables per day, ≥3 pieces of fruit per day, fish at least twice per week, unprocessed red meat no more than twice per week, and processed meat no more than twice per week. A score of ≥4 indicated a healthy diet, while a score below 4 indicated an unhealthy diet. Additionally, daily sun exposure duration (hours/day, continuous variable) and sleep pattern (classified as healthy or unhealthy based on a healthy sleep score derived from sleep traits including insomnia, sleep duration, chronotype, daytime sleepiness, and snoring [range 0–5], with a healthy sleep pattern defined as a sleep

score ≥4) were defined. Additional information regarding the covariate definition and classification is provided in Supplementary data 3.

## Statistical analyses

Baseline characteristics were presented as median (25th–75th percentile) for continuous variables and frequency (percentage) for categorical variables. Differences in baseline characteristics between participants with and without depression were assessed using the Mann–Whitney U test for continuous variables and the chi-square test for categorical variables. Pearson correlation coefficients were used to examine correlations between environmental exposures and air pollutants, and point-biserial correlation coefficients were used to assess the associations between environmental exposures, air pollutants, and depression. Missing data were handled using Multiple Imputation by Chained Equations (MICE).

We performed Cox proportional hazards regression analyses to examine the associations of exposure to residential green space, blue space, and natural environment–as well as to air pollutants ($PM_{2.5}$, $PM_{10}$, $NO_2$, $NO_x$, and the APS)–with the risk of depression. We constructed three models by sequentially adding covariates: Model 1 was adjusted for age, sex and ethnicity; Model 2 was further adjusted for educational level, household income and employment status; and Model 3 was fully adjusted for all aforementioned variables as well as BMI, smoking status, drinking status, physical activity, diet and antidepressant use. Participants were categorized into tertiles based on exposure levels, with the lowest tertile serving as the reference group. Exposures were also analyzed as continuous variables, and HRs were estimated per 5% increment. The RCS model was employed to explore potential exposure–response relationships. To explore potential interactions, we used the "mediation" package in R based on the accelerated failure time (AFT) model (via survreg () function) to examine whether air pollutants mediate the associations between green space, blue space, natural environment, and depression risk.

For metabolomics analysis, we first assessed the associations between the metabolites and green space and natural environment by using multivariable linear regression models, adjusting for covariates included in Model 3. A False Discovery Rate (FDR)-corrected $P < 0.05$ was considered statistically significant. Missing values for the 249 metabolites, assumed to be related to the detection limit, were imputed as half of the minimum detectable value, consistent with a previous study[56]. To construct metabolic signatures associated with green space, natural environment and APS, we followed the previous studies[57,58], employing both elastic net and LASSO regression to identify relevant metabolites. The 10-fold cross-validation process was employed to ensure accuracy by dividing the data into 10 equal subsets. In each iteration, the model was trained on 9 subsets and tested on the remaining subset. Metabolic signatures indicative of green space, the natural environment, and APS were then derived using a weighted approach based on the coefficients from the LASSO regression. Before further analysis, the scores of metabolic signatures were standardized using z-scores (mean = 0, SD = 1) and then stratified into quintiles: low (lowest quantile), moderate (quantile 2–4), and high (highest quantile) exposure[57]. The associations between metabolic signatures and the risk of depression among cancer survivors were examined using Cox regression based on Models 1–3. The population attributable fraction (PAF) was calculated to estimate the proportion of depression cases among cancer survivors that could potentially be prevented if specific risk factors were eliminated. Finally, metabolic signatures were evaluated as potential mediators using mediation analysis conducted with the "mediation" package, which is based on the AFT model implemented through the survreg () function (Supplementary Fig. 16).

Stratified analyses were performed by age (<65 years vs. ≥65 years), sex (female vs. male), ethnicity (White vs. non-White), educational level (university or college vs. other), BMI (normal vs. overweight vs. obese), employment status (currently unemployed vs. currently employed), household income (high: ≥£52,000 vs. middle: £18,000–£51,999 vs. low: <£18,000), physical activity (low vs. high), smoking status (current vs. not current), drinking status (current vs. not current), diet (healthy vs. not healthy) and antidepressant use (yes vs. no). Interaction terms between these stratifying factors and exposures were incorporated into the Cox models to examine potential effect modifications, and P-values for interaction were calculated.

A series of sensitivity analyses was carried out to evaluate the robustness of these findings. First, we conducted separate analyses for the major cancer subtypes, including breast, melanoma skin, prostate, and colorectal cancers, as well as for specific cancer types such as lung cancer and non-melanoma skin cancer. Then, we further repeated the primary analysis by (1) further adjusting for air pollutants ($PM_{2.5}$, $PM_{10}$, $NO_2$, and $NO_x$), both individually and simultaneously and the APS, respectively; (2) additionally adjusting for cancer type, sleep pattern, and daily sun exposure duration (only for two types of skin cancer), respectively; (3) removing antidepressant use from the covariate set; (4) excluding survivors who used antidepressants at baseline, were diagnosed with incident depression within one or three years post-cancer, to minimize the impact of reverse causality; (5) excluding death cases within ten years post-cancer; (6) restricting the analysis to individuals who had lived at their current address for over 10 years prior to baseline.

All statistical analyses were conducted using R 4.2.1. A two-tailed P value of <0.05 defined statistical significance.

## Statistics & reproducibility

No statistical method was used to predetermine the sample size. The study was not randomized. The investigators were not blinded to allocation during the study or outcome assessment. We excluded individuals diagnosed with cancer after baseline, those with less than five years of cancer survivorship, participants with missing data on green space, blue space, and natural environment exposures, those with a diagnosis of depression prior to baseline, and those diagnosed with other psychiatric disorders, to investigate the associations between environmental exposures and the risk of depression. All replication attempts were successful.

## Reporting summary

Further information on research design is available in the Nature Portfolio Reporting Summary linked to this article.

## Data availability

The UK Biobank patient-level data are available under restricted access for bona fide researchers; access can be obtained by applying at http://ukbiobank.ac.uk/register-apply/. Raw data are protected and are not available due to data privacy laws. All participants provided informed written consent to take part in the study. Ethics approval for the UK Biobank was granted by the North West Multi-Centre Research Ethics Committee in 2006 and was updated regularly after that (https://www.ukbiobank.ac.uk/learn-more-about-uk-biobank/about-us/ethics). This study was conducted after approval by the UK Biobank under application reference 724597. Source data are provided with this paper.

## Code availability

The analysis code is available on GitHub (https://github.com/garic019/NCOMMS-25-08485-zjh) and has been archived on Zenodo (https://zenodo.org/records/17684180).

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

## Acknowledgements

This research was conducted using data from the UK Biobank study (Project ID: 724597). We would like to thank all UK Biobank participants and the UK Biobank management team. X.L. was supported by the National Natural Science Foundation of China (No. 82204019 and 82470543), Healthy Zhejiang One Million People Cohort (No. K-20230085), and the Zhejiang Provincial Clinical Research Center for CANCER (No. 2022E50008 and 2024ZY01056), and Key R&D Program of Zhejiang (No. 2023C03049). Y.D. was supported by the Key Research and Development Program of Zhejiang Province (No. 2024C03143). J.Z. was supported by the Fundamental Research Funds for the Central Universities (No. 2024BSSXM20) and Key Laboratory of Integrated Care for Geriatric Chronic Diseases, Yunnan Provincial Education Department (No. 2024HTHLYB05). L.X. was supported by the Key Laboratory of Integrated Care for Geriatric Chronic Diseases, Yunnan Provincial Education Department (2024HTHLYB01). Figure 1 was created in BioRender under a licensed agreement (Agreement number: NL29BITH86; https://BioRender.com/v51frxu).

## Author contributions

J.Z., formal analysis (lead), writing-original draft (lead), conceptualization (lead), methodology (equal); J.Y., software (lead), visualization (lead), writing-original draft (equal); E.X., methodology (lead), writing-review and editing (equal); L.X., writing-review and editing (equal), visualization (equal); J.S., writing-review and editing (equal); S.Z., methodology (supporting), writing-review and editing (supporting); T.L., visualization (supporting). H.C., software (supporting); Z.S., writing-review and editing (equal); W.W., writing-review and editing (equal); Y.H., writing-review and editing (equal), methodology (supporting); Y.D., conceptualization (lead), writing-review and editing (lead), supervision (equal); X.L., conceptualization (lead), writing-review and editing (lead), supervision (lead), methodology (equal). All authors critically reviewed the manuscript for important intellectual content. X.L. and Y.D. are the study guarantors. The corresponding authors attest that all listed authors meet authorship criteria and that no others meeting the criteria have been omitted.

## Competing interests

The authors declare no competing interests.
