## [Transparent Peer Review File · Nature Communications]

Residential Green Space, Air Pollution, and Related Metabolites in Association with Depression Among Cancer Survivors

Corresponding Author: Dr Xue Li

Version 0:

Reviewer comments:

Reviewer #1

(Remarks to the Author)

Comments to Author:

The study led by Zhao et al. aimed to assess the longitudinal associations of residential green space, blue space and natural environment with depression risk among cancer survivors, in addition to the potential mediating role of four air pollution variables and plasma metabolites. While this is an important topic of significance to public health, and the sample size of the study is large due to the usage of UK Biobank, there are many major concerns regarding the study design and methodology of the presented study, resulting in flaws in the method and invalidity of study findings. Below are detailed comments and suggestions.

Major Concerns:

1. Temporal misalignment of the exposure, mediator, and outcome. A fundamental assumption of the presented study is that metabolic signatures would serve as a biological mediation in the association between green space and incident of depression among cancer survivors. Additionally, they also examine how air pollution mediates these associations. A major concern is the temporal misalignment between the exposure, mediator, and outcome, which resulted in a major flaw of the study design and led to the invalidity of the study findings. Specifically, according to the method section, assessment of the green space was conducted using data in 2005 and 2007, while the recruitment of the cohort started between 2006 to 2010; regarding the outcome assessment, as the authors stated, "incident depression cases were identified from hospital admissions, primary care records, and self-reports... Time-to-event was measured from recruitment until the first occurrence of depression diagnosis, death, loss to follow-up, or study conclusion, whichever occurred first...". Meanwhile, for the metabolomics mediator, measurement of 251 metabolic biomarkers for EDTA plasma samples collected from a randomly selected subset of approximately 280,000 UKB participants between June 2019 and April 2020 (Phase 1) and April 2020 and June 2022 (Phase 2). Hence, the timing of the assessment on the biological mediator (i.e., metabolomics signatures) very likely occurred after the onset of the outcome, especially for those participants who had experienced incidents of depression before 2019/2020. In this case, such temporal misalignment violates the epidemiological framework where the mediator would need to occur after exposure and before the onset of the outcome. After all, metabolomics profiling, especially cross-sectional ones, more often times only reflect transient/acute status of the human metabolome and thus cannot serve as reliable surrogates of one's metabolomics profile in years before. Thus, there is a serious concern regarding temporal misalignment, which violates the rules of mediator from the epidemiological perspective.

2. Overstatements. The manuscript uses definitive language where conditional phrasing would be more appropriate. The entire manuscript should be revised to ensure scientific validity and accuracy of the results and statements reported. For instance, in the introduction (lines 43–45), the authors claim that improvements in treatment have led to increased cancer survival rates. While treatment advancements may contribute, other factors (e.g., early detection, lifestyle changes) also play roles. In another example, in the results section (lines 138–139) states, "air pollution played a major mediator role in the associations of environmental exposures with depression." The term "major" is qualitative and subjective. Given that significant results were only observed for select air pollutants, a more precise description is needed.

3. Lack of justification examining air pollutants as a potential mediator. Given that greenspace and air pollution levels are

highly correlated, it remains unclear and confusing why the study team would treat air pollutants as a mediator/effect modifier rather than independent risk factors, or assess their joint effects.

4. Questionable Source Usage & Statistical Reporting Inconsistencies. Overall, the manuscript lacks transparency regarding cited sources and statistical interpretations. For example, in the introduction (lines 45–46), the authors state, "approximately 17% of cancer survivors experience moderate to severe depression." This figure is from a German study with 1,002 participants, which does not necessarily represent the UK-based study population. The authors should clarify population differences when citing such data. In another example, the mediation analysis (lines 138–153) claims that three of the four variables were significant mediators. However, Table S5 (300m buffer zone) suggests that significance did not change when air pollution variables were included. The discrepancy between hazard ratios in Table S5 and beta coefficients in Table 2 further complicates interpretation. The mediation model should maintain the same exposure variables. The authors should review how the results are reported and ensure reporting accuracy.

5. Lack of Methodological Transparency & Statistical Rigor. Several methodological inconsistencies undermine the study's credibility. Below are a few examples:

- o Lines 418–419 mention four statistical tests for comparing categorical and continuous variables, but it is unclear which test was used for each.
- o The p-value for age (<0.001) in Table 1 appears unlikely and should be explicitly justified.
- o Lines 426–427 describe "stratification" in Model 1 and "adjustment" in Model 2 for socioeconomic status. There are methodological differences between stratification and adjustment, and which was used for each model should be clearly explained.
- o The manuscript states that continuous variables are presented as "means \pm standard error," but Table 1 (line 641) lists them as "means \pm SD." This discrepancy needs resolution.
- In the methods section (lines 407–414), it is unclear and the authors should clarify if air pollutants were included as covariates, mediators, or other. Also, lines 477–478 indicate that pollutants were "initially" included separately, and which suggests that these were later combined in some way in subsequent analyses.

6. Definition & Classification of Environmental Exposures. The definitions and derivations of the "natural environment," "green spaces," and "blue spaces" are unclear (lines 358–370). Specifically:

- o Does "natural environment" include privately owned land?
- o How were spaces that contain both green and blue features classified?
- o Why were 300m and 1000m buffer zones selected? Justification should be provided, along with population distribution data for these zones.
- o Additional buffer distances (e.g., 500m) should be considered for exploration, and as other studies have done.
- A map visualizing study locations, green/blue spaces, and natural environments would enhance clarity.

7. Cancer Type Considerations & Sensitivity Analyses. The study examines depression risk among all cancer survivors without accounting for cancer type. Different cancers may have varying relationships with depression, prognosis, and treatment experiences. For example, lung cancer is strongly linked to air pollution and smoking. On the other hand, skin cancer survivors may interact with green/blue spaces differently due to sun exposure concerns. Cancer stage data should be incorporated, and sensitivity analyses by cancer type should be conducted to assess differential impacts. Additionally, the study has defined long-term cancer survivorship as >5 years since diagnosis (lines 349–351), yet the definition of long-term survivorship can vary by cancer type. Additionally, data on years since diagnosis and cancer type distributions should be included. An additional analysis at a 10-year survival point should be considered, along with citations to justify methodology.

8. Grammatical Considerations. The manuscript will benefit from a thorough proof reading by native English speakers, and should be reviewed for grammatical clarity and consistency. There are numerous grammatical issues throughout the manuscript that are not up to publishing standards. Examples:

- o In the abstract (line 33) association should be plural.
- o 'Respectively' should be used after multiple classifications (e.g., lines 360–361).
- o Lines 469–474 lack a verb, and 'buffer' should be plural when referring to both buffer zones rather than a single zone.

Minor Concerns:

1. The abstract includes just one sentence on the study objective before diving into the results. Please include details on the methods and study population as part of the abstract.
2. In the abstract (line 29), "2.5" in PM_{2.5} should be subscripted.
3. The introduction (lines 74–75) indicates that metabolites are assessed only in relation to green spaces and the natural environment, but the tables and figures clearly include the assessment of blue spaces as well. This should be clarified.
4. Lines 81–92: The results section discusses the results of females when split by sex, yet Table 1 reports on males. Either change the Results section to discuss the male results or change Table 1 to include the results for only females or both sexes to align with results reported in the manuscript.
5. Some figures, such as the flow chart included in Supplemental Figure S1, are blurry and not up to manuscript publishing standards. The HRs and 95% CIs are not legible in Supplemental Figure S3. The last four figures at the end of the manuscript are missing figure titles.

6. For the outcome variable (depression), please provide more information on any diagnosis time restrictions within the study timeframe. While recruitment occurred from 2006–2010 (line 337), please clarify the study's conclusion date and inclusion criteria for depressive diagnoses.
7. Remove references to 'trend tests' (lines 110 and 120), as this is unclear. Specify the exact statistical method used.
8. Line 130: Clarify which exposures the results refer to.
9. Provide details on how antidepressant use among participants was handled.
10. Avoid starting sentences with conjunctions (e.g., line 73).
11. Change "metabolic signature" to "metabolic signatures" (i.e., plural).

Reviewer #2

(Remarks to the Author)

This manuscript presents result of statistical analysis of existing data from the ongoing UK Biobank prospective population study. Major findings include inverse associations between residential greenness measures and the risk of depression in cancer survivors. The authors also utilized the risk data set on almost 250 metabolic biomarkers to develop metabolic indices associated with green spaces and natural environments, and to show that these indices partially mediated the effects of green spaces on depression risks. The authors also assessed mediation effects of four common ambient air pollutants. There are serious problems with the results of this mediation analysis and their interpretation. Overall, the authors obtained a very rich dataset and conducted a lot of statistical tests, but the results are under-interpreted and the text needs revisions for clarity and consistency.

Air pollution is known to be a positive confounder or positive mediator of the inverse associations between green spaces and systemic diseases. Adjusting for air pollution usually moves the observed effect of green spaces towards the null effect. In this study, however, adjusting for air pollution increased the observed protective effects of residential greenness on depression; in mediation analysis the direct effect of residential greens and indirect effect of air pollution had different signs. As residential greenness is negatively correlated with air pollution levels, this implies that air pollution was also negatively associated with the risk of depression (a counter-intuitive apparent protective effect of air pollution). Unfortunately, the authors did not report any data on the association between air pollution and depression. They presented "masking" mediation effects of air pollution without any explanations. Another problem is that various tables and figures used either Hazard Ratios or regression coefficients without any explanations. The signs of many regression coefficients did not match the direction of health effects (e.g., positive regression coefficients for HR values below 1). Also, the values of regression coefficients appeared to be not consistent with HR values presented elsewhere. The authors need to address these major issues as well as minor issues listed below.

Lines 35-37. The statement about lowering pollution level in order to reduce depression risk is contrary to the results of analysis that demonstrated that the indirect effect and direct effect had different directions. Based on these findings, reducing air pollution would increase the risk of depression.

Line 47. Provide references to support the statement about "limited recognition of depression in cancer patients". I believe that depression in cancer survivors is well recognized.

Line 55. Which results were inconsistent? Please provide references to studies that showed detrimental effects of green spaces.

Line 69. While the authors identified a set of biomarkers associated with green spaces, there is no interpretation as to whether the observed effects on individual biomarkers were beneficial.

Lines 71-78. Air pollution is not even mentioned here while elsewhere it presented prominently.

Line 89. The statement about low intake of fruits and vegetables in individuals diagnosed with depression is misleading. According to Table 1, depressed individuals consumed substantially more fruits and vegetables than controls: 49.7% vs. 28.7% reported high consumption and 42.0% vs. 64.9% reported no intake, respectively.

Lines 133-134. This subtitle implies that the observed negative correlation was entirely due to green space reducing air pollution (cause-effect). The evidence of green spaces reducing air pollution (rather than being negatively correlated with it) is not very strong. A recent review estimated that increasing residential greenness by one standard deviation would reduce air pollution by a mere 0.8% (Venter Z, Hassani A, Stange E, Schneider P, Castell N. Reassessing the role of urban green space in air pollution control. Proc Natl Acad Sci. 2024;121).

Line 142. I think that the mediation proportion (masking effect) should be calculated as a percentage of the total effect (the effect of exposure variable not adjusted for the mediator) rather than the direct effect (the effect of exposure variable after adjusting for the mediator).

Line 155. Provide a list of 249 metabolites analyzed in supplemental materials.

Lines 158 -159. Are these effects on large VLDL and other biomarkers beneficial according to the existing knowledge? Provide some interpretation of these findings.

Lines 220-221. The authors should discuss and interpret the direction of the mediation effects.

Lines 245-246 "...excluded patients who experienced depression... within three years of cancer diagnosis". This is not consistent with the Methods section where the authors stated that individuals who had depression within 2 years of cancer were excluded.

Lines 250-252. Again, explain the directions of direct and indirect effects and what it means in terms of the effects of air pollution on depression. Present associations between air pollution and depression elsewhere in the paper.

Lines 253-258. Indeed, previous studies reported negative associations between air pollution and depression. However, this study apparently produced contradictory results which are not reported explicitly by the authors. As the associations between green space and air pollution and between green space and depression are both negative, the observed patterns (adjusting for air pollution increased the magnitude of the observed protective effects of green space on depression; direct and indirect effects have different signs) could only be explained by a negative (inverse) association between air pollution and depression. Perhaps there was an error in statistical analysis; otherwise, the apparent protective effect of air pollution on depression is an intriguing finding that needs to be explained and discussed in the text.

Lines 293-295 and 305-306. It would be helpful to have more information on the analysis of metabolic biomarkers. The metabolic signatures associated with green spaces/natural environment could be influenced by social and behavioral confounding factors that are correlated with residential greenness and affect metabolism. For example, individuals who self-selected to live in greener areas might have more health-oriented behavior and healthier nutrition.

Lines 337-330. It is the other way around: air pollution mediated the effect of greenness.

Line 355. Two years here, three years elsewhere.

Line 388. List these 249 biomarkers in Supplements.

Lines 425-431. This description needs to be edited. It seems that models 2 and 3 were adjusted for age, sex, and ethnicity. Was Model 1 adjusted for these variables? The current description implies that age, sex and ethnicity were only used as stratification factors in Model 1.

Lines 435-436. An incomplete sentence.

Line 468. Explain that PAF means Population Attributable Fraction.

Lines 469-474. This is an incomplete sentence.

Table 2. How are these regression coefficients related to HR values presented in Fig. 3 and elsewhere? Please explain why the direct effect is positive and the indirect effect is negative. According to results presented elsewhere (HR values below 1), the regression coefficients for direct effects should be negative.

Figure 1. Please explain why all confidence intervals collapse at the zero green space or natural environment value. Very few individuals had zero greenness values, so the confidence interval at the left end should be wide.

Figure 3. Explain the values of the direct effect and mediation effect and how HR values can be estimated from these values. I assume that these are regression coefficients which would need to be multiplied by 0.05 for 5% increase in green space and exponentiated. The green space=> depression direct effect value is 5.08. This positive value corresponds with HR>1 (detrimental effect), that is contrary to the text and other figures and tables.

Figure S1. Explain two lines that seem to be similar: ...6,447 diagnosed within 3 years... AND ...25,758 ... diagnosed within three years... What diseases were they diagnosed with? Which one is about depression?

Figure S2. All values for HR (95% CI) are unreadable. Some funny characters are displayed instead of numbers. Model 3 was adjusted for age, sex, ethnicity. However, elsewhere descriptions for model 3 do not include these covariates. Please clarify which models were adjusted for age, sex and ethnicity.

Table S4. Please include depression in this table and provide point biserial correlation coefficients with air pollutants and other continuous variables.

Table S7. I assume that these values are regression coefficients, but it is unclear how they are related to HR values presented elsewhere. All indirect effects of air pollutants are negative while most direct effects are positive suggesting that greater green space was associated with greater risk of depression after adjusting for air pollution. This contradicts protective effects of green space presented elsewhere.

Tables S11 and S12. Adjusting for air pollutants increased the effects of green space (smaller HR values that correspond to greater absolute values of negative regression coefficients). These findings correspond to negative confounding or masking effects of air pollution, which is consistent with the opposite directions of direct and indirect effects in Table 2 (but not with their signs: direct effects should be negative and indirect effects should be positive to produce the patterns in Tables S11-S12). Since air pollution is negatively associated with green spaces, it must be negatively associated with depression to produce the observed patterns. This very unexpected finding needs to be explained.

Reviewer #3

(Remarks to the Author)

Version 1:

Reviewer comments:

Reviewer #1

(Remarks to the Author)

The authors have done substantial work to address most of the previous comments and the manuscript has improved. I applaud the authors' efforts in running additional sensitivity analyses to ensure the findings are robust and consistent. Below are some additional comments based on the study team's response:

1. Air pollution estimates were derived from 2010 land-use regression models, yet the cohort was enrolled 2006–2010. The authors should explicitly acknowledge the temporal gap and potential exposure misclassification due to use of fixed-year models.
2. Regarding responses to my previous comment #1, the author provided clarification on the sample collection time on the samples undergoing the metabolomics profiling. The authors later stated that "In our study, we considered 249 available metabolic biomarkers (excluding glucose-lactate and spectrometer-corrected alanine)". It will be best for the author to describe in the manuscript why in particular glucose-lactate and spectrometer-corrected alanine were excluded from the analysis.
3. Related to the previous comment, the authors stated in their response that "metabolomic profiling was performed in two distinct phases—Phase 1 (June 2019 to 489 April 2020) and Phase 2 (April 2020 to June 2022)". Metabolomics profiling are known to have significant batch effects and the study team should provide details in the method section how the batch effects are corrected and adjusted. Additionally, metabolomics data are known to have a lot of 0 or non-detects across samples (i.e., not all metabolites can be detected in every single bio sample), it remains unclear how the study team deal with these non-detects/0 values, and whether or not imputation is done. In general, QAQC data should be provided for the metabolomics profiling and data processing.
4. In the response letter and in the revised manuscript, the authors stated that they "conducted a sensitivity analysis restricted to individuals who had lived at their current address for more than 10 years prior to baseline". It will be best if the author can specify and describe how many individuals they included in this analysis in the main text.
5. I appreciate the authors conducted additional sensitivity analysis by stratifying the main analysis by cancer type. Indeed some cancer types show stronger trends than the others-- it will be helpful for the study team to briefly describe such interesting observations in the result and discussion sections. Specifically, please move the description of methods for the sensitivity analyses by cancer type completed that are not accompanied by results (lines 230-264) to the methods section. The sensitivity analyses by cancer type yielded some interesting results. Consider including reporting on those that are meaningful briefly in the results section. Please include the N of patients by cancer type in your Supplementary Tables (e.g., in the headings).
6. Line 71-72, PM2.5 should be called "fine particulate matter", not "particulate matter".
7. The use of AFT modeling for mediation is reasonable given Cox limitations, but the interpretation of β vs HR still remains non-intuitive for many readers. It may help if the authors include a brief schematic or supplemental figure explaining AFT-based mediation interpretation for clarity.
8. The validity of baseline metabolomic profiles as long-term mediators in cancer survivors over many years remains biologically debatable. More discussion of how metabolomic stability over time may affect mediation validity are warranted.
9. The authors should revise the abstract to include sufficient details for someone to know the purpose, methods, and main results after reading the abstract. As an example, please mention where the geographic area of focus / where the cancer survivors are from.
10. While the authors stated that air pollutants were assessed as a mediator based on previous literature that green space may reduce air pollution, there are a lot of limitations to this assumption that need to be addressed and considered in the manuscript. This should be acknowledged as a limitation, as a wide variety of factors (wind speed, rainfall, etc.) can impact this.

Reviewer #2

(Remarks to the Author)

The authors very thoroughly revised the manuscript and provided exceptionally detailed responses to reviewers' comments. The revised version presented interesting and convincing findings. My minor comments are below.

Abstract. Provide confidence intervals for the effect estimates, e.g., 12.2% (0.3%; 22.6%) for green space tertile 2 vs. tertile 1. It would also make sense to state that adjusting for air pollution slightly attenuated these estimates, but the effects of the natural environment and green space at the 1000 m buffer remained significant. (These results are in Table S12).

Graphical Abstract. As a graphical abstract should stand on its own, please explain what models 1, 2, and 3 are. Perhaps it makes sense to simplify and leave only one model that is presented in the Abstract.

Line 546. The Bonferroni correction is too punitive. It would be more appropriate to use an FDR (False Discovery Rate) correction.

Figures 2 and 3. There is only one model presented in each figure, but the legends describe three models. Describe only the model presented here.

Fig. 2 and 3. Specify what variable is on the X axis and what variable is denoted by colors. I assume that exposure to air pollution is on the X axis, and exposure to green space is denoted by the color. Fig 2 shows that there are no associations between green/blue/natural spaces and depression risk at low levels of air pollution. Explanations?

Page 15. Briefly describe the findings presented in Supplementary Tables 17-31, focusing on the general patterns.

Lines 318-320. It seems that the effects of greenness were observed at low air pollution levels only. "Stronger effects" would mean that the protective effects of greenness were observed at all levels of air pollution.

Reviewer #3

(Remarks to the Author)

Version 2:

Reviewer comments:

Reviewer #1

(Remarks to the Author)

The authors have addressed all my previous comments and I have no further comment to add.

RESPONSE TO REVIEWERS

Comments from the Reviewers

Reviewer 1:

The study led by Zhao et al. aimed to assess the longitudinal associations of residential green space, blue space and natural environment with depression risk among cancer survivors, in addition to the potential mediating role of four air pollution variables and plasma metabolites. While this is an important topic of significance to public health, and the sample size of the study is large due to the usage of UK Biobank, there are many major concerns regarding the study design and methodology of the presented study, resulting in flaws in the method and invalidity of study findings. Below are detailed comments and suggestions.

Response: We sincerely thank you for the positive and insightful comments that have helped improve our manuscript. We fully agree that the associations of residential green space, blue space, and the natural environment with depression risk among cancer survivors represent an important topic with significant public health implications. In accordance with your suggestions, we have treated air pollution as a key exposure and conducted joint analyses with green space, blue space, and the natural environment in the revised version. Importantly, we clarified the study design and methods of the mediation analysis involving metabolites in the association between environmental exposures and depression among cancer survivors, ensuring alignment with the epidemiological criteria for mediators. The revisions are detailed below, and we hope for your approval.

Major Concerns:

1. Temporal misalignment of the exposure, mediator, and outcome. A fundamental assumption of the presented study is that metabolic signatures

would serve as a biological mediation in the association between green space and incident of depression among cancer survivors. Additionally, they also examine how air pollution mediates these associations. A major concern is the temporal misalignment between the exposure, mediator, and outcome, which resulted in a major flaw of the study design and led to the invalidity of the study findings. Specifically, according to the method section, assessment of the green space was conducted using data in 2005 and 2007, while the recruitment of the cohort started between 2006 to 2010; regarding the outcome assessment, as the authors stated, “incident depression cases were identified from hospital admissions, primary care records, and self-reports...Time-to-event was measured from recruitment until the first occurrence of depression diagnosis, death, loss to follow-up, or study conclusion, whichever occurred first...”. Meanwhile, for the metabolomics mediator, measurement of 251 metabolic biomarkers for EDTA plasma samples collected from a randomly selected subset of approximately 280,000 UKB participants between June 2019 and April 2020 (Phase 1) and April 2020 and June 2022 (Phase 2). Hence, the timing of the assessment on the biological mediator (i.e., metabolomics signatures) very likely occurred after the onset of the outcome, especially for those participants who had experienced incidents of depression before 2019/2020. In this case, such temporal misalignment violates the epidemiological framework where the mediator would need to occur after exposure and before the onset of the outcome. After all, metabolomics profiling, especially cross-sectional ones, more often times only reflect transient/acute status of the human metabolome and thus cannot serve as reliable surrogates of one’s metabolomics profile in years before. Thus, there is a serious concern regarding temporal misalignment, which violates the rules of mediator from the epidemiological perspective.

Response: Thank you for your valuable comments, and we apologize for the confusion caused by the unclear description of the timing of metabolomic

sample collection and analysis. All plasma samples used for metabolomic analysis were collected at baseline between 2006 and 2010, at the time of recruitment. And the measurement of 251 metabolic biomarkers was conducted between 2019 and 2022, primarily on baseline samples (collected in 2006~2010) from a randomly selected subset of approximately 280,000 UK Biobank participants. We have revised the Methods section to clarify and emphasize the timing of metabolomic sample collection and measurement (**Lines 483-497**).

In the revised manuscript, we included cancer survivors who had been diagnosed with cancer prior to study enrollment and excluded individuals with a history of depression at baseline. Therefore, the metabolomic data represent the metabolic profiles of cancer survivors prior to the onset of depression. This ensures that metabolomic measurements were taken after the exposure and before the outcome, thus meeting the temporal requirement for mediators from an epidemiological perspective (**Supplementary Figure S10**).

Indeed, residential green space and blue space were calculated using land use data from the 2005 Generalized Land Use Database (GLUD), while data on the natural environment were linked using the 2007 Land Cover Map from the Centre for Ecology and Hydrology (CEH). Therefore, to further ensure that the assessment of green space, which was based on 2005 and 2007 data, did not compromise the robustness of our findings—given that cohort recruitment took place between 2006 and 2010—we conducted a sensitivity analysis restricted to individuals who had lived at their current address for more than 10 years prior to baseline (**Supplementary Tables S17 and S18**). The associations between green space and natural environment at 1000 m buffer and reduced depression risk remained significant, with hazard ratios (HRs) for Tertile 3 vs. Tertile 1 of 0.831 (95% CI: 0.704–0.981) and 0.794 (95% CI: 0.675–0.934), respectively. In addition, blue space at 1000 m buffer also showed a protective effect against

depression, with an HR for Tertile 3 vs. Tertile 1 of 0.850 (95% CI: 0.723–0.999). Furthermore, according to The European Environment – state and outlook 2020, residential environments in high-income countries such as the United Kingdom are typically characterized by long-term stability ¹. This supports the methodological rationale for using land use data from 2005 and 2007 to assess exposure to green and blue space in our study. Lastly, due to the lack of dynamic exposure assessments, this limitation has been explicitly acknowledged and discussed in the limitations section (**Lines 383-389**).

Method, Page 26, Line 483-497: *“Baseline plasma samples, obtained at the time of participant enrollment in the UK Biobank (2006–2010), were utilized for high-throughput NMR metabolomic profiling. These samples were analyzed by Nightingale Health, which measured 251 metabolic biomarkers in plasma from a randomly selected subset of approximately 280,000 participants from the UK Biobank. In contrast to the timing of sample collection, metabolomic profiling was performed in two distinct phases—Phase 1 (June 2019 to April 2020) and Phase 2 (April 2020 to June 2022)—utilizing state-of-the-art spectrometric techniques in Finland. The process involved comprehensive profiling procedures and rigorous quality control measures, as previously described ^{2,3}. The biomarkers investigated span a range of metabolic pathways, including lipoprotein lipids across 14 subclasses, fatty acid compositions, fatty acids, and low-molecular-weight metabolites. In our study, we considered 249 available metabolic biomarkers (excluding glucose-lactate and spectrometer-corrected alanine). Each biomarker value was transformed using a natural logarithm transformation ($\ln [x+1]$) and subsequently standardized via Z-normalization prior to analysis.”*

Limitation, Page 21, Line 383-389: *“Second, while the spatial distribution of land use characteristics, including green space and blue space, is relatively stable in the UK over short periods, estimating exposure solely based on*

participants' baseline residence may introduce bias. Additionally, although exposure was assessed at fixed residential addresses without accounting for participants' activity patterns or residential mobility, sensitivity analyses restricted to individuals who had lived at their current address for more than 10 years suggested a consistently robust direction of the associations.”

Supplementary Table S17. Associations between environmental exposures at 300 m buffer and the risk of depression, restricted to individuals who had lived at their current address for more than 10 years before baseline.

Exposures	Model 1		Model 2		Model 3	
	HR (95% CI)	P	HR (95% CI)	P	HR (95% CI)	P
Green space						
Tertile 1	Ref	–	Ref	–	Ref	–
Tertile 2	1.107 (0.948-1.294)	0.199	1.057 (0.904-1.235)	0.490	1.023 (0.875-1.197)	0.773
Tertile 3	0.885 (0.751-1.044)	0.146	0.894 (0.758-1.054)	0.183	0.922 (0.781-1.088)	0.337
P trend		0.152		0.189		0.344
Per 5% increment	0.986 (0.972-1.000)	0.054	0.989 (0.975-1.004)	0.147	0.992 (0.978-1.007)	0.324
Blue space						
Tertile 1	Ref	–	Ref	–	Ref	–
Tertile 2	0.958 (0.820-1.118)	0.584	0.974 (0.834-1.138)	0.741	0.969 (0.829-1.132)	0.691
Tertile 3	0.843 (0.717-0.990)	0.037	0.859 (0.731-1.009)	0.064	0.850 (0.723-0.999)	0.049
P trend		0.039		0.067		0.051
Per 5% increment	0.939 (0.845-1.044)	0.246	0.947 (0.853-1.052)	0.309	0.941 (0.841-1.053)	0.292
Natural environment						
Tertile 1	Ref	–	Ref	–	Ref	–
Tertile 2	1.060 (0.906-1.239)	0.468	1.044 (0.893-1.221)	0.590	1.008 (0.862-1.179)	0.917
Tertile 3	0.866 (0.735-1.020)	0.086	0.889 (0.754-1.047)	0.160	0.930 (0.789-1.096)	0.387
P trend		0.087		0.164		0.394
Per 5% increment	0.983 (0.969-0.996)	0.013	0.987 (0.973-1.001)	0.059	0.991 (0.977-1.004)	0.183

Model 1 was adjusted for age, sex and ethnicity; Model 2 was further adjusted for educational level, household income and employment status; Model 3 was fully adjusted for all aforementioned variables as well as body mass index, smoking status, drinking status, physical activity, diet and antidepressant use. Abbreviations: CI, confidence interval; HR, hazard ratio; Ref, reference.

Supplementary Table S18. Associations between environmental exposures at the 1000 m buffer and the risk of depression, restricted to individuals who had lived at their current address for more than 10 years before baseline.

Exposures	Model 1		Model 2		Model 3	
	HR (95% CI)	P	HR (95% CI)	P	HR (95% CI)	P
Green space						
Tertile 1	Ref	–	Ref	–	Ref	–
Tertile 2	0.955 (0.817-1.116)	0.563	0.921 (0.788-1.077)	0.304	0.890 (0.761-1.041)	0.146
Tertile 3	0.795 (0.674-0.936)	0.006	0.804 (0.682-0.947)	0.009	0.831 (0.704-0.981)	0.028
P trend		0.006		0.009		0.028
Per 5% increment	0.974 (0.959-0.988)	<0.001	0.975 (0.961-0.990)	0.001	0.980 (0.965-0.995)	0.010
Blue space						
Tertile 1	Ref	–	Ref	–	Ref	–
Tertile 2	0.890 (0.761-1.041)	0.144	0.901 (0.770-1.054)	0.194	0.846 (0.723-0.990)	0.037
Tertile 3	0.861 (0.735-1.009)	0.065	0.871 (0.743-1.020)	0.087	0.853 (0.728-0.999)	0.049
P trend		0.062		0.084		0.045
Per 5% increment	0.985 (0.911-1.066)	0.713	0.990 (0.916-1.070)	0.800	0.992 (0.919-1.069)	0.826
Natural environment						
Tertile 1	Ref	–	Ref	–	Ref	–
Tertile 2	0.782 (0.669-0.914)	0.002	0.763 (0.652-0.892)	0.001	0.779 (0.665-0.911)	0.002
Tertile 3	0.716 (0.609-0.841)	<0.001	0.734 (0.625-0.863)	<0.001	0.794 (0.675-0.934)	0.005
P trend		<0.001		<0.001		0.005
Per 5% increment	0.971 (0.959-0.985)	<0.001	0.974 (0.961-0.988)	<0.001	0.980 (0.967-0.994)	0.005

Model 1 was adjusted for age, sex and ethnicity; Model 2 was further adjusted for educational level, household income and employment status; Model 3 was fully adjusted for all aforementioned variables as well as body mass index, smoking status, drinking status, physical activity, diet and antidepressant use. Abbreviations: CI, confidence interval; HR, hazard ratio; Ref, reference.

2. Overstatements. The manuscript uses definitive language where conditional phrasing would be more appropriate. The entire manuscript should be revised to ensure scientific validity and accuracy of the results and statements reported. For instance, In the introduction (lines 43–45), the authors claim that improvements in treatment have led to increased cancer survival rates. While treatment advancements may contribute, other factors (e.g., early detection, lifestyle changes) also play roles.

In another example, in the results section (lines 138–139) states, "air pollution played a major mediator role in the associations of environmental exposures with depression." The term "major" is qualitative and subjective. Given that significant results were only observed for select air pollutants, a more precise description is needed.

Response: Thank you for your suggestion. In the revised manuscript, we have avoided definitive language and adopted more conditional phrasing to more accurately and objectively present the background and describe the findings, thereby avoiding overstatements (**Lines 47-50; Lines 178-182; Lines 220-229**).

Introduction, Page 4, Lines 47-50: *“Although the global incidence and prevalence of cancer have steadily increased in recent decades, advances in early diagnosis, treatment, clinical management, and post-diagnostic lifestyle modifications have contributed to a decline in cancer mortality⁴⁻⁶.”*

Results, Page 10-11, Lines 178-182: *“Mediation analysis results indicated that while green space, blue space, and natural environment had significant direct effects on the risk of depression among cancer survivors, no significant mediation effects or proportion mediated were observed for NO₂, NO_x, PM_{2.5}, PM₁₀ or APS in the associations between green space, blue space, natural environment and depression risk (**Supplementary Tables S5 and S6**).”*

Results, Page 13, Lines 220-229: “Mediation analysis indicated that the green space- and natural environment-related metabolic signatures mediated the protective associations between green space, natural environment, and depression among cancer survivors, with mediation effects of 1.163 (95%CI: 0.045-2.472, $P = 0.040$) and 1.225 (95%CI: 0.081-2.561, $P = 0.036$), respectively (**Figure 4C**). However, only the mediation proportion of the green space-related metabolic signature was statistically significant, with a mediation proportion of 5.676% ($P = 0.042$). For the APS-related metabolic signature, its mediation of the association between APS and depression among cancer survivors was not statistically significant, as neither the mediation effect nor the mediation proportion reached significance (**Figure 5C**).”

3. Lack of justification examining air pollutants as a potential mediator. Given that greenspace and air pollution levels are highly correlated, it remains unclear and confusing why the study team would treat air pollutants as a mediator/effect modifier rather than independent risk factors, or assess their joint effects.

Response: Thank you for your suggestion. The consideration of air pollutants as a potential mediator was initially based on previous literature indicating that green space may reduce air pollution levels, and that exposure to air pollutants is associated with an increased risk of depression ⁷. The correlation analysis indicated that greenspace and air pollution levels are correlated. Following your suggestion, we treated air pollutants as independent risk factors and assessed their joint effects.

Specifically, in the revised manuscript, we have incorporated your suggestion by analyzing the association between air pollutants and depression among cancer survivors (**Table 2**). The results indicated that in the crude model, NO₂, NO_x, and PM_{2.5} were associated with an increased risk of depression as components of air pollution. In the full model, NO₂ and NO_x were still associated with an increased risk of depression, with hazard ratios (HRs) for Tertile 3

versus Tertile 1 of 1.140 (95% CI: 1.001–1.299, $P = 0.048$) and 1.143 (95% CI: 1.004–1.303, $P = 0.044$), respectively. Additionally, we assessed their overall impact by constructing an air pollution score through the weighted and normalized summation of NO_2 , NO_x , and $\text{PM}_{2.5}$. The results indicated that the air pollution score was positively associated with depression risk, with HRs for Tertile 3 versus Tertile 1 of 1.152 (95% CI: 1.011–1.312) (**Lines 138-150**). Furthermore, we performed a joint analysis of air pollutants and green space, blue space, and natural environment, and found that lower air pollution levels, combined with higher levels of green space, blue space, and natural environment, were associated with a lower risk of depression among cancer survivors (**Lines 152-177, Figures 2 and 3**).

Results, Page 8-9, Lines 138-150: *“In Models 1 and 2, higher levels of air pollutants (NO_2 , NO_x , and $\text{PM}_{2.5}$) were associated with an increased risk of depression among cancer survivors (Table 2). In the fully adjusted Model 3, cancer survivors exposed to the highest tertile of NO_2 had a 14.0% higher risk of depression compared to those in the lowest tertile (HR = 1.140, 95% CI: 1.001–1.299, $P = 0.048$). Similarly, exposure to the highest tertile of NO_x was associated with a 14.3% higher risk of depression (HR = 1.143, 95% CI: 1.004–1.303, $P = 0.039$). Subsequently, an air pollution score (APS) was calculated based on weighted concentrations of NO_2 , NO_x , and $\text{PM}_{2.5}$. Results from Model 3 indicated tertile 3 of the APS was associated with a 15.2% higher risk of depression (HR = 1.152; 95% CI: 1.011–1.312; $P = 0.033$) compared to the lowest tertile. Cox regression models revealed significant positive dose–response associations between NO_2 , NO_x , and the APS and depression risk, when tertiles were treated as a continuous variable (P for trend = 0.045 for NO_2 ; 0.039 for NO_x ; and 0.029 for APS, respectively).”*

Results, Page 9-10, Lines 152-177: *“Joint analyses of air pollutants and green space, blue space, and natural environment were conducted (Figures 2 and*

3). Results showed that, compared with participants in Tertile 1 of green space at 300 m buffer and Tertile 3 of NO_2 , NO_x , $\text{PM}_{2.5}$, and the APS, those in Tertile 3 of green space and Tertile 1 of air pollutants had a lower risk of depression by 21.1% (HR = 0.789, 95% CI: 0.670–0.929, $P = 0.005$), 15.3% (HR = 0.847, 95% CI: 0.718–1.001, $P = 0.051$), 17.8% (HR = 0.833, 95% CI: 0.708–0.980, $P = 0.027$), and 18.1% (HR = 0.819, 95% CI: 0.696–0.962, $P = 0.015$), respectively. Compared with participants in Tertile 1 of natural environment at 300 m buffer and Tertile 3 of NO_2 , NO_x , $\text{PM}_{2.5}$, and the APS, those in Tertile 3 of natural environment and Tertile 1 of air pollutants had a lower risk of depression by 16.0% (HR = 0.840, 95% CI: 0.714, 0.989, $P = 0.036$), 14.1% (HR = 0.859, 95% CI: 0.731, 1.009, $P = 0.064$), 14.8% (HR = 0.852, 95% CI: 0.727, 0.999, $P = 0.049$), and 15.2% (HR = 0.848, 95% CI: 0.723, 0.993, $P = 0.041$), respectively. Furthermore, compared with participants in Tertile 1 of green space at 1000 m buffer and Tertile 3 of NO_2 , NO_x , $\text{PM}_{2.5}$, and the APS, those in Tertile 3 of green space and Tertile 1 of air pollutants had a significantly lower risk of depression by 17.9% (HR = 0.821, 95% CI: 0.705, 0.956, $P = 0.011$), 16.4% (HR = 0.836, 95% CI: 0.712, 0.982, $P = 0.029$), 16.1% (HR = 0.839, 95% CI: 0.714, 0.986, $P = 0.033$), and 16.4% (HR = 0.836, 95% CI: 0.716, 0.977, $P = 0.024$), respectively. Similarly, compared with participants in Tertile 1 of natural environment at 1000 m buffer and Tertile 3 of air pollutants, those in Tertile 3 of natural environment and Tertile 1 of NO_2 , NO_x , $\text{PM}_{2.5}$, and the APS exhibited a reduced depression risk by 16.3% (HR = 0.837, 95% CI: 0.720–0.973, $P = 0.021$), 17.7% (HR = 0.823, 95% CI: 0.703–0.964, $P = 0.016$), 15.3% (HR = 0.847, 95% CI: 0.724–0.991, $P = 0.038$), and 15.2% (HR = 0.834, 95% CI: 0.716–0.972, $P = 0.020$), respectively. Joint analyses of blue space at 300 m and 1000 m buffers with individual air pollutants were presented in **Figures 2 and 3**, respectively.”

Table 2. Association between air pollutants and depression risk among cancer survivors.

Air pollutants	Model 1		Model 2		Model 3	
	HR (95% CI)	P value	HR (95% CI)	P value	HR (95% CI)	P value
NO₂						
Tertile 1	Ref		Ref		Ref	
Tertile 2	1.132 (0.993-1.290)	0.064	1.045 (0.916-1.192)	0.512	1.026 (0.900-1.171)	0.698
Tertile 3	1.299 (1.142-1.477)	<0.001	1.207 (1.060-1.374)	0.005	1.140 (1.001-1.299)	0.048
P trend		<0.001		0.004		0.045
Per 5% increment	1.014 (1.007-1.021)	<0.001	1.010 (1.003-1.017)	0.003	1.008 (1.001-1.015)	0.028
NO_x						
Tertile 1	Ref		Ref		Ref	
Tertile 2	1.135 (0.994-1.295)	0.061	1.060 (0.928-1.210)	0.393	1.026 (0.898-1.172)	0.704
Tertile 3	1.334 (1.173-1.516)	<0.001	1.218 (1.070-1.386)	0.003	1.143 (1.004-1.303)	0.044
P trend		<0.001		0.002		0.039
Per 5% increment	1.007 (1.004-1.010)	<0.001	1.005 (1.002-1.008)	0.003	1.003 (1.000-1.007)	0.035
PM₁₀						
Tertile 1	Ref		Ref		Ref	
Tertile 2	1.014 (0.892-1.154)	0.827	0.970 (0.853-1.104)	0.649	0.892 (0.783-1.016)	0.084
Tertile 3	1.124 (0.990-1.276)	0.070	1.083 (0.953-1.229)	0.221	0.963 (0.847-1.094)	0.560
P trend		0.069		0.214		0.604
Per 5% increment	1.027 (1.000-1.055)	0.053	1.014 (0.987-1.042)	0.311	0.997 (0.970-1.025)	0.833
PM_{2.5}						
Tertile 1	Ref		Ref		Ref	
Tertile 2	1.067 (0.935-1.217)	0.335	0.998 (0.874-1.139)	0.974	0.958 (0.839-1.094)	0.529
Tertile 3	1.298 (1.143-1.473)	<0.001	1.177 (1.036-1.338)	0.012	1.107 (0.974-1.259)	0.120
P trend		<0.001		0.010		0.099
Per 5% increment	1.134 (1.082-1.189)	<0.001	1.089 (1.038-1.143)	0.001	1.050 (1.000-1.101)	0.048
Air pollution score						
Tertile 1	Ref		Ref		Ref	
Tertile 2	1.123 (0.984-1.282)	0.086	1.050 (0.919-1.199)	0.476	1.017 (0.890-1.162)	0.803
Tertile 3	1.342 (1.181-1.526)	<0.001	1.229 (1.080-1.398)	0.002	1.152 (1.011-1.312)	0.033
P trend		<0.001		0.001		0.029
Per 5% increment	1.073 (1.043-1.105)	<0.001	1.051 (1.020-1.083)	0.001	1.035 (1.004-1.067)	0.028

Model 1 was adjusted for age, sex and ethnicity; Model 2 was further adjusted for educational level, household income and employment status; Model 3 was fully adjusted for all aforementioned variables as well as body mass index, smoking status, drinking status, physical activity, diet and antidepressant use.

Abbreviations: NO₂, nitrogen dioxide; NO_x, nitrogen oxides; PM₁₀, particulate matter (PM) with aerodynamic diameter ≤10 µm; PM_{2.5}, PM with aerodynamic diameter < 2.5 µm.

Figure 2. Joint analysis of air pollutants and green space, blue space, and natural environment at 300 m buffer on depression risk among cancer survivors.

Model 1 was adjusted for age, sex and ethnicity; Model 2 was further adjusted for educational level, household income and employment status; Model 3 was fully adjusted for all aforementioned variables as well as body mass index, smoking status, drinking status, physical activity, diet and antidepressant use. Abbreviations: NO₂, nitrogen dioxide; NO_x, nitrogen oxides; PM_{2.5}, particulate matter with aerodynamic diameter < 2.5 μm; HR, hazard ratio; CI, confidence interval.

Figure 3. Joint analysis of air pollutants and green space, blue space, and natural environment at 1000 m buffer on depression risk among cancer survivors.

Model 1 was adjusted for age, sex and ethnicity; Model 2 was further adjusted for educational level, household income and employment status; Model 3 was fully adjusted for all aforementioned variables as well as body mass index, smoking status, drinking status, physical activity, diet and antidepressant use. Abbreviations: NO₂, nitrogen dioxide; NO_x, nitrogen oxides; PM_{2.5}, particulate matter with aerodynamic diameter < 2.5 μm; HR, hazard ratio; CI, confidence interval.

4. Questionable Source Usage & Statistical Reporting Inconsistencies. Overall, the manuscript lacks transparency regarding cited sources and statistical interpretations. For example, In the introduction (lines 45–46), the authors state, "approximately 17% of cancer survivors experience moderate to severe depression." This figure is from a German study with 1,002 participants, which does not necessarily represent the UK-based study population. The authors should clarify population differences when citing such data.

In another example, the mediation analysis (lines 138–153) claims that three of the four variables were significant mediators. However, Table S5 (300m buffer zone) suggests that significance did not change when air pollution variables were included. The discrepancy between hazard ratios in Table S5 and beta coefficients in Table 2 further complicates interpretation. The mediation model should maintain the same exposure variables. The authors should review how the results are reported and ensure reporting accuracy.

Response: Thank you for your comments. We agree that more representative references should be cited when discussing and evaluating depression among cancer survivors, and that the details of the cited sources should be clearly presented. To strengthen the evidence base of our statements, we have revised the manuscript to include more representative literature—particularly a meta-analysis assessing the prevalence of depression among cancer patients ⁸—and have refined the descriptions of the cited data to enhance specificity and accuracy (*Lines 50-51*).

Additionally, in the revised manuscript, we have provided a more accurate and detailed description of the mediation analysis. The analysis presented in original Table S5 was based on the main model without additional adjustment for air pollutants. In contrast, original Table S11 includes additional adjustment for specific air pollutants. In the updated analysis, although the hazard ratios slightly increased after adjusting for air pollutants (NO₂, NO_x, PM_{2.5} and PM₁₀

individually), the protective associations between green space, natural environment, and depression remained statistically significant (**Supplementary Tables S11 and S12**). Furthermore, we conducted additional sensitivity analyses assessing the associations between environmental exposures and risk of depression with separate adjustments for all air pollutants and the composite air pollution score (**Supplementary Table S13**).

We apologize for the potential confusion caused by the discrepancy between the hazard ratios in original Table S5 and the beta coefficients in original Table 2. This arises from the difference in modeling approaches. Specifically, Table S5 results are based on the Cox proportional hazards model ('coxph ()'), whereas the mediation analysis in Table 2 was conducted using the accelerated failure time (AFT) model fitted via 'survreg ()'.

In the AFT framework:

- The regression coefficient (β) represents the effect of an exposure on the log of survival time (i.e., time to depression onset).
- A positive β indicates a longer time to event (lower risk of depression), whereas a negative β indicates a shorter time to event (higher risk).
- Therefore, the direction of the effect in AFT is inversely related to the interpretation of HRs in Cox models. For instance, a protective factor would yield $HR < 1$ in Cox but a positive β in AFT.

We adopted 'survreg ()' for mediation analysis because currently there is no well-established and robust method to perform mediation within the nested Cox proportional hazards framework. To avoid potential misunderstanding, we have added detailed clarifications in the corresponding table notes and figure captions. Additionally, in the revised manuscript and updated analysis, we did not observe any significant mediating effect of air pollutants in the associations

between green space, blue space, and natural environment and depression (**Supplementary Table S5** and **S6**). The corresponding Methods section, which includes a more detailed description of the nested framework for mediation analysis, and the Results section have been revised accordingly (**Lines 539-542**).

Introduction, Page 4, Lines 50-51: *“The pooled prevalence of depression among cancer patients ranges from 8% to 24% ⁸, increasing with disease recurrence and progression ^{9,10}.”*

Methods, Page 29, Lines 539-542: *“To explore potential interactions, we used the ‘mediation’ package of R based on the accelerated failure time (AFT) model (via ‘survreg ()’) to examine whether air pollutants mediate the associations between green space, blue space, natural environment, and depression risk.”*

Supplementary Table S11. Associations between environmental exposures at the 300 m buffer and risk of depression after adjusting separately for air pollutants.

Exposures	Original *		Adjusted for NO ₂		Adjusted for NO _x		Adjusted for PM ₁₀		Adjusted for PM _{2.5}	
	HR (95% CI)	P	HR (95% CI)	P	HR (95% CI)	P	HR (95% CI)	P	HR (95% CI)	P
Green space										
Tertile 1	Ref	–	Ref	–	Ref	–	Ref	–	Ref	–
Tertile 2	0.971 (0.856-1.101)	0.646	0.995 (0.874-1.132)	0.937	0.988 (0.870-1.123)	0.856	0.968 (0.853-1.098)	0.608	0.988 (0.869-1.124)	0.859
Tertile 3	0.902 (0.791-1.029)	0.126	0.962 (0.825-1.121)	0.621	0.946 (0.819-1.092)	0.448	0.891 (0.777-1.022)	0.099	0.951 (0.817-1.108)	0.523
Trend		0.127		0.631		0.454		0.101		0.533
Per 5% increment	0.992 (0.980-1.004)	0.192	1.000 (0.985-1.014)	0.955	0.997 (0.984-1.011)	0.688	0.991 (0.978-1.003)	0.143	0.998 (0.984-1.013)	0.838
Blue space										
Tertile 1	Ref	–	Ref	–	Ref	–	Ref	–	Ref	–
Tertile 2	0.987 (0.871-1.119)	0.840	1.015 (0.893-1.155)	0.818	1.008 (0.888-1.145)	0.898	0.985 (0.869-1.117)	0.812	1.009 (0.888-1.148)	0.887
Tertile 3	0.867 (0.763-0.986)	0.030	0.893 (0.783-1.020)	0.095	0.886 (0.778-1.010)	0.069	0.865 (0.760-0.984)	0.028	0.891 (0.780-1.018)	0.090
Trend		0.031		0.091		0.068		0.028		0.087
Per 5% increment	0.966 (0.869-1.073)	0.519	0.974 (0.876-1.081)	0.616	0.972 (0.874-1.079)	0.592	0.965 (0.868-1.073)	0.513	0.975 (0.878-1.083)	0.634
Natural environment										
Tertile 1	Ref	–	Ref	–	Ref	–	Ref	–	Ref	–
Tertile 2	0.940 (0.829-1.066)	0.333	0.967 (0.849-1.102)	0.614	0.961 (0.845-1.092)	0.540	0.936 (0.825-1.062)	0.305	0.961 (0.844-1.094)	0.545
Tertile 3	0.903 (0.792-1.028)	0.124	0.966 (0.828-1.126)	0.657	0.952 (0.822-1.102)	0.509	0.890 (0.776-1.020)	0.094	0.957 (0.820-1.118)	0.579
Trend		0.123		0.646		0.501		0.092		0.563
Per 5% increment	0.992 (0.982-1.003)	0.176	0.999 (0.986-1.012)	0.885	0.997 (0.985-1.010)	0.679	0.991 (0.979-1.002)	0.121	0.998 (0.985-1.012)	0.803

* Cox proportional hazard model adjusted by age, sex, ethnicity, educational level, household income, employment status, body mass index, smoking status, drinking status, physical activity, diet and antidepressant use.

Abbreviations: CI, confidence interval; HR, hazard ratio; NO_x, nitrogen oxides; NO₂, nitrogen dioxide; PM_{2.5}, particulate matter with aerodynamic diameter < 2.5 µm; PM₁₀, particulate matter with aerodynamic diameter <10 µm; Ref, reference.

Supplementary Table S12. Associations between environmental exposures at the 1000 m buffer and risk of depression after adjusting separately for air pollutants.

Exposures	Original *		Adjusted for NO ₂		Adjusted for NO _x		Adjusted for PM ₁₀		Adjusted for PM _{2.5}	
	HR (95% CI)	P	HR (95% CI)	P	HR (95% CI)	P	HR (95% CI)	P	HR (95% CI)	P
Green space										
Tertile 1	Ref	–	Ref	–	Ref	–	Ref	–	Ref	–
Tertile 2	0.878 (0.774-0.997)	0.044	0.895 (0.781-1.025)	0.108	0.893 (0.784-1.017)	0.087	0.875 (0.771-0.993)	0.039	0.889 (0.781-1.014)	0.079
Tertile 3	0.842 (0.739-0.960)	0.010	0.877 (0.741-1.038)	0.127	0.872 (0.753-1.010)	0.068	0.830 (0.725-0.950)	0.007	0.870 (0.745-1.015)	0.077
Trend		0.010		0.121		0.067		0.007		0.070
Per 5% increment	0.983 (0.971-0.995)	0.006	0.985 (0.969-1.002)	0.075	0.986 (0.972-1.000)	0.043	0.981 (0.968-0.993)	0.003	0.985 (0.970-1.000)	0.043
Blue space										
Tertile 1	Ref	–	Ref	–	Ref	–	Ref	–	Ref	–
Tertile 2	0.879 (0.774-0.997)	0.045	0.893 (0.786-1.014)	0.081	0.888 (0.782-1.008)	0.066	0.878 (0.773-0.996)	0.043	0.889 (0.783-1.010)	0.070
Tertile 3	0.896 (0.789-1.017)	0.088	0.904 (0.796-1.026)	0.119	0.898 (0.791-1.019)	0.096	0.895 (0.789-1.016)	0.087	0.902 (0.794-1.024)	0.110
Trend		0.085		0.117		0.094		0.084		0.108
Per 5% increment	0.995 (0.938-1.055)	0.868	0.998 (0.941-1.059)	0.958	0.996 (0.939-1.057)	0.893	0.995 (0.938-1.055)	0.868	0.997 (0.940-1.058)	0.933
Natural environment										
Tertile 1	Ref	–	Ref	–	Ref	–	Ref	–	Ref	–
Tertile 2	0.814 (0.717-0.923)	0.001	0.820 (0.715-0.940)	0.005	0.824 (0.723-0.939)	0.004	0.810 (0.713-0.919)	0.001	0.820 (0.719-0.935)	0.003
Tertile 3	0.818 (0.719-0.930)	0.002	0.831 (0.700-0.986)	0.034	0.839 (0.724-0.973)	0.020	0.803 (0.701-0.919)	0.001	0.834 (0.713-0.976)	0.024
Trend		0.002		0.029		0.018		0.001		0.017
Per 5% increment	0.983 (0.973-0.994)	0.002	0.983 (0.968-0.999)	0.034	0.985 (0.973-0.998)	0.022	0.981 (0.970-0.992)	0.001	0.984 (0.970-0.997)	0.020

* Cox proportional hazard model adjusted by age, sex, ethnicity, educational level, household income, employment status, body mass index, smoking status, drinking status, physical activity, diet and antidepressant use.

Abbreviations: CI, confidence interval; HR, hazard ratio; NO_x, nitrogen oxides; NO₂, nitrogen dioxide; PM_{2.5}, particulate matter with aerodynamic diameter < 2.5 µm; PM₁₀, particulate matter with aerodynamic diameter <10 µm; Ref, reference.

Supplementary Table S13. Associations between environmental exposures at the 300 m and 1000 m buffers and risk of depression after adjusting separately for all air pollutants and air pollution score.

Exposures	300 m buffer				1000 m buffer			
	Adjusted for all air pollutants		Adjusted for air pollution score		Adjusted for all air pollutants		Adjusted for air pollution score	
	HR (95% CI)	P	HR (95% CI)	P	HR (95% CI)	P	HR (95% CI)	P
Green space								
Tertile 1	Ref	–	Ref	–	Ref	–	Ref	–
Tertile 2	0.995 (0.874-1.133)	0.944	0.993 (0.873-1.129)	0.915	0.889 (0.772-1.023)	0.099	0.894 (0.783-1.021)	0.099
Tertile 3	0.958 (0.817-1.124)	0.598	0.960 (0.825-1.117)	0.597	0.857 (0.712-1.030)	0.100	0.878 (0.749-1.029)	0.108
Trend		0.613		0.606		0.114		0.108
Per 5% increment	0.999 (0.983-1.014)	0.872	0.999 (0.985-1.014)	0.924	0.980 (0.962-0.999)	0.035	0.986 (0.970-1.001)	0.066
Blue space								
Tertile 1	Ref	–	Ref	–	Ref	–	Ref	–
Tertile 2	1.016 (0.892-1.156)	0.813	1.014 (0.892-1.152)	0.836	0.891 (0.784-1.013)	0.077	0.891 (0.785-1.012)	0.076
Tertile 3	0.893 (0.781-1.021)	0.098	0.893 (0.782-1.019)	0.093	0.902 (0.794-1.025)	0.114	0.902 (0.795-1.024)	0.110
Trend		0.094		0.090		0.114		0.108
Per 5% increment	0.974 (0.877-1.082)	0.621	0.974 (0.877-1.082)	0.624	0.999 (0.941-1.060)	0.976	0.997 (0.940-1.058)	0.933
Natural environment								
Tertile 1	Ref	–	Ref	–	Ref	–	Ref	–
Tertile 2	0.969 (0.850-1.105)	0.640	0.966 (0.848-1.100)	0.601	0.810 (0.703-0.933)	0.003	0.823 (0.720-0.941)	0.004
Tertile 3	0.962 (0.820-1.128)	0.632	0.966 (0.828-1.127)	0.661	0.804 (0.667-0.969)	0.022	0.839 (0.714-0.986)	0.033
Trend		0.620		0.647		0.017		0.026
Per 5% increment	0.998 (0.984-1.012)	0.763	0.999 (0.986-1.012)	0.889	0.978 (0.961-0.995)	0.011	0.984 (0.971-0.999)	0.032

Abbreviations: CI, confidence interval; HR, hazard ratio; Ref, reference.

Supplementary Table S5. Mediation analysis of air pollutants and air pollution score in the association between green space, blue space, natural environment at 300 m buffer and depression risk.

Variable	Indirect effect		Direct effect		Proportion Mediated (%) *	
	β (95 % CI)	P value	β (95 % CI)	P value	β (95 % CI)	P value
Green space						
NO ₂	1.97 (-0.28, 4.50)	0.088	0.12 (-4.04, 4.11)	0.942	94.48 (-672.86, 644.6)	0.270
NO _x	1.36 (-0.36, 3.04)	0.136	0.71 (-2.82, 4.13)	0.682	65.69 (-346.22, 716.91)	0.276
PM _{2.5}	1.60 (-0.44, 3.87)	0.138	0.42 (-3.60, 4.12)	0.816	79.32 (-620.31, 814.51)	0.300
PM ₁₀	-0.45 (-1.70, 0.96)	0.472	2.46 (-0.88, 5.51)	0.158	-22.63 (-359.18, 194.34)	0.590
Air pollution score	1.87 (-0.43, 4.21)	0.120	0.19 (-3.42, 4.02)	0.920	90.79 (-723.76, 807.84)	0.258
Blue space						
NO ₂	1.96 (-0.01, 3.95)	0.052	7.35 (-19.14, 50.47)	0.604	21.07 (-118.27, 219.19)	0.538
NO _x	1.49 (-0.08, 3.16)	0.074	7.90 (-18.28, 55.32)	0.550	15.90 (-121.18, 113.01)	0.538
PM _{2.5}	2.01 (-0.14, 4.61)	0.072	7.01 (-19.7, 51.85)	0.584	22.30 (-233.38, 177.95)	0.550
PM ₁₀	-0.13 (-1.32, 1.05)	0.844	9.65 (-18.36, 49.9)	0.512	-1.39 (-48.56, 45.16)	0.874
Air pollution score	2.01 (0.06, 3.97)	0.048	7.21 (-19.44, 53.7)	0.582	21.79 (-129.44, 184.86)	0.520
Natural environment						
NO ₂	1.74 (-0.32, 3.76)	0.112	0.26 (-3.30, 3.72)	0.840	86.81 (-452.57, 1014.78)	0.230
NO _x	1.30 (-0.37, 2.96)	0.140	0.69 (-2.68, 3.90)	0.722	65.31 (-576.94, 965.58)	0.286
PM _{2.5}	1.46 (-0.68, 3.52)	0.164	0.47 (-3.45, 3.75)	0.794	75.59 (-794.29, 907.54)	0.350
PM ₁₀	-0.51 (-2.02, 0.67)	0.380	2.43 (-0.60, 5.90)	0.118	-26.54 (-388.45, 311.94)	0.548
Air pollution score	1.72 (-0.33, 3.71)	0.116	0.26 (-3.31, 3.74)	0.854	86.81 (-513.22, 832.75)	0.256

The analysis was conducted using Model 3, adjusted for age, sex and ethnicity, educational level, household income, employment status, body mass index, smoking status, drinking status, physical activity, diet and antidepressant use. The regression coefficient (β) represents the effect of exposure on log time to depression onset, where positive values indicate delayed onset (lower risk) and negative values indicate earlier onset (higher risk). * The proportion mediated was quantified as the ratio of the mediation effect to the total effect. Abbreviations: NO₂, nitrogen dioxide; NO_x, nitrogen oxides;

PM₁₀, particulate matter (PM) with aerodynamic diameter ≤10 µm; PM_{2.5}, PM with aerodynamic diameter < 2.5 µm.

Supplementary Table S6. Mediation analysis of air pollutants and air pollution score in the association between green space, blue space, natural environment at 1000 m buffer and depression risk.

Variable	Indirect effect		Direct effect		Proportion Mediated (%)	
	β (95 % CI)	P value	β (95 % CI)	P value	β (95 % CI)	P value
Green space						
NO ₂	0.50 (-2.48, 3.67)	0.754	3.83 (-0.37, 7.77)	0.076	11.62 (-81.59, 113.25)	0.750
NO _x	0.69 (-1.26, 2.59)	0.472	3.64 (0.14, 6.80)	0.038	15.88 (-45.9, 87.97)	0.476
PM _{2.5}	0.39 (-1.87, 2.55)	0.772	3.93 (0.08, 7.32)	0.050	9.10 (-50.85, 101.74)	0.770
PM ₁₀	-0.70 (-2.04, 0.48)	0.234	4.96 (1.78, 8.02)	0.004	-16.52 (-83.23, 13.96)	0.244
Air pollution score	0.65 (-2.23, 3.28)	0.608	3.67 (-0.55, 7.44)	0.080	15.10 (-76.95, 126.8)	0.618
Blue space						
NO ₂	2.10 (0.26, 4.30)	0.018	0.71 (-26.23, 38.24)	0.918	74.68 (-168.22, 232.93)	0.802
NO _x	1.05 (-0.03, 2.30)	0.060	1.97 (-24.64, 41.53)	0.850	34.69 (-128.31, 87.81)	0.834
PM _{2.5}	1.43 (0.01, 2.97)	0.048	1.19 (-24.16, 35.40)	0.906	54.60 (-168.01, 172.44)	0.832
PM ₁₀	-0.04 (-0.51, 0.43)	0.840	2.43 (-22.34, 36.76)	0.804	-1.64 (-15.39, 12.83)	0.990
Air pollution score	1.68 (0.03, 3.29)	0.048	1.18 (-25.39, 37.43)	0.892	58.7 (-143.03, 186.24)	0.780
Natural environment						
NO ₂	0.02 (-2.96, 2.78)	0.998	3.99 (0.41, 8.06)	0.034	0.56 (-95.80, 85.60)	0.998
NO _x	0.45 (-1.36, 2.30)	0.606	3.56 (0.44, 6.53)	0.028	11.29 (-40.95, 72.55)	0.604
PM _{2.5}	0.05 (-2.23, 2.18)	0.986	3.96 (0.67, 7.21)	0.016	1.30 (-77.05, 65.11)	0.986
PM ₁₀	-0.72 (-1.84, 0.34)	0.208	4.66 (2.15, 7.25)	<0.001	-18.15 (-71.08, 8.88)	0.212
Air pollution score	0.27 (-2.14, 2.65)	0.832	3.75 (0.22, 7.06)	0.042	6.67 (-73.06, 91.50)	0.830

The analysis was conducted using Model 3, adjusted for age, sex and ethnicity, educational level, household income, employment status, body mass index, smoking status, drinking status, physical activity, diet and antidepressant use. The regression coefficient (β) represents the effect of exposure on

log time to depression onset, where positive values indicate delayed onset (lower risk) and negative values indicate earlier onset (higher risk). * The proportion mediated was quantified as the ratio of the mediation effect to the total effect. Abbreviations: NO₂, nitrogen dioxide; NO_x, nitrogen oxides; PM₁₀, particulate matter (PM) with aerodynamic diameter ≤10 μm; PM_{2.5}, PM with aerodynamic diameter < 2.5 μm.

5. Lack of Methodological Transparency & Statistical Rigor. Several methodological inconsistencies undermine the study's credibility. Below are a few examples:

- Lines 418–419 mention four statistical tests for comparing categorical and continuous variables, but it is unclear which test was used for each.
- The p-value for age (<0.001) in Table 1 appears unlikely and should be explicitly justified.
- Lines 426–427 describe "stratification" in Model 1 and "adjustment" in Model 2 for socioeconomic status. There are methodological differences between stratification and adjustment, and which was used for each model should be clearly explained.
- The manuscript states that continuous variables are presented as "means \pm standard error," but Table 1 (line 641) lists them as "means \pm SD." This discrepancy needs resolution.
- In the methods section (lines 407–414), it is unclear and the authors should clarify if air pollutants were included as covariates, mediators, or other. Also, lines 477–478 indicate that pollutants were "initially" included separately, and which suggests that these were later combined in some way in subsequent analyses.

Response: Thank you for your suggestion. Following your advice, we have improved the methodological transparency and refined the descriptions of the statistical analyses in the manuscript. Specifically:

(1) We have clearly specified the statistical methods used for each variable in the revised Methods section (*Lines 519-523*).

Methods, Page 28, Lines 519-523: "Baseline characteristics were presented

as median (25th-75th percentile) for continuous variables and frequency (percentage) for categorical variables. Differences in baseline characteristics between participants with and without depression were assessed using the Mann–Whitney U test for continuous variables and the chi-square test for categorical variables.”

(2) We repeated the analysis for age differences. First, we tested the normality of the age variable and found it to be non-normally distributed. Therefore, we used the Mann–Whitney U test to compare age differences between the non-depression and depression groups. We have revised the presentation of age to be displayed as median (25th percentile, 75th percentile) to better reflect its distribution. The results showed a Wilcoxon rank-sum test statistic (W) of 15,338,899 with a P-value = $7.795e-06 < 0.001$. Accordingly, we have revised the Methods section and updated the results in the 'Baseline characteristics of cancer survivors' to reflect these changes (**Table 1**).

(3) We acknowledge the misuse of the term "stratified by." In fact: Model 1 adjusted for age, sex and ethnicity; Model 2 was further adjusted for educational level, household income and employment status; Model 3 was fully adjusted for all aforementioned variables as well as body mass index, smoking status, drinking status, physical activity, dietary factors and antidepressant use (**Lines 531-535**). Subgroup analyses were performed independently and are now clearly outlined in the Statistical Analyses section of the revised manuscript (**Lines 566-572**).

Methods, Page 29, Lines 531-535: *“We constructed three models by sequentially adding covariates: Model 1 was adjusted for age, sex and ethnicity; Model 2 was further adjusted for educational level, household income and employment status; and Model 3 was fully adjusted for all aforementioned variables as well as BMI, smoking status, drinking status, physical activity, dietary factors and antidepressant use.”*

Methods, Page 30-31, Lines 566-572: “Stratified analyses were performed by age (<65 years vs. ≥65 years), sex (female vs. male), ethnicity (White vs. non-White), educational level (university or college vs. other), BMI (normal vs. overweight vs. obese), employment status (currently unemployed vs. currently employed), household income (high: ≥ £52,000 vs. middle: £18,000 - £51,999 vs. low: < £18,000), physical activity (low vs. high), smoking status (current vs. not current), drinking status (current vs. not current), diet (healthy vs. not healthy) and antidepressant use (yes vs. no).”

(4) As a non-normally distributed continuous variable, age is now consistently reported as median (25th, 75th percentile) throughout the manuscript, replacing the previous presentation as mean ± standard deviation (SD).

Table 1, Page 40: “age is presented as median (25th, 75th percentile).”

(5) As per your suggestion, in the revised manuscript, we treated air pollutants as one of the primary exposures and examined their associations with depression among cancer survivors (**Table 2**). In addition, air pollutants were incorporated both as covariates in sensitivity analyses and as potential mediators in mediation analyses. Specifically, in the sensitivity analyses, we first assessed the impact of adjusting for individual air pollutants (PM_{2.5}, PM₁₀, NO₂, and NO_x) on the associations between green/blue space or natural environment and depression risk (**Supplementary Table S11** and **S12**). Furthermore, we added the results of an additional sensitivity analysis in which all four pollutants were simultaneously adjusted for in the main model (**Supplementary Table S13**). Following your suggestion, we also constructed a composite air pollution score (APS) and included it in the main model as part of the sensitivity analysis (**Supplementary Table S13**). Finally, we explicitly detailed the roles of both the individual air pollutants and the APS in our analytical framework, as described in the Methods section (**Lines 455-470**).

Methods, Page 25, Lines 455-470: “Four air pollutants, NO_x , NO_2 , PM_{10} , and $PM_{2.5}$, were calculated by the UK Biobank by using land-use regression (LUR) models developed by the ESCAPE (European Study of Cohorts for Air Pollution Effects) project ^{11,12}. Following the UK Biobank recommendations, air pollutant concentrations from the year 2010 were uniformly applied in our analysis. To capture the joint exposure to various air pollutants, we created a weighted APS by summing up concentrations of 3 ambient air pollutants ($PM_{2.5}$, NO_2 and NO_x), weighted by the multivariable-adjusted (model 3) risk estimates (β coefficients) ¹³. The β coefficients were derived from the final model, with each air pollutant included as a single independent variable, one at a time. The equation was: air pollution score = $(\beta_{PM_{2.5}} \times PM_{2.5} + \beta_{NO_2} \times NO_2 + \beta_{NO_x} \times NO_x) \times (3/\text{sum of the } \beta \text{ coefficients})$. In addition to being primary exposures, air pollutants were also treated as potential mediators to investigate their mediating role in the associations between green space, blue space, and natural environment with depression risk among cancer survivors. Furthermore, they were included as additional covariates beyond those in Model 3 for sensitivity analyses. Further details are provided in the Statistical Analysis section.”

6. Definition & Classification of Environmental Exposures. The definitions and derivations of the "natural environment," "green spaces," and "blue spaces" are unclear (lines 358–370). Specifically: (1) Does "natural environment" include privately owned land?; (2) How were spaces that contain both green and blue features classified?; (3) Why were 300m and 1000m buffer zones selected? Justification should be provided, along with population distribution data for these zones.; Additional buffer distances (e.g., 500m) should be considered for exploration, and as other studies have done; (4) A map visualizing study locations, green/blue spaces, and natural environments would enhance clarity.

Response: Thank you for your comments. First, data on the distribution of the natural environment were obtained from the Land Cover Map (LCM) 2007 (25

m × 25 m resolution). The LCM 2007, developed by the Centre for Ecology and Hydrology (CEH), comprises 23 land cover classes. In our analysis, Classes 1–21 were reclassified as natural environment. It is important to note that Classes 22 and 23, which include buildings and gardens, were not considered part of the natural environment. Therefore, our definition of the natural environment excluded privately owned land such as residential gardens¹⁴. A corresponding clarification has been added to the Methods section to explicitly address this point (**Lines 451-454**).

Second, green and blue space were calculated separately using land use data from the 2005 Generalized Land Use Database (GLUD) for England. In cases where a space contained both green and blue features, each feature was identified and quantified independently¹⁵⁻¹⁷.

Third, the 300 m and 1000 m buffer zones were selected based on the availability of relevant data within the UK Biobank. we agree with your suggestion that a 500-meter buffer could be valuable for further exploration. However, this buffer was not included in the original data modeling and therefore was not available in our dataset. Specifically, based on the UK Biobank’s official documentation, existing evidence linking greenspace density to health outcomes, and relevant public policy on greenspace accessibility, they applied proximity criteria of 300 meters and 1 kilometer to represent nearby and wider-area greenspace exposure relative to participants’ home locations¹⁸⁻²⁰ (Natural England annual report and accounts 2010 to 2011, doi: <https://www.gov.uk/government/publications/natural-england-annual-report-and-accounts-2010-to-2011>). Therefore, we acknowledged this in the limitations section and highlighted that future study using alternative residential green space buffer sizes (e.g., 500-meter buffers) could be valuable for further exploring the potential protective effects of green space on depression among cancer survivors (**Lines 396-399**). Regarding population distribution data, due

to the anonymized nature of the UK Biobank dataset, we were unable to access individual-level address information. This limitation prevented us from further providing population distribution data at 300 m and 1000 m buffer zones. In fact, when calculating residential green space exposure, the percentage of land classified as greenspace—as a proportion of all land use types—was modeled using data from the 2005 Generalized Land Use Database (GLUD) for England, based on the 2001 Census Output Areas. Data were extracted at the finest available administrative geographic unit: the Lower Layer Super Output Areas (LSOAs), which typically contain populations of 1,000 to 3,000 people and 400 to 1,200 households. Each residential address collected at baseline was allocated a circular buffer of 1,000 meters, and the proportion of public greenspace (public greenspace %) within that buffer was used in the analysis²¹. However, the underlying geographic and population datasets used in these UKB-derived variables are not openly accessible to researchers. In the revised manuscript, we have visualized the distribution of cancer survivors across different levels of green space, blue space, and natural environment exposure at 300 m and 1000 m buffer zones (**Supplementary Figure S13**).

Fourth, we agree that a map visualizing study locations, green/blue space, and natural environment would enhance the clarity. As precise geographic location data in the UK Biobank were unavailable, we were unable to generate maps visualizing study locations, green/blue space, and natural environment. However, we have included the distribution of cancer survivors across different levels of green space, blue space, and natural environment exposure at 300 m and 1000 m buffer zones (**Supplementary Figure S13**).

Methods, Page 24-25, Lines 451-454: *“The LCM includes 23 land cover classes; in our study, Classes 1–21 were classified as natural environment, while Classes 22 and 23 (buildings and gardens) were excluded¹⁴. Thus, privately owned lands like residential gardens were not considered part of the*

natural environment.”

Limitation, Page 22, Lines 396-399: *“Fifth, due to data limitations in the UK Biobank, our analysis was restricted to 300 m and 1000 m buffers. Future studies employing alternative buffer sizes for residential green space and natural environment (e.g., 500 m) may provide further insights.”*

Supplementary Figure S13. Distribution of cancer survivors across levels of green space, blue space, and natural environment exposure at 300 m (A, B and C) and 1000 m (D, E and F) buffers.

7. Cancer Type Considerations & Sensitivity Analyses. The study examines depression risk among all cancer survivors without accounting for cancer type. Different cancers may have varying relationships with depression, prognosis, and treatment experiences. For example, lung cancer is strongly linked to air pollution and smoking. On the other hand, skin cancer survivors may interact with green/blue spaces differently due to sun exposure concerns. Cancer stage data should be incorporated, and sensitivity analyses by cancer type should be conducted to assess differential impacts. Additionally, the study has defined long-term cancer survivorship as >5 years since diagnosis (lines 349–351), yet the definition of long-term survivorship can vary by cancer type. Additionally, data on years since diagnosis and cancer type distributions should be included. An additional analysis at a 10-year survival point should be considered, along with citations to justify methodology.

Response: Thank you for your valuable suggestions. In the revised manuscript, we have included analyses by cancer type and conducted a series of sensitivity analyses to further investigate the association between residential green space and depression among cancer survivors. Specifically, we conducted separate analyses for the four major cancer types, including breast cancer (**Supplementary Tables S19-20**), prostate cancer (**Supplementary Tables S27-28**), colorectal cancer (**Supplementary Tables S29-30**), and melanoma skin cancer (**Supplementary Tables S21-22**). In addition, based on your suggestion, we additionally conducted sensitivity analyses for lung cancer (**Supplementary Tables S25-26**) and non-melanoma skin cancer (**Supplementary Tables S23-24**). For the analyses of melanoma skin cancer and non-melanoma skin cancer, we further adjusted for daily sun exposure duration (hours/day, continuous variable; UK Biobank Field ID: 1050/1060).

Among all cancer survivors, long-term survivors of breast cancer constituted the largest group (n = 7,365). According to the GLOBOCAN 2022 report, breast

cancer accounted for 2.3 million new cases and 670,000 deaths globally. It is the most commonly diagnosed cancer among women, representing 25% of all cancer cases and 15.5% of cancer-related deaths, highlighting its significant global disease burden ²². Importantly, we observed particularly strong protective effects of green space and natural environment at 1000 m buffer on depression risk in this subgroup (**Supplementary tables S19-20**). In the fully adjusted model, individuals in Tertile 3 of green space had a 21.9% lower risk of depression compared to those in Tertile 1 (HR = 0.781, 95% CI: 0.631, 0.966, P for trend = 0.022). Similarly, compared to those in Tertile 1, individuals in Tertile 2 and Tertile 3 of natural environment exposure had 25.7% and 23.9% lower risks of depression, respectively (HR = 0.743, 95% CI: 0.603–0.914; HR = 0.761, 95% CI: 0.618–0.936; P for trend = 0.009).

For lung cancer with only 225 cases surviving more than five years in our study population), our findings showed that although green space and natural environment exposure did not demonstrate inverse association with depression risk, individuals in Tertile 3 of blue space exposure at 1000 m buffer had a 79.5% lower risk of depression compared to those in Tertile 1 (HR = 0.205, 95% CI: 0.045–0.930; p for trend = 0.047) (**Supplementary Tables S25-26**). Based on your comment that lung cancer is strongly associated with air pollution and smoking, we further adjusted for the air pollutants on top of Model 3 (which already included smoking status as a covariate), defining this as Model 4. Results from Model 4 showed that blue space at 1000 m buffer in Tertile 3 remained significantly associated with a reduced risk of depression (HR = 0.098, 95% CI: 0.015-0.652) among cancer survivors. Notably, although green space and natural environment were not statistically significantly associated with lung cancer, this may be partly due to the limited statistical power from the small sample size of 225 cases, and the findings should therefore be interpreted with caution.

In line with your suggestion, we included an additional analysis among melanoma skin cancer survivors (n = 1,721 in our study population) in the revised manuscript (**Supplementary tables S21-22**). Given that individuals with skin cancer may interact with green and blue space differently due to concerns about sun exposure, we further adjusted for daily sun exposure duration (hours/day, continuous variable; UK Biobank Field ID: 1050/1060). It is worth noting that, in line with previous studies, individuals with non-melanoma skin cancer were not included in the main analyses due to their excellent prognosis and extremely low mortality²³. However, in response to your comment, we identified an additional 14,896 non-melanoma skin cancer cases from our dataset and conducted a supplementary analysis. The results showed that residential natural environment at 1000 m buffer were still associated with a lower risk of depression in this group, with HRs_{Tertile 3 vs. Tertile 1} of 0.780 (95% CI: 0.610–0.998), respectively (**Supplementary tables S23-24**). In the above analyses of melanoma skin cancer and non-melanoma skin cancer, we further adjusted for daily sun exposure duration (hours/day, continuous variable; UK Biobank Field ID: 1050/1060) based on Model 3.

To enhance the robustness of our findings, we have conducted an additional sensitivity analysis using a 10-year survival threshold (**Supplementary tables S15-16**). The analysis showed that among cancer survivors with a survival time of at least 10 years, green space and natural environment at 1000 m buffer continued to show significant inverse associations. Compared to Tertile 1, participants in Tertile 3 of green space had a 14.2% lower risk of depression, with a HR of 0.858 (95% CI: 0.750-0.980), P for trend = 0.019. Similarly, those in Tertile 3 of natural environment had an 17.7% lower risk of depression, with an HR of 0.823 (95% CI: 0.721–0.939), P for trend = 0.004. These findings were consistent with our main results. Furthermore, to improve clarity, we have added visualizations illustrating the distribution of cancer types and survival times among cancer survivors, as well as the distribution of cancer types and

diagnosis-to-enrollment intervals among cancer survivors, in the revised manuscript (**Supplementary Figures S11-12**). The 5-year survival threshold is a widely used clinical and epidemiological standard in cancer research, as it marks a critical turning point following treatment for most cancer types, beyond which the risk of recurrence or cancer-related mortality substantially decreases. Therefore, based on prior literature ^{6,23}, we defined individuals who survived more than five years after a cancer diagnosis as cancer survivors in our study, and we have also added the corresponding references in the revised manuscript (**Lines 432-438**). Additionally, we acknowledge that cancer stage is a critical prognostic factor due to its direct association with disease severity and survival. However, detailed analysis was not feasible given that cancer stage information is not systematically curated or released in the UK Biobank. This limitation has been explicitly addressed in the revised manuscript (**Lines 399-401**).

Methods, Page 23-24, Line 432-438: *“The cohort initially included 93,677 participants with cancer. We excluded individuals diagnosed with cancer after baseline (n = 65,195), those with less than five years of cancer survivorship (n = 714) ^{6,23}, participants with missing data on green space, blue space, and natural environment exposures (n = 3,393), those with a diagnosis of depression prior to baseline (n = 303), and those diagnosed with other psychiatric disorders (n = 2,565). After these exclusions, a total of 21,507 cancer survivors were included in the final analysis (**Supplementary Figures S11 and S12**).”*

Limitation, Page 22, Lines 399-401: *“Finally, despite extensive confounder adjustment, residual confounding may persist. For instance, cancer stage—a key prognostic factor—could not be assessed due to its unavailability in the UK Biobank.”*

Supplementary Figure S11. Distribution of cancer types and survival time among cancer survivors.

Supplementary Figure S12. Distribution of cancer types and diagnosis-to-enrollment interval among cancer survivors.

8. Grammatical Considerations. The manuscript will benefit from a thorough proof reading by native English speakers, and should be reviewed for grammatical clarity and consistency. There are numerous grammatical issues throughout the manuscript that are not up to publishing standards. Examples:

- In the abstract (line 33) association should be plural.
- ‘Respectively’ should be used after multiple classifications (e.g., lines 360–361).
- Lines 469–474 lack a verb, and ‘buffer’ should be plural when referring to both buffer zones rather than a single zone.

Response: We apologize for any grammatical confusion. We have carefully reviewed the revised manuscript for grammatical issues, and it has been further proofread and edited by native English speakers to ensure clarity and language accuracy (**Lines 53-56; Lines 441-443; Lines 566-572**).

Introduction, Page 4, Lines 53-56: *“The substantial burden of depression in cancer patients underscores the critical need to identify modifiable factors associated with mental health, particularly protective factors, which could serve as potential targets for future interventions and reveal novel therapeutic strategies.”*

Methods, Page 24, Lines 441-443: *“The percentages of residential green space and blue space, classified as ‘greenspace’ and ‘water’, respectively, were calculated as proportions of the total area of all land-use types.”*

Methods, Page 30-31, Lines 566-572: *“Stratified analyses were performed by age (<65 years vs. ≥65 years), sex (female vs. male), ethnicity (White vs. non-White), educational level (university or college vs. other), BMI (normal vs. overweight vs. obese), employment status (currently unemployed vs. currently employed), household income (high: ≥ £52,000 vs. middle: £18,000 - £51,999*

vs. low: < £18,000), physical activity (low vs. high), smoking status (current vs. not current), drinking status (current vs. not current), diet (healthy vs. not healthy) and antidepressant use (yes vs. no).”

Minor Concerns:

1. The abstract includes just one sentence on the study objective before diving into the results. Please include details on the methods and study population as part of the abstract.

Response: Thank you for your suggestions. We have added details on the methods and study population as part of the abstract (**Lines 28-31**).

Abstract, Page 2, Lines 28-31: *“A total of 21,507 cancer survivors were included. Associations were assessed using Cox proportional hazards models, and relevant metabolites were identified using elastic net and Lasso regression.”*

2. In the abstract (line 29), “2.5” in PM2.5 should be subscripted.

Response: Thank you for your comments. We have subscripted '2.5' in 'PM_{2.5}' as suggested and have thoroughly reviewed the entire manuscript.

3. The introduction (lines 74–75) indicates that metabolites are assessed only in relation to green spaces and the natural environment, but the tables and figures clearly include the assessment of blue spaces as well. This should be clarified.

Response: Thank you for your comments. In fact, we analyzed the associations of green space, blue space, and the natural environment with depression among cancer survivors. The results of main analysis showed statistically significant inverse associations of green space and natural environment (at 1000 m buffer) with depression, whereas no such association was observed for blue space. Therefore, in the metabolite analyses, we focused

only on green space and the natural environment. In line with your suggestion, we have carefully reviewed the tables and figures related to the metabolite analyses and confirmed that no analyses involving blue space-related metabolites were conducted.

4. Lines 81–92: The results section discusses the results of females when split by sex, yet Table 1 reports on males. Either change the Results section to discuss the male results or change Table 1 to include the results for only females or both sexes to align with results reported in the manuscript.

Response: Thank you for your detailed comments. We have revised Table 1 to include the results for both sexes and have made modifications to the results section accordingly.

5. Some figures, such as the flow chart included in Supplemental Figure S1, are blurry and not up to manuscript publishing standards. The HRs and 95% CIs are not legible in Supplemental Figure S3. The last four figures at the end of the manuscript are missing figure titles.

Response: Thank you for your suggestion. We have improved the clarity of the original Supplementary Figure S1 (**Supplementary Figure S10**) and revised the original Supplementary Figure S3 accordingly (**Supplementary Figures S7 and S8**) to enhance the presentation of hazard ratios and 95% confidence intervals, and added appropriate titles to the supplementary figures.

6. For the outcome variable (depression), please provide more information on any diagnosis time restrictions within the study timeframe. While recruitment occurred from 2006–2010 (line 337), please clarify the study's conclusion date and inclusion criteria for depressive diagnoses.

Response: Thank you for your suggestion. For the outcome variable (depression), we excluded all individuals with a diagnosis of depression prior to

cancer diagnosis or before recruitment (**Lines 432-438**). The end dates for follow-up regarding depression diagnoses were: October 31, 2022 for England; August 31, 2022 for Scotland; and March 31, 2022 for Wales. These details have been clarified in the Methods section of the revised manuscript (**Lines 477-481**).

Methods, Pages 23-24, Lines 432-438: *“The cohort initially included 93,677 participants with cancer. We excluded individuals diagnosed with cancer after baseline (n = 65,195), those with less than five years of cancer survivorship (n = 714)^{6,23}, participants with missing data on green space, blue space, and natural environment exposures (n = 3,393), those with a diagnosis of depression prior to baseline (n = 303), and those diagnosed with other psychiatric disorders (n = 2,565). After these exclusions, a total of 21,507 cancer survivors were included in the final analysis (**Supplementary Figures S11 and S12**).”*

Methods, Page 26, Lines 477-481: *“Time-to-event was measured from the date of recruitment to the first occurrence of a depression diagnosis, death, loss to follow-up, or the end of follow-up (October 31, 2022 for England; August 31, 2022 for Scotland; and March 31, 2022 for Wales), whichever came first.”*

7. Remove references to ‘trend tests’ (lines 110 and 120), as this is unclear. Specify the exact statistical method used.

Response: Thank you for your suggestion. Thank you for your suggestion. ‘Trend tests’ is a statistical term in our study referring to the use of Cox regression to examine whether there is a trend between the exposure variable (‘green space’) and the outcome variable (‘depression risk’), such as whether depression risk decreases with increasing green space exposure. To specify the exact statistical method, we have added the details of the statistical method used for the ‘trend tests’ in the revised manuscript (**Lines 123-126**).

Results, Page 8, Lines 123-126: “Cox regression models revealed a significant inverse association between green space and natural environment and depression risk when tertiles were treated as a continuous variable (P for trend = 0.010 for green space; P for trend = 0.002 for natural environment).”

8. Line 130: Clarify which exposures the results refer to.

Response: Thank you for your suggestion. We have clarified the exposures that the results refer to in the revised manuscript.

9. Provide details on how antidepressant use among participants was handled.

Response: Thank you for your comments. We extracted data on antidepressant use from the UK Biobank using Data-Fields 20003 and conducted two analyses. First, we included antidepressant use as an additional covariate in Model 3 for the main analysis. All primary analyses in the revised manuscript have been adjusted for antidepressant use. Second, we conducted a sensitivity analysis by excluding participants using antidepressants (**Supplementary Tables S15 and S16**). The results showed that, after excluding cancer survivors who were using antidepressants at baseline, green space at 1000 m buffer in the highest tertile (Tertile 3) remained associated with a 14.7% lower risk of depression compared to the lowest tertile (Tertile 1); similarly, natural environment exposure in Tertile 3 was associated with a 20.3% lower risk.

Supplementary Table S15. Associations between environmental exposures at the 300 m buffer and the risk of depression after excluding participants using antidepressants at baseline, those diagnosed with depression within 1 year or 3 years post-cancer, and those who died within 10 years after cancer diagnosis.

Exposures	Excluding participants using antidepressants at baseline		Excluding depression cases within 1 year post-cancer		Excluding depression cases within 3 years post-cancer		Excluding death cases within 10 years post-cancer	
	HR (95% CI)	P	HR (95% CI)	P	HR (95% CI)	P	HR (95% CI)	P
Green space								
Tertile 1	Ref	–	Ref	–	Ref	–	Ref	–
Tertile 2	1.015 (0.895-1.151)	0.814	0.989 (0.868-1.125)	0.862	0.977 (0.851-1.121)	0.739	0.980 (0.862-1.115)	0.763
Tertile 3	0.896 (0.786-1.022)	0.102	0.929 (0.811-1.063)	0.283	0.931 (0.807-1.074)	0.326	0.916 (0.801-1.048)	0.202
Trend		0.103		0.283		0.326		0.203
Per 5% increment	0.991 (0.979-1.003)	0.123	0.994 (0.982-1.006)	0.338	0.994 (0.981-1.007)	0.335	0.993 (0.981-1.006)	0.290
Blue space								
Tertile 1	Ref	–	Ref	–	Ref	–	Ref	–
Tertile 2	0.980 (0.865-1.111)	0.754	0.990 (0.870-1.126)	0.877	0.969 (0.845-1.111)	0.653	0.985 (0.867-1.119)	0.819
Tertile 3	0.881 (0.775-1.002)	0.053	0.877 (0.769-1.000)	0.051	0.872 (0.758-1.002)	0.054	0.852 (0.747-0.972)	0.017
Trend		0.054		0.052		0.054		0.018
Per 5% increment	0.946 (0.857-1.044)	0.267	0.981 (0.884-1.087)	0.710	0.939 (0.830-1.061)	0.313	0.972 (0.874-1.080)	0.596
Natural environment								
Tertile 1	Ref	–	Ref	–	Ref	–	Ref	–
Tertile 2	0.980 (0.864-1.111)	0.755	0.942 (0.828-1.072)	0.364	0.922 (0.804-1.058)	0.250	0.931 (0.819-1.059)	0.280
Tertile 3	0.893 (0.784-1.017)	0.088	0.920 (0.805-1.052)	0.222	0.914 (0.793-1.052)	0.210	0.907 (0.794-1.036)	0.151
Trend		0.088		0.221		0.208		0.150
Per 5% increment	0.991 (0.980-1.002)	0.114	0.994 (0.983-1.005)	0.290	0.993 (0.981-1.005)	0.247	0.993 (0.982-1.004)	0.240

All analyses were performed using Model 3 as the base model. Abbreviations: CI, confidence interval; HR, hazard ratio; Ref, reference.

Supplementary Table S16. Associations between environmental exposures at the 1000 m buffer and the risk of depression after excluding participants using antidepressants at baseline, those diagnosed with depression within 1 year or 3 years post-cancer, and those who died within 10 years after cancer diagnosis.

Exposures	Excluding participants using antidepressants at baseline		Excluding depression cases within 1 year post-cancer		Excluding depression cases within 3 years post-cancer		Excluding death cases within 10 years post-cancer	
	HR (95% CI)	P	HR (95% CI)	P	HR (95% CI)	P	HR (95% CI)	P
Green space								
Tertile 1	Ref	–	Ref	–	Ref	–	Ref	–
Tertile 2	0.910 (0.802-1.032)	0.142	0.895 (0.786-1.019)	0.094	0.882 (0.768-1.013)	0.076	0.886 (0.779-1.009)	0.068
Tertile 3	0.853 (0.749-0.972)	0.017	0.864 (0.756-0.988)	0.032	0.856 (0.742-0.987)	0.032	0.858 (0.750-0.980)	0.024
Trend		0.017		0.033		0.032		0.025
Per 5% increment	0.983 (0.972-0.995)	0.006	0.985 (0.973-0.997)	0.017	0.984 (0.971-0.997)	0.014	0.985 (0.973-0.998)	0.019
Blue space								
Tertile 1	Ref	–	Ref	–	Ref	–	Ref	–
Tertile 2	0.932 (0.822-1.058)	0.277	0.901 (0.792-1.026)	0.114	0.913 (0.795-1.048)	0.195	0.880 (0.773-1.001)	0.051
Tertile 3	0.902 (0.795-1.024)	0.110	0.903 (0.793-1.028)	0.124	0.915 (0.797-1.051)	0.211	0.887 (0.779-1.009)	0.069
Trend		0.109		0.121		0.209		0.066
Per 5% increment	0.976 (0.920-1.035)	0.415	1.001 (0.943-1.062)	0.975	0.983 (0.920-1.051)	0.621	0.998 (0.940-1.059)	0.940
Natural environment								
Tertile 1	Ref	–	Ref	–	Ref	–	Ref	–
Tertile 2	0.813 (0.717-0.923)	0.001	0.826 (0.726-0.941)	0.004	0.844 (0.735-0.969)	0.016	0.815 (0.716-0.927)	0.002
Tertile 3	0.797 (0.700-0.907)	0.001	0.833 (0.729-0.951)	0.007	0.831 (0.721-0.957)	0.010	0.823 (0.721-0.939)	0.004
Trend		0.001		0.007		0.010		0.004
Per 5% increment	0.982 (0.971-0.993)	0.001	0.985 (0.974-0.996)	0.007	0.984 (0.972-0.995)	0.006	0.985 (0.974-0.996)	0.006

All analyses were performed using Model 3 as the base model. Abbreviations: CI, confidence interval; HR, hazard ratio; Ref, reference.

10. Avoid starting sentences with conjunctions (e.g., line 73).

Response: Thank you for your suggestion. In accordance with your advice, we have revised the manuscript to avoid starting sentences with conjunctions.

11. Change "metabolic signature" to "metabolic signatures" (i.e., plural).

Response: Thank you for your suggestion. In accordance with your advice, we have changed 'metabolic signature' to 'metabolic signatures' and further reviewed the manuscript for grammatical accuracy.

Reviewer 2:

1. This manuscript presents result of statistical analysis of existing data from the ongoing UK Biobank prospective population study. Major findings include inverse associations between residential greenness measures and the risk of depression in cancer survivors. The authors also utilized the risk data set on almost 250 metabolic biomarkers to develop metabolic indices associated with green spaces and natural environments, and to show that these indices partially mediated the effects of green spaces on depression risks. The authors also assessed mediation effects of four common ambient air pollutants. There are serious problems with the results of this mediation analysis and their interpretation. Overall, the authors obtained a very rich dataset and conducted a lot of statistical tests, but the results are under-interpreted and the text needs revisions for clarity and consistency.

Response: Thank you for your positive evaluation of our manuscript and your valuable suggestions. In response to your comments, we reanalyzed the data and focused on clarifying and revising the methods and interpretation of the mediation analysis. We have also made textual revisions to improve clarity and consistency throughout the manuscript. We hope these revisions address your concerns. Below, we provide point-by-point responses to each of your suggestions.

2. Air pollution is known to be a positive confounder or positive mediator of the inverse associations between green spaces and systemic diseases. Adjusting for air pollution usually moves the observed effect of green spaces towards the null effect. In this study, however, adjusting for air pollution increased the observed protective effects of residential greenness on depression; in mediation analysis the direct effect of residential greens and indirect effect of air pollution had different signs. As residential greenness is negatively correlated with air pollution levels, this implies that air pollution was also

negatively associated with the risk of depression (a counter-intuitive apparent protective effect of air pollution). Unfortunately, the authors did not report any data on the association between air pollution and depression. They presented “masking” mediation effects of air pollution without any explanations. Another problem is that various tables and figures used either Hazard Ratios or regression coefficients without any explanations. The signs of many regression coefficients did not match the direction of health effects (e.g., positive regression coefficients for HR values below 1). Also, the values of regression coefficients appeared to be not consistent with HR values presented elsewhere.

Response: Thank you for your comments. We agree with your comment that green space are traditionally considered to reduce the risk of systemic diseases partly by lowering air pollution. In response to Reviewer 1’s request, we redefined the study population to include a total of 21,507 baseline cancer survivors with at least five years of survivorship and no prior diagnosis of depression. In the revised manuscript, we treated air pollution as an exposure on par with green space, blue space, and natural environment, and conducted joint and mediation analyses accordingly (**Lines 136-177**).

First, we incorporated an analysis of the association between air pollutants and depression among cancer survivors (**Page 41, Table 2**). The results showed that in the crude model, NO₂, NO_x, and PM_{2.5} were each associated with an increased risk of depression. In the fully adjusted model, NO₂ and NO_x remained significantly associated with higher depression risk, with hazard ratios (HRs) for Tertile 3 versus Tertile 1 of 1.140 (95% CI: 1.001, 1.299, P = 0.048) and 1.143 (95% CI: 1.004, 1.303, P = 0.044), respectively. Additionally, we constructed an air pollution score (APS) by weighted and normalized summation of NO₂, NO_x, and PM_{2.5} to assess their joint effect. The APS was positively associated with depression risk, with HR for Tertile 3 versus Tertile 1 of 1.152 (95% CI: 1.011, 1.312). In addition, the joint analysis of air pollutants

and green space, blue space, and natural environment in relation to depression risk among cancer survivors is presented in **Figures 2 and 3**.

Second, correlation analyses showed significant negative associations between green space, blue space, natural environment, and air pollutants, consistent with previous studies (**Supplementary Table S2**). Considering that green space, blue space, and natural environment were associated with reduced depression risk among cancer survivors, while air pollutants were linked to increased risk, we conducted mediation analyses to explore whether air pollution acts as a mediator in these relationships. The updated results indicated that neither individual air pollutants nor the air pollution score appeared to mediate the associations. Moreover, since the directions of most indirect and direct effects' β coefficients were consistent, following your suggestion, we applied the classic proportion mediated calculation method, defined as the ratio of the mediation effect to the total effect. The corresponding proportion mediated was not statistically significant. Additionally, following your suggestion, we performed further adjustments for specific air pollutants. In the updated analysis, although hazard ratios slightly increased after adjusting individually for NO₂, NO_x, PM_{2.5}, and PM₁₀, the protective associations between green space, natural environment, and depression remained statistically significant (**Supplementary Tables S11 and S12**). We also conducted sensitivity analyses adjusting separately for all air pollutants and for the composite air pollution score, yielding results similar to single pollutant adjustments (**Supplementary Table S13**). These findings suggest that, despite the negative correlation between air pollutants and green space, blue space, and natural environment, and the opposite directions of their associations with depression risk, their effects on depression risk among cancer survivors appear to be relatively independent. This may, at least in part, be explained by the review provided by you, which estimated that increasing residential greenness by one standard deviation reduces air pollution by only about 0.8%²⁴.

Third, we apologize for not clearly explaining the directionality of the regression coefficients and HRs. In the mediation analysis, mediation was conducted using the accelerated failure time (AFT) model fitted via 'survreg ()', which differs from the Cox proportional hazards model ('coxph ()'). We adopted 'survreg ()' for mediation analysis because, currently, there is no well-established and robust method to perform mediation within the nested Cox proportional hazards framework. In the AFT framework: (1) The regression coefficient (β) represents the effect of an exposure on the log of survival time (i.e., time to depression onset). (2) A positive β indicates a longer time to event (lower risk of depression), whereas a negative β indicates a shorter time to event (higher risk). (3) Therefore, the direction of the effect in AFT is inversely related to the interpretation of HRs in Cox models. For instance, a protective factor would yield $HR < 1$ in Cox but a positive β in AFT.

Without considering the underlying assumptions of the AFT model and the Cox proportional hazards model, $e^{(-\beta)}$ can be approximately equal to HR. However, directly using $e^{(-\beta)}$ to convert the β from the AFT model to HR in the Cox model is not appropriate. Although both models are used for survival analysis, they approach the problem from different perspectives—risk and time-to-event, respectively. The conversion between the two requires more complex considerations and may involve different assumptions regarding survival distributions. Therefore, in the manuscript, we did not choose to convert the β into the more easily interpretable HR. To avoid potential misunderstanding, we have added detailed clarifications in the corresponding table notes and figure captions to aid readers' understanding (**Supplementary Tables S5 and S6**).

Results, Pages 8-10, Lines 136-177: *“Associations between air pollutants and depression risk among cancer survivors In Models 1 and 2, higher levels of air pollutants (NO_2 , NO_x , and $PM_{2.5}$) were associated with an increased risk of depression among cancer survivors (Table 2). In the fully adjusted Model*

3, cancer survivors exposed to the highest tertile of NO₂ had a 14.0% higher risk of depression compared to those in the lowest tertile (HR = 1.140, 95% CI: 1.001–1.299, P = 0.048). Similarly, exposure to the highest tertile of NO_x was associated with a 14.3% higher risk of depression (HR = 1.143, 95% CI: 1.004–1.303, P = 0.039). Subsequently, an air pollution score (APS) was calculated based on weighted concentrations of NO₂, NO_x, and PM_{2.5}. Results from Model 3 indicated tertile 3 of the APS was associated with a 15.2% higher risk of depression (HR = 1.152; 95% CI: 1.011–1.312; P = 0.033) compared to the lowest tertile. Cox regression models revealed significant positive dose–response associations between NO₂, NO_x, and the APS and depression risk, when tertiles were treated as a continuous variable (P for trend = 0.045 for NO₂; 0.039 for NO_x; and 0.029 for APS, respectively).

Joint and mediation analysis Joint analyses of air pollutants and green space, blue space, and natural environment were conducted (**Figures 2 and 3**). Results showed that, compared with participants in Tertile 1 of green space at 300 m buffer and Tertile 3 of NO₂, NO_x, PM_{2.5}, and the APS, those in Tertile 3 of green space and Tertile 1 of air pollutants had a lower risk of depression by 21.1% (HR = 0.789, 95% CI: 0.670–0.929, P = 0.005), 15.3% (HR = 0.847, 95% CI: 0.718–1.001, P = 0.051), 17.8% (HR = 0.833, 95% CI: 0.708–0.980, P = 0.027), and 18.1% (HR = 0.819, 95% CI: 0.696–0.962, P = 0.015), respectively. Compared with participants in Tertile 1 of natural environment at 300 m buffer and Tertile 3 of NO₂, NO_x, PM_{2.5}, and the APS, those in Tertile 3 of natural environment and Tertile 1 of air pollutants had a lower risk of depression by 16.0% (HR = 0.840, 95% CI: 0.714, 0.989, P = 0.036), 14.1% (HR = 0.859, 95% CI: 0.731, 1.009, P = 0.064), 14.8% (HR = 0.852, 95% CI: 0.727, 0.999, P = 0.049), and 15.2% (HR = 0.848, 95% CI: 0.723, 0.993, P = 0.041), respectively. Furthermore, compared with participants in Tertile 1 of green space at 1000 m buffer and Tertile 3 of NO₂, NO_x, PM_{2.5}, and the APS, those in Tertile 3 of green space and Tertile 1 of air pollutants had a significantly lower risk of depression

by 17.9% (HR = 0.821, 95% CI: 0.705, 0.956, P = 0.011), 16.4% (HR = 0.836, 95% CI: 0.712, 0.982, P = 0.029), 16.1% (HR = 0.839, 95% CI: 0.714, 0.986, P = 0.033), and 16.4% (HR = 0.836, 95% CI: 0.716, 0.977, P = 0.024), respectively. Similarly, compared with participants in Tertile 1 of natural environment at 1000 m buffer and Tertile 3 of air pollutants, those in Tertile 3 of natural environment and Tertile 1 of NO₂, NO_x, PM_{2.5}, and the APS exhibited a reduced depression risk by 16.3% (HR = 0.837, 95% CI: 0.720–0.973, P = 0.021), 17.7% (HR = 0.823, 95% CI: 0.703–0.964, P = 0.016), 15.3% (HR = 0.847, 95% CI: 0.724–0.991, P = 0.038), and 15.2% (HR = 0.834, 95% CI: 0.716–0.972, P = 0.020), respectively. Joint analyses of blue space at 300 m and 1000 m buffers with individual air pollutants were presented in **Figures 2 and 3**, respectively.”

Supplementary Tables S5 and S6: “The regression coefficient (β) represents the effect of exposure on log time to depression onset, where positive values indicate delayed onset (lower risk) and negative values indicate earlier onset (higher risk). * The proportion mediated was quantified as the ratio of the mediation effect to the total effect.”

Table 2. Association between air pollutants and depression risk among cancer survivors.

Air pollutants	Model 1		Model 2		Model 3	
	HR (95% CI)	P value	HR (95% CI)	P value	HR (95% CI)	P value
NO₂						
Tertile 1	Ref		Ref		Ref	
Tertile 2	1.132 (0.993-1.290)	0.064	1.045 (0.916-1.192)	0.512	1.026 (0.900-1.171)	0.698
Tertile 3	1.299 (1.142-1.477)	<0.001	1.207 (1.060-1.374)	0.005	1.140 (1.001-1.299)	0.048
P trend		<0.001		0.004		0.045
Per 5% increment	1.014 (1.007-1.021)	<0.001	1.010 (1.003-1.017)	0.003	1.008 (1.001-1.015)	0.028
NO_x						
Tertile 1	Ref		Ref		Ref	
Tertile 2	1.135 (0.994-1.295)	0.061	1.060 (0.928-1.210)	0.393	1.026 (0.898-1.172)	0.704
Tertile 3	1.334 (1.173-1.516)	<0.001	1.218 (1.070-1.386)	0.003	1.143 (1.004-1.303)	0.044
P trend		<0.001		0.002		0.039
Per 5% increment	1.007 (1.004-1.010)	<0.001	1.005 (1.002-1.008)	0.003	1.003 (1.000-1.007)	0.035
PM₁₀						
Tertile 1	Ref		Ref		Ref	
Tertile 2	1.014 (0.892-1.154)	0.827	0.970 (0.853-1.104)	0.649	0.892 (0.783-1.016)	0.084
Tertile 3	1.124 (0.990-1.276)	0.070	1.083 (0.953-1.229)	0.221	0.963 (0.847-1.094)	0.560
P trend		0.069		0.214		0.604
Per 5% increment	1.027 (1.000-1.055)	0.053	1.014 (0.987-1.042)	0.311	0.997 (0.970-1.025)	0.833
PM_{2.5}						
Tertile 1	Ref		Ref		Ref	
Tertile 2	1.067 (0.935-1.217)	0.335	0.998 (0.874-1.139)	0.974	0.958 (0.839-1.094)	0.529
Tertile 3	1.298 (1.143-1.473)	<0.001	1.177 (1.036-1.338)	0.012	1.107 (0.974-1.259)	0.120
P trend		<0.001		0.010		0.099
Per 5% increment	1.134 (1.082-1.189)	<0.001	1.089 (1.038-1.143)	0.001	1.050 (1.000-1.101)	0.048
Air pollution score						
Tertile 1	Ref		Ref		Ref	
Tertile 2	1.123 (0.984-1.282)	0.086	1.050 (0.919-1.199)	0.476	1.017 (0.890-1.162)	0.803
Tertile 3	1.342 (1.181-1.526)	<0.001	1.229 (1.080-1.398)	0.002	1.152 (1.011-1.312)	0.033
P trend		<0.001		0.001		0.029
Per 5% increment	1.073 (1.043-1.105)	<0.001	1.051 (1.020-1.083)	0.001	1.035 (1.004-1.067)	0.028

Model 1 was adjusted for age, sex and ethnicity; Model 2 was further adjusted for educational level, household income and employment status; Model 3 was fully adjusted for all aforementioned variables as well as body mass index, smoking status, drinking status, physical activity, diet and antidepressant use.

Abbreviations: NO₂, nitrogen dioxide; NO_x, nitrogen oxides; PM₁₀, particulate matter (PM) with aerodynamic diameter ≤10 µm; PM_{2.5}, PM with aerodynamic diameter < 2.5 µm.

Figure 2. Joint analysis of air pollutants and green space, blue space, and natural environment at 300 m buffer on depression risk among cancer survivors.

Model 1 was adjusted for age, sex and ethnicity; Model 2 was further adjusted for educational level, household income and employment status; Model 3 was fully adjusted for all aforementioned variables as well as body mass index, smoking status, drinking status, physical activity, diet and antidepressant use. Abbreviations: NO₂, nitrogen dioxide; NO_x, nitrogen oxides; PM_{2.5}, particulate matter with aerodynamic diameter < 2.5 μm; HR, hazard ratio; CI, confidence interval.

Figure 3. Joint analysis of air pollutants and green space, blue space, and natural environment at 1000 m buffer on depression risk among cancer survivors.

Model 1 was adjusted for age, sex and ethnicity; Model 2 was further adjusted for educational level, household income and employment status; Model 3 was fully adjusted for all aforementioned variables as well as body mass index, smoking status, drinking status, physical activity, diet and antidepressant use. Abbreviations: NO₂, nitrogen dioxide; NO_x, nitrogen oxides; PM_{2.5}, particulate matter with aerodynamic diameter < 2.5 μm; HR, hazard ratio; CI, confidence interval.

Supplementary Table S2. Correlation matrix of environmental exposures.

Exposures	Green space (300m buffer)	Green space (1000m buffer)	Blue space (300m buffer)	Blue space (1000m buffer)	Natural environment (300m buffer)	Natural environment (1000m buffer)	NO ₂	NO _x	PM ₁₀	PM _{2.5}
Green space (300m buffer)	1									
Green space (1000m buffer)	0.853 ***	1								
Blue space (300m buffer)	0.049 ***	0.020 **	1							
Blue space (1000m buffer)	0.022 **	0.003	0.722 ***	1						
Natural environment (300m buffer)	0.884 ***	0.761 ***	0.137 ***	0.086 ***	1					
Natural environment (1000m buffer)	0.824 ***	0.968 ***	0.103 ***	0.112 ***	0.786 ***	1				
NO ₂	-0.643 ***	-0.735 ***	-0.082 ***	-0.070 ***	-0.628 ***	-0.756 ***	1			
NO _x	-0.532 ***	-0.559 ***	-0.069 ***	-0.038 ***	-0.551 ***	-0.582 ***	0.922 ***	1		
PM ₁₀	-0.419 ***	-0.390 ***	-0.052 ***	-0.013	-0.446 ***	-0.406 ***	0.520 ***	0.527 ***	1	
PM _{2.5}	-0.638 ***	-0.643 ***	-0.098 ***	-0.055 ***	-0.643 ***	-0.674 ***	0.864 ***	0.847 ***	0.548 ***	1
Depression	-0.016 *	-0.023 ***	-0.009	-0.007	-0.019 **	-0.030 ***	0.028 ***	0.030 ***	0.013	0.036 ***

*, P < 0.05; **, P < 0.01; ***, P < 0.001.

Abbreviations: NO₂, nitrogen dioxide; NO_x, nitrogen oxides; PM₁₀, particulate matter (PM) with aerodynamic diameter ≤10 µm; PM_{2.5}, PM with aerodynamic diameter < 2.5 µm.

Supplementary Table S11. Associations between environmental exposures at 300 m buffer and risk of depression after adjusting separately for air pollutants.

Exposures	Original *		Adjusted for NO ₂		Adjusted for NO _x		Adjusted for PM ₁₀		Adjusted for PM _{2.5}	
	HR (95% CI)	P	HR (95% CI)	P	HR (95% CI)	P	HR (95% CI)	P	HR (95% CI)	P
Green space										
Tertile 1	Ref	–	Ref	–	Ref	–	Ref	–	Ref	–
Tertile 2	0.971 (0.856-1.101)	0.646	0.995 (0.874-1.132)	0.937	0.988 (0.870-1.123)	0.856	0.968 (0.853-1.098)	0.608	0.988 (0.869-1.124)	0.859

Tertile 3	0.902 (0.791-1.029)	0.126	0.962 (0.825-1.121)	0.621	0.946 (0.819-1.092)	0.448	0.891 (0.777-1.022)	0.099	0.951 (0.817-1.108)	0.523
Trend		0.127		0.631		0.454		0.101		0.533
Per 5% increment	0.992 (0.980-1.004)	0.192	1.000 (0.985-1.014)	0.955	0.997 (0.984-1.011)	0.688	0.991 (0.978-1.003)	0.143	0.998 (0.984-1.013)	0.838
Blue space										
Tertile 1	Ref	–	Ref	–	Ref	–	Ref	–	Ref	–
Tertile 2	0.987 (0.871-1.119)	0.840	1.015 (0.893-1.155)	0.818	1.008 (0.888-1.145)	0.898	0.985 (0.869-1.117)	0.812	1.009 (0.888-1.148)	0.887
Tertile 3	0.867 (0.763-0.986)	0.030	0.893 (0.783-1.020)	0.095	0.886 (0.778-1.010)	0.069	0.865 (0.760-0.984)	0.028	0.891 (0.780-1.018)	0.090
Trend		0.031		0.091		0.068		0.028		0.087
Per 5% increment	0.966 (0.869-1.073)	0.519	0.974 (0.876-1.081)	0.616	0.972 (0.874-1.079)	0.592	0.965 (0.868-1.073)	0.513	0.975 (0.878-1.083)	0.634
Natural environment										
Tertile 1	Ref	–	Ref	–	Ref	–	Ref	–	Ref	–
Tertile 2	0.940 (0.829-1.066)	0.333	0.967 (0.849-1.102)	0.614	0.961 (0.845-1.092)	0.540	0.936 (0.825-1.062)	0.305	0.961 (0.844-1.094)	0.545
Tertile 3	0.903 (0.792-1.028)	0.124	0.966 (0.828-1.126)	0.657	0.952 (0.822-1.102)	0.509	0.890 (0.776-1.020)	0.094	0.957 (0.820-1.118)	0.579
Trend		0.123		0.646		0.501		0.092		0.563
Per 5% increment	0.992 (0.982-1.003)	0.176	0.999 (0.986-1.012)	0.885	0.997 (0.985-1.010)	0.679	0.991 (0.979-1.002)	0.121	0.998 (0.985-1.012)	0.803

* Cox proportional hazard model adjusted by age, sex, ethnicity, educational level, household income, employment status, body mass index, smoking status, drinking status, physical activity, diet and antidepressant use.

Abbreviations: CI, confidence interval; HR, hazard ratio; NO_x, nitrogen oxides; NO₂, nitrogen dioxide; PM_{2.5}, particulate matter with aerodynamic diameter < 2.5 µm; PM₁₀, particulate matter with aerodynamic diameter <10 µm; Ref, reference.

Supplementary Table S12. Associations between environmental exposures at the 1000 m buffer and risk of depression after adjusting separately for air pollutants.

Exposures	Original *		Adjusted for NO ₂		Adjusted for NO _x		Adjusted for PM ₁₀		Adjusted for PM _{2.5}	
	HR (95% CI)	P	HR (95% CI)	P	HR (95% CI)	P	HR (95% CI)	P	HR (95% CI)	P
Green space										
Tertile 1	Ref	–	Ref	–	Ref	–	Ref	–	Ref	–
Tertile 2	0.878 (0.774-0.997)	0.044	0.895 (0.781-1.025)	0.108	0.893 (0.784-1.017)	0.087	0.875 (0.771-0.993)	0.039	0.889 (0.781-1.014)	0.079
Tertile 3	0.842 (0.739-0.960)	0.010	0.877 (0.741-1.038)	0.127	0.872 (0.753-1.010)	0.068	0.830 (0.725-0.950)	0.007	0.870 (0.745-1.015)	0.077
Trend		0.010		0.121		0.067		0.007		0.070
Per 5% increment	0.983 (0.971-0.995)	0.006	0.985 (0.969-1.002)	0.075	0.986 (0.972-1.000)	0.043	0.981 (0.968-0.993)	0.003	0.985 (0.970-1.000)	0.043
Blue space										
Tertile 1	Ref	–	Ref	–	Ref	–	Ref	–	Ref	–
Tertile 2	0.879 (0.774-0.997)	0.045	0.893 (0.786-1.014)	0.081	0.888 (0.782-1.008)	0.066	0.878 (0.773-0.996)	0.043	0.889 (0.783-1.010)	0.070
Tertile 3	0.896 (0.789-1.017)	0.088	0.904 (0.796-1.026)	0.119	0.898 (0.791-1.019)	0.096	0.895 (0.789-1.016)	0.087	0.902 (0.794-1.024)	0.110
Trend		0.085		0.117		0.094		0.084		0.108
Per 5% increment	0.995 (0.938-1.055)	0.868	0.998 (0.941-1.059)	0.958	0.996 (0.939-1.057)	0.893	0.995 (0.938-1.055)	0.868	0.997 (0.940-1.058)	0.933
Natural environment										
Tertile 1	Ref	–	Ref	–	Ref	–	Ref	–	Ref	–
Tertile 2	0.814 (0.717-0.923)	0.001	0.820 (0.715-0.940)	0.005	0.824 (0.723-0.939)	0.004	0.810 (0.713-0.919)	0.001	0.820 (0.719-0.935)	0.003
Tertile 3	0.818 (0.719-0.930)	0.002	0.831 (0.700-0.986)	0.034	0.839 (0.724-0.973)	0.020	0.803 (0.701-0.919)	0.001	0.834 (0.713-0.976)	0.024
Trend		0.002		0.029		0.018		0.001		0.017
Per 5% increment	0.983 (0.973-0.994)	0.002	0.983 (0.968-0.999)	0.034	0.985 (0.973-0.998)	0.022	0.981 (0.970-0.992)	0.001	0.984 (0.970-0.997)	0.020

* Cox proportional hazard model adjusted by age, sex, ethnicity, educational level, household income, employment status, body mass index, smoking status, drinking status, physical activity, diet and antidepressant use.

Abbreviations: CI, confidence interval; HR, hazard ratio; NO_x, nitrogen oxides; NO₂, nitrogen dioxide; PM_{2.5}, particulate matter with aerodynamic diameter < 2.5 µm; PM₁₀, particulate matter with aerodynamic diameter <10 µm; Ref, reference.

Supplementary Table S13. Associations between environmental exposures at the 300 m and 1000 m buffers and risk of depression after adjusting separately for all air pollutants and air pollution score.

Exposures	300 m buffer				1000 m buffer			
	Adjusted for all air pollutants		Adjusted for air pollution score		Adjusted for all air pollutants		Adjusted for air pollution score	
	HR (95% CI)	P	HR (95% CI)	P	HR (95% CI)	P	HR (95% CI)	P
Green space								
Tertile 1	Ref	–	Ref	–	Ref	–	Ref	–
Tertile 2	0.995 (0.874-1.133)	0.944	0.993 (0.873-1.129)	0.915	0.889 (0.772-1.023)	0.099	0.894 (0.783-1.021)	0.099
Tertile 3	0.958 (0.817-1.124)	0.598	0.960 (0.825-1.117)	0.597	0.857 (0.712-1.030)	0.100	0.878 (0.749-1.029)	0.108
Trend		0.613		0.606		0.114		0.108
Per 5% increment	0.999 (0.983-1.014)	0.872	0.999 (0.985-1.014)	0.924	0.980 (0.962-0.999)	0.035	0.986 (0.970-1.001)	0.066
Blue space								
Tertile 1	Ref	–	Ref	–	Ref	–	Ref	–
Tertile 2	1.016 (0.892-1.156)	0.813	1.014 (0.892-1.152)	0.836	0.891 (0.784-1.013)	0.077	0.891 (0.785-1.012)	0.076
Tertile 3	0.893 (0.781-1.021)	0.098	0.893 (0.782-1.019)	0.093	0.902 (0.794-1.025)	0.114	0.902 (0.795-1.024)	0.110
Trend		0.094		0.090		0.114		0.108
Per 5% increment	0.974 (0.877-1.082)	0.621	0.974 (0.877-1.082)	0.624	0.999 (0.941-1.060)	0.976	0.997 (0.940-1.058)	0.933
Natural environment								
Tertile 1	Ref	–	Ref	–	Ref	–	Ref	–
Tertile 2	0.969 (0.850-1.105)	0.640	0.966 (0.848-1.100)	0.601	0.810 (0.703-0.933)	0.003	0.823 (0.720-0.941)	0.004
Tertile 3	0.962 (0.820-1.128)	0.632	0.966 (0.828-1.127)	0.661	0.804 (0.667-0.969)	0.022	0.839 (0.714-0.986)	0.033
Trend		0.620		0.647		0.017		0.026
Per 5% increment	0.998 (0.984-1.012)	0.763	0.999 (0.986-1.012)	0.889	0.978 (0.961-0.995)	0.011	0.984 (0.971-0.999)	0.032

The analysis was conducted based on Model 3, adjusted for age, sex, ethnicity, educational level, household income, employment status, body mass index, smoking status, drinking status, physical activity, diet and antidepressant use, with additionally adjusted for all air pollutants or APS.

Abbreviations: APS, air pollution score; CI, confidence interval; HR, hazard ratio; Ref, reference.

3. The authors need to address these major issues as well as minor issues listed below.

Lines 35-37. The statement about lowering pollution level in order to reduce depression risk is contrary to the results of analysis that demonstrated that the indirect effect and direct effect had different directions. Based on these findings, reducing air pollution would increase the risk of depression.

Response: Thank you for your comments. We apologize for any confusion regarding the directions of direct and indirect effects. Mediation analysis was conducted using an accelerated failure time (AFT) model implemented via 'survreg ()', which models log-transformed survival time and differs fundamentally from the Cox proportional hazards model ('coxph ()') in its parametric assumptions and interpretation of coefficients. In the AFT framework: (1) The regression coefficient (β) reflects the effect of an exposure on the logarithm of survival time (i.e., time to depression onset); (2) A positive β indicates a longer time to event (i.e., a lower risk of depression), while a negative β indicates a shorter time to event (i.e., a higher risk); and (3) Therefore, the direction of effects in the AFT model is inversely interpreted compared to hazard ratios (HRs) in Cox models—for example, a protective factor would yield $HR < 1$ in Cox but a positive β in AFT. For green space, a positive β indicates a prolonged time to depression onset, reflecting a reduced risk, whereas a negative β suggests a shortened time to onset, reflecting an increased risk of depression. We adopted 'survreg ()' for mediation analysis because there is currently no well-established and robust method for performing mediation within a nested Cox proportional hazards framework. To prevent potential misunderstanding, we have added detailed clarifications in the corresponding table notes and figure captions to aid reader interpretation.

Importantly, we fully agree with you that examining the association between air pollutants and depression is of great relevance. In response, we assessed the

impact of individual air pollutants on depression risk and further developed an air pollution score to evaluate its association with depression. Our results showed that higher levels of NO₂ (HR for tertile 3 vs. tertile 1 = 1.299, 95% CI: 1.142, 1.477), NO_x, (HR = 1.334, 95% CI: 1.173, 1.516), and PM_{2.5} were significantly associated with increased risk of depression. In the fully adjusted Model 3, both NO₂ (HR = 1.140, 95% CI: 1.001, 1.299) and NO_x, (HR = 1.143, 95% CI: 1.004, 1.303) remained significantly associated with elevated depression risk. Furthermore, the air pollution score was constructed by weighting and combining NO₂, NO_x, and PM_{2.5}. Further analysis indicated that the air pollution score was significantly associated with an increased depression risk in the final Model 3, with a HR_{per 5% increment} of 1.035 (95% CI: 1.004–1.067) and HR_{Tertile 3 vs. Tertile 1} of 1.152 (95% CI: 1.011-1.312) (**Table 2**).

After reanalyzing the data with updated inclusion criteria, we observed protective associations of green space (at 1000 m buffer), blue space (at 300 m buffer), and natural environment (at 1000 m buffer) with depression (**Figure 1**), as well as adverse effects of air pollutants on depression risk (**Table 2**). Additionally, green space, blue space, and natural environment were inversely associated with air pollution exposure (**Supplementary Table S2**). However, mediation analysis did not support a mediating role of air pollution in the associations between environmental exposures and depression (**Supplementary Tables S5 and S6**). One possible reason is that residential greenness may have only a limited effect on reducing air pollution levels, as noted in the review provided by you (Zander S. Venter et al.), which estimated that a one standard deviation increase in residential greenness reduces air pollution by only approximately 0.8%²⁴. In addition, we conducted joint analyses of air pollution with green space and natural environment exposures (**Figures 2 and 3**).

A

B

Figure 1. Associations between exposure to green space, blue space, and natural environment at 300 m (A) and 1000 m (B) buffers and incident depression among cancer survivors.

Model 1 was adjusted for age, sex and ethnicity; Model 2 was further adjusted for educational level, household income and employment status; Model 3 was fully adjusted for all aforementioned variables as well as body mass index, smoking status, drinking status, physical activity, diet and antidepressant use. Abbreviations: HR, hazard ratio; CI, confidence interval.

4. Line 47. Provide references to support the statement about “limited recognition of depression in cancer patients”. I believe that depression in cancer survivors is well recognized.

Response: Apologies for the inappropriate phrasing. We have made revisions to better highlight the focus of the study (**Lines 53-56**).

***Introduction, Page 4, Lines 53-56:** “The substantial burden of depression in cancer patients underscores the critical need to identify modifiable factors associated with mental health, particularly protective factors, which could serve as potential targets for future interventions and reveal novel therapeutic strategies.”*

5. Line 55. Which results were inconsistent? Please provide references to studies that showed detrimental effects of green spaces.

Response: We apologize for the lack of clarity in our previous description. While the protective effect of green space on depression may vary in magnitude and statistical significance across different study populations, the majority of prior studies have demonstrated a protective association ²⁵. Interestingly, a study by Dr. Andrew Tomita reported that green space was associated with an increased risk of depression in Model 2 (which adjusted for gender, marital status, education, age, race or ethnicity, employment status, and land typology as a measure of development), with an odds ratio (OR) of 1.01 (95% CI: 1.01–1.02) ²⁶. However, this association was not statistically significant in Model 1, which did not adjust for any covariates (OR = 1.00, 95% CI: 1.00–1.01), suggesting that potential confounding may influence the observed relationship. Considering the relatively consistent protective effect of green space on depression, we have rewritten the sentence and added appropriate references to make the statement more precise, while emphasizing that cancer survivors, a psychologically vulnerable group, have not been specifically investigated

(Lines 61-64).

Introduction, Page 4, Lines 61-64: “Although a meta-analysis supports improving green space exposure to prevent depression in the general population ²⁵, most studies are cross-sectional and have not specifically investigated cancer survivors, a psychologically vulnerable group.”

6. Line 69. While the authors identified a set of biomarkers associated with green spaces, there is no interpretation as to whether the observed effects on individual biomarkers were beneficial.

Response: Thank you for your valuable suggestion. In addition to evaluating the overall associations between metabolites linked to green space and natural environment with depression, and following your suggestion, we also conducted individual analyses for 45 metabolites associated with green space and 58 metabolites associated with natural environment in relation to depression risk (**Supplementary Figures S4 and S5**). The lack of statistical significance for individual metabolites after false discovery rate correction was primarily due to the small independent effects of many metabolites on depression, coupled with limited sample size, which reduced the statistical power to detect significant associations. Nevertheless, we discussed several metabolites—specifically leucine, glutamine, and tyrosine—which were consistently positively associated with green space and the natural environment. Notably, although their associations with depression did not reach statistical significance, the point estimates suggested a potential protective association. Thus, we conducted a systematic review of the literature on these three metabolites and found prior evidence suggesting that they may have protective effects against mental disorders, including depression; we have incorporated a discussion of these metabolites in the Discussion section (**Lines 343-378**).

Discussion, Pages 19-20, Lines 343-378: “Our study identified a comprehensive panel of metabolites associated with green space, the natural environment, and the APS, which were subsequently integrated into metabolic signatures. The results showed that metabolic signatures related to the natural environment and green space were associated with reduced depression risk, while the APS-related signature was linked to an increased risk among cancer survivors. Metabolic signatures related to the natural environment and green space act as potential mediators in their associations with depression among cancer survivors. Specifically, we found that three amino acids—leucine, glutamine, and tyrosine—were consistently and positively associated with green space and the natural environment. Previous studies have shown that leucine supplementation can alleviate social avoidance and depressive-like behaviors in mice ²⁷. The glutamate–glutamine cycle and glutamatergic neurotransmission play critical roles in normal brain function, highlighting the importance of maintaining glutamine homeostasis in the brain. A deficiency of neuronal glutamine in the medial prefrontal cortex of rodents has been linked to depressive behaviors and mild cognitive impairment ²⁸. Furthermore, mendelian randomization evidence supports a potential causal relationship between downregulation of glutamine and major depressive disorder, anhedonia, fatigue, and low mood ²⁹. For tyrosine, previous studies have reported significantly reduced levels in the blood and urine of patients with depression ³⁰, with concentrations showing a significant negative correlation with Hamilton Depression Rating Scale (HAM-D) scores ³¹. Additionally, 3-Hydroxybutyrate, a ketone body that was consistently positively associated with green space and the natural environment, has been shown to significantly reduce anxiety-like behaviors in rats when administered through long-term dietary supplementation ³². Interestingly, for large Very Low-Density Lipoprotein (VLDL), we found that cholesteryl esters in chylomicrons and extremely large VLDL were significantly negatively associated with exposure to natural

environment and green space, whereas the phospholipids-to-total lipids ratio in large VLDL showed a significant positive association. VLDL cholesterol has previously been identified as part of the metabolomic signature of depression, and positively associated with depressive symptoms³³. In contrast, phospholipids, as structural core components of VLDL—particularly phosphatidylcholines and phosphatidylethanolamines—are essential constituents of cell membranes in the central nervous system, playing critical roles in maintaining neurotransmitter systems, brain function, and emotional regulation³⁴. These findings further underscore the importance of incorporating metabolic health into the understanding of the benefits of nature exposures.”

7. Lines 71-78. Air pollution is not even mentioned here while elsewhere it presented prominently.

Response: Thank you for your suggestion. We have revised the Introduction to explicitly clarify the current state of research on the association between air pollution and depression risk, highlight the existing gap in studies among cancer survivors, and indicate that we conducted relevant analyses on air pollution (**Lines 70-90**).

Introduction, Page 6, Lines 70-90: *“Ambient air pollution—including nitrogen dioxide (NO₂), nitrogen oxides (NO_x), particulate matter with aerodynamic diameter ≤10 μm (PM₁₀), and particulate matter (PM_{2.5})—is a major global health concern and has been associated with an increased risk of depression following both short- and long-term exposure^{35,36}. However, its impact on depression among cancer survivors—a population particularly vulnerable to mental health issues—remains unclear. A recent study has identified several new metabolic effectors associated with depression, revealing dysregulation of multiple metabolites in the condition³⁷. Metabolomics enables comprehensive analysis of metabolites in biological samples, providing valuable insights into the dynamic metabolic processes influenced by green space, blue space,*

natural environment, and air pollutants. However, evidence remains limited regarding the specific metabolites associated with these environmental exposures and whether disrupted circulating metabolism serves as a key intermediary linking such exposures to the development of depression.

Herein, we conducted a prospective investigation into the associations of green space, blue space, the natural environment, and ambient air pollution with the risk of depression among cancer survivors. We further identified metabolic signatures associated with environmental exposures, prospectively examined their associations with depression risk among cancer survivors, and explored the potential mediating effects of these metabolic signatures on the associations between environmental exposures and depression.”

8. Line 89. The statement about low intake of fruits and vegetables in individuals diagnosed with depression is misleading. According to Table 1, depressed individuals consumed substantially more fruits and vegetables than controls: 49.7% vs. 28.7% reported high consumption and 42.0% vs. 64.9% reported no intake, respectively.

Response: Apologies for the confusion and the inaccurate description. In the revised manuscript, to more comprehensively capture dietary patterns ³⁸, we evaluated diet quality using a healthy diet score derived from the intake of vegetables, fruits, fish, unprocessed red meat, and processed meat (Field IDs: 1289, 1299, 1309, 1319, 1329, 1339, 1349, 1369, 1379, 1389). Specifically, one point was awarded for each of the following favorable dietary components: (1) vegetable intake of at least four tablespoons per day; (2) fruit intake of at least three pieces per day; (3) fish consumption at least twice per week; (4) unprocessed red meat intake no more than twice per week; and (5) processed meat intake no more than twice per week. The total score ranged from 0 to 5, with a score of ≥ 4 considered indicative of a healthy diet. In the depression group, 64.9% of participants had a healthy diet, compared to 63.2% in the non-

depression group, with no statistically significant difference observed between the two groups ($P = 0.224$). We have revised the results accordingly to ensure accurate and objective reporting (**Lines 94-98**).

Results, Page 6, Lines 94-98: *“Compared to participants without depression, those diagnosed with depression were younger, more likely to be female, had lower educational levels and household income, a higher rate of employment, a higher prevalence of obesity, lower physical activity, higher rates of current smoking, lower rates of current drinking, and comparable dietary health status (Table 1).”*

Table 1. Baseline characteristics of cancer survivors in this study.

Variable	Total	Individuals without depression	Individuals with depression	P value
Number of participants	21507	20080	1427	
Age (year)^a	62.00 [56.00, 66.00]	62.00 [56.00, 66.00]	61.00 [55.00, 65.00]	<0.001
Sex, n (%)				
Female	13414 (62.4)	12382 (61.7)	1032 (72.3)	<0.001
Male	8093 (37.6)	7698 (38.3)	395 (27.7)	
Ethnicity, n (%)				
Non-white	1654 (7.7)	1553 (7.7)	101 (7.1)	0.397
White	19853 (92.3)	18527 (92.3)	1326 (92.9)	
Educational level, n (%)				
University or college	6386 (29.7)	6058 (30.2)	328 (23.0)	<0.001
Other	15121 (70.3)	14022 (69.8)	1099 (77.0)	
BMI, n (%)				
Normal	7168 (33.3)	6773 (33.7)	395 (27.7)	<0.001
Overweight	8930 (41.5)	8375 (41.7)	555 (38.9)	
Obese	5409 (25.1)	4932 (24.6)	477 (33.4)	
Employment status, n (%)				
Currently unemployed	8929 (41.5)	8438 (42.0)	491 (34.4)	<0.001
Currently employed	12578 (58.5)	11642 (58.0)	936 (65.6)	
Household income, n (%)				
Low	6599 (30.7)	5979 (29.8)	620 (43.4)	<0.001
Middle	11133 (51.8)	10469 (52.1)	664 (46.5)	
High	3775 (17.6)	3632 (18.1)	143 (10.0)	
Physical activity (MET), n (%)^b				
Low	10930 (50.8)	10157 (50.6)	773 (54.2)	0.010
High	10577 (49.2)	9923 (49.4)	654 (45.8)	
Smoking status, n (%)				
Not current	19656 (91.4)	18444 (91.9)	1212 (84.9)	<0.001
Current	1851 (8.6)	1636 (8.1)	215 (15.1)	
Drinking status, n (%)				
Not current	1837 (8.5)	1659 (8.3)	178 (12.5)	<0.001
Current	19670 (91.5)	18421 (91.7)	1249 (87.5)	
Diet, n (%)				
Healthy	7881 (36.6)	7380 (36.8)	501 (35.1)	0.224
Not healthy	13626 (63.4)	12700 (63.2)	926 (64.9)	
Antidepressant use, n (%)				
No	19886 (92.5)	19098 (95.1)	788 (55.2)	<0.001
Yes	1621 (7.5)	982 (4.9)	639 (44.8)	

^a, age is presented as median (25th, 75th percentile); ^b, physical activity: MET scores (range 0-21, positively correlated with weekly physical activity) below and above the cohort median represent low and high physical activity, respectively.

Abbreviations: BMI, body mass index; MET, metabolic equivalent of task.

9. Lines 133-134. This subtitle implies that the observed negative correlation was entirely due to green space reducing air pollution (cause-effect). The evidence of green spaces reducing air pollution (rather than being negatively correlated with it) is not very strong. A recent review estimated that increasing residential greenness by one standard deviation would reduce air pollution by a mere 0.8% (Venter Z, Hassani A, Stange E, Schneider P, Castell N. Reassessing the role of urban green space in air pollution control. Proc Natl Acad Sci. 2024;121).

Response: Thank you for your insightful suggestion. We have followed your advice and, in the updated manuscript, changed the title to *“Residential Green Space, Air Pollution, and Related Metabolites in Association with Depression Risk Among Cancer Survivors”*.

In the updated manuscript, air pollutants—including NO₂, NO_x, PM_{2.5}, PM₁₀, and air pollution score—were treated as one of the primary exposures to analyze their association with depression risk among cancer survivors. And then we explore the potential mediating role of air pollutants in the relationship between environmental exposures and depression in this population. Notably, our updated findings seem to align with your perspective. Although green space, blue space, and natural environmental were negatively correlated with air pollutants (**Supplementary Table S2**), and air pollutants were associated with increased depression risk among cancer survivors (**Table 2**), the mediating effect of air pollutants was not statistically significant (**Supplementary Tables S5-6**). This suggests that the role of green space in reducing depression through lowering air pollution may not be as strong as previously assumed. Instead, these environmental factors may exert relatively independent effects on depression risk in cancer survivors. In support of our findings, we cited the study you recommended by Zander S. Venter et al., published in Proceedings of the National Academy of Sciences (PNAS) in the Discussion section (**Lines**

318-326).

Furthermore, we performed a joint analysis of air pollutants and green space, blue space, and natural environment, and found that lower levels of air pollution, combined with higher levels of green space, blue space, and natural environment, were associated with a stronger protective effect against depression among cancer survivors (**Figures 2 and 3**). In addition, we identified metabolites associated with the air pollution score, and investigated their association with depression risk as well as their potential mediating roles in the air pollution–depression pathway (**Figure 5**).

Discussion, Page 18, Lines 318-326: *“Another key finding was that high levels of green space and natural environment, combined with low levels of air pollutants, conferred stronger protective effects against depression among cancer survivors. Interestingly, despite the negative correlation between air pollution and green space/natural environment, mediation analysis suggested that air pollutants did not significantly mediate this relationship, indicating their independent contributions to depression risk in this population. A recent review estimated that increasing residential greenness by one standard deviation would reduce air pollution by only 0.8%, supporting our finding that its capacity to mitigate air pollution is limited ²⁴.”*

A

B

C

Figure 5. Air pollution score-related metabolic signature (A), its associations with risk of incident depression among cancer survivors (B), and mediating effects (C) in the relationship between air pollution score and depression.

Metabolic signatures were categorized into quintiles: low (lowest quintile), moderate (quintiles 2–4), and high (highest quintile). Model 1 was adjusted for age, sex and ethnicity; Model 2 was further adjusted for educational level, household income and employment status; Model 3 was fully adjusted for all aforementioned variables as well as body mass index, smoking status, drinking status, physical activity, diet and antidepressant use. The mediation analysis was conducted using Model 3. Abbreviations: APS, air pollution score; SD, standard deviation; PAF, population attributable fraction; CI, confidence interval.

10. Line 142. I think that the mediation proportion (masking effect) should be calculated as a percentage of the total effect (the effect of exposure variable not adjusted for the mediator) rather than the direct effect (the effect of exposure variable after adjusting for the mediator).

Response: Thank you for your comment. In the original analysis, because the direct and indirect effects are in opposite directions (under the influence of air pollution, the indirect effect of green space opposes its direct effect), this represents a typical masking effect. Therefore, we calculated the proportion of the mediation effect relative to the direct effect rather than the total effect. This is because the direct and indirect effects often operate in opposite directions, which can lead to a substantially attenuated or even reversed total effect. Using the total effect as the denominator in such cases may result in unstable or misleading estimates, particularly when the total effect approaches zero. In contrast, using the direct effect as the reference provides a more meaningful and interpretable measure of the extent to which the mediator suppresses or counteracts the relationship between the exposure and outcome.

Importantly, the updated results indicated that neither individual air pollutants nor the air pollution score appeared to mediate the association between green space, blue space, and natural environment and the risk of depression among cancer survivors. Moreover, since the directions of most indirect and direct effects' β coefficients were consistent, following your suggestion, we applied the classic proportion mediated calculation method, defined as the ratio of the mediation effect to the total effect (**Supplementary Tables S5-6**).

11. Line 155. Provide a list of 249 metabolites analyzed in supplemental materials.

Response: We appreciate your suggestion and have added a list of 249 biomarkers in the supplementary materials (**Supplementary Table S7**).

12. Lines 158 -159. Are these effects on large VLDL and other biomarkers beneficial according to the existing knowledge? Provide some interpretation of these findings.

Response: Thank you for your valuable suggestion. We further interpreted these findings in the context of existing evidence (**Lines 349-378**), including discussions of green space– or natural environment–related metabolites, such as large VLDL particles and several metabolites potentially associated with depression risk, including three amino acids—leucine, glutamine, and tyrosine.

Discussion, Pages 20-21, Lines 349-378: *“Metabolic signatures related to the natural environment and green space act as potential mediators in their associations with depression among cancer survivors. Specifically, we found that three amino acids—leucine, glutamine, and tyrosine—were consistently positively associated with green space and the natural environment. Previous studies have shown that leucine supplementation can alleviate social avoidance and depressive-like behaviors in mice ²⁷. The glutamate–glutamine cycle and glutamatergic neurotransmission play critical roles in normal brain function, highlighting the importance of maintaining glutamine homeostasis in the brain. A deficiency of neuronal glutamine in the medial prefrontal cortex of rodents has been linked to depressive behaviors and mild cognitive impairment ²⁸. Furthermore, mendelian randomization evidence supports a potential causal relationship between downregulation of glutamine and major depressive disorder, anhedonia, fatigue, and low mood ²⁹. For tyrosine, previous studies have reported significantly reduced levels in the blood and urine of patients with depression ³⁰, with concentrations showing a significant negative correlation with Hamilton Depression Rating Scale (HAM-D) scores ³¹. Additionally, 3-Hydroxybutyrate, a ketone body that was consistently positively associated with green space and the natural environment, has been shown to significantly reduce anxiety-like behaviors in rats when administered through long-term*

*dietary supplementation*³². Interestingly, for large Very Low-Density Lipoprotein (VLDL), we found that cholesteryl esters in chylomicrons and extremely large VLDL were significantly negatively associated with exposure to natural environment and green space, whereas the phospholipids-to-total lipids ratio in large VLDL showed a significant positive association. VLDL cholesterol has previously been identified as part of the metabolomic signature of depression, and positively associated with depressive symptoms³³. In contrast, phospholipids, as structural core components of VLDL—particularly phosphatidylcholines and phosphatidylethanolamines—are essential constituents of cell membranes in the central nervous system, playing critical roles in maintaining neurotransmitter systems, brain function, and emotional regulation³⁴. These findings further underscore the importance of incorporating metabolic health into the understanding of the benefits of nature exposure.”

13. Lines 220-221. The authors should discuss and interpret the direction of the mediation effects.

Response: Thank you for your valuable suggestion. The direction of the mediation effects has been further interpreted in the figure legends, as well as in the Results and Discussion sections.

14. Lines 245-246 “...excluded patients who experienced depression... within three years of cancer diagnosis”. This is not consistent with the Methods section where the authors stated that individuals who had depression within 2 years of cancer were excluded.

Response: Apologies for the inconsistency. In the original manuscript, we actually excluded patients who experienced depression within three years of their cancer diagnosis. In the updated manuscript, we treated this as part of the sensitivity analyses. Specifically, we conducted additional sensitivity analyses excluding depression cases occurring within one year and within three years

following the cancer diagnosis. The results of these sensitivity analyses were generally consistent with those of the main analysis (**Supplementary Tables S15 and S16**). We have made corresponding revisions to the Methods section to reflect these changes (**Lines 575-582**).

Methods, Pages 31-32, Lines 575-582: *“Several sensitivity analyses were conducted to confirm the robustness of our main findings, including: (1) separate adjustment for air pollutants ($PM_{2.5}$, PM_{10} , NO_2 , and NO_x); (2) simultaneous adjustment for all four air pollutants; (3) adjustment for the APS; (4) adjustment for cancer type; (5) adjustment for sleep pattern; (6) exclusion of participants using antidepressants at baseline; (7) exclusion of depression cases within one year post-cancer; (8) exclusion of depression cases within three years post-cancer; (9) exclusion of death cases within ten years post-cancer; and (10) restriction to individuals living at their current address for over 10 years before baseline.”*

15. Lines 250-252. Again, explain the directions of direct and indirect effects and what it means in terms of the effects of air pollution on depression. Present associations between air pollution and depression elsewhere in the paper.

Response: We apologize for any confusion regarding the directions of direct and indirect effects. Mediation analysis was conducted using an accelerated failure time (AFT) model implemented via ‘survreg ()’, which models log-transformed survival time and differs fundamentally from the Cox proportional hazards model (‘coxph ()’) in its parametric assumptions and interpretation of coefficients. In the AFT framework: (1) The regression coefficient (β) reflects the effect of an exposure on the logarithm of survival time (i.e., time to depression onset); (2) A positive β indicates a longer time to event (i.e., a lower risk of depression), while a negative β indicates a shorter time to event (i.e., a higher risk); and (3) Therefore, the direction of effects in the AFT model is inversely interpreted compared to hazard ratios (HRs) in Cox models—for

example, a protective factor would yield $HR < 1$ in Cox but a positive β in AFT. For green space, a positive β indicates a prolonged time to depression onset, reflecting a reduced risk, whereas a negative β suggests a shortened time to onset, reflecting an increased risk of depression. We adopted 'survreg ()' for mediation analysis because there is currently no well-established and robust method for performing mediation within a nested Cox proportional hazards framework. To prevent potential misunderstanding, we have added detailed clarifications in the corresponding table notes and figure captions to aid reader interpretation.

Importantly, we fully agree with you and Reviewer 1 that examining the association between air pollutants and depression is of great relevance. In response, we assessed the impact of individual air pollutants on depression risk and further developed an air pollution score to evaluate its association with depression. Our results showed that higher levels of NO_2 (HR for tertile 3 vs. tertile 1 = 1.299, 95% CI: 1.142, 1.477), NO_x , (HR = 1.334, 95% CI: 1.173, 1.516), and $PM_{2.5}$ were significantly associated with increased risk of depression. In the fully adjusted Model 3, both NO_2 (HR = 1.140, 95% CI: 1.001, 1.299) and NO_x , (HR = 1.143, 95% CI: 1.004, 1.303) remained significantly associated with elevated depression risk (**Table 2**). Furthermore, the air pollution score—constructed by combining NO_2 , NO_x , and $PM_{2.5}$ that were significant in Model 1—was also found to be positively associated with depression risk in a dose-response manner (**Table 2**).

Accordingly, in the updated manuscript, we described the associations between air pollution and depression in the general population in the Introduction section (**Lines 70-75**). We added the analysis results on the association between air pollution and depression among cancer survivors in the Results section (**Lines 136-150**), and further discussed the corresponding findings in the Discussion section (**Lines 299-317**).

Introduction, Page 5, Lines 70-75: “Ambient air pollution—including nitrogen dioxide (NO₂), nitrogen oxides (NO_x), particulate matter with aerodynamic diameter ≤10 μm (PM₁₀), and particulate matter (PM_{2.5})—is a major global health concern and has been associated with an increased risk of depression following both short- and long-term exposure^{35,36}. However, its impact on depression among cancer survivors—a population particularly vulnerable to mental health issues—remains unclear.”

Results, Pages 8-9, Lines 136-150: “Associations between air pollutants and depression risk among cancer survivors In Models 1 and 2, higher levels of air pollutants (NO₂, NO_x, and PM_{2.5}) were associated with an increased risk of depression among cancer survivors (**Table 2**). In the fully adjusted Model 3, cancer survivors exposed to the highest tertile of NO₂ had a 14.0% higher risk of depression compared to those in the lowest tertile (HR = 1.140, 95% CI: 1.001, 1.299, P = 0.048). Similarly, exposure to the highest tertile of NO_x was associated with a 14.3% higher risk of depression (HR = 1.143, 95% CI: 1.004, 1.303, P = 0.039). Subsequently, an air pollution score (APS) was calculated based on weighted concentrations of NO₂, NO_x, and PM_{2.5}. Results from Model 3 indicated tertile 3 of the APS was associated with a 15.2% higher risk of depression (HR = 1.152; 95% CI: 1.011, 1.312; P = 0.033) compared to the lowest tertile. Cox regression models revealed significant positive dose–response associations between NO₂, NO_x, and the APS and depression risk, when tertiles were treated as a continuous variable (P for trend = 0.045 for NO₂; 0.039 for NO_x; and 0.029 for APS, respectively).”

Discussion, Pages 17-18, Lines 299-317: “Furthermore, our results indicated that high levels (tertile 3) of NO₂ and NO_x were associated with a 14.0% and 14.3% increased risk of depression among cancer survivors, respectively, compared to tertile 1, while a high APS was linked to a 15.2% increased risk. Each 5% increment in PM_{2.5} was also found to be associated with an increased

risk of depression among cancer survivors. A previous study among Korean adult cancer survivors reported that exposure to PM_{10} was associated with an increased risk of depressive symptoms and suicidal ideation, whereas no significant association was observed for NO_2 ³⁹. Notably, although this study applied propensity score matching between cancer survivors and the general population, its cross-sectional design limited the ability to infer causal relationships between air pollution and depression. In addition, the lack of data on NO_x and $PM_{2.5}$ further hindered a comprehensive assessment of air pollutant exposures. In our study, we did not observe a significant association between PM_{10} exposure and depression risk among cancer survivors. Compared with $PM_{2.5}$, NO_2 and NO_x , the effect of PM_{10} on depression appeared to be weaker, potentially requiring a longer exposure duration to trigger its impact⁴⁰. Our study further confirmed the detrimental effects of air pollutants on depression risk among cancer survivors—a population particularly susceptible to mental health disorders—highlighting that controlling air pollution may offer additional mental health benefits, especially for vulnerable groups.”

16. Lines 253-258. Indeed, previous studies reported negative associations between air pollution and depression. However, this study apparently produced contradictory results which are not reported explicitly by the authors. As the associations between green space and air pollution and between green space and depression are both negative, the observed patterns (adjusting for air pollution increased the magnitude of the observed protective effects of green space on depression; direct and indirect effects have different signs) could only be explained by a negative (inverse) association between air pollution and depression. Perhaps there was an error in statistical analysis; otherwise, the apparent protective effect of air pollution on depression is an intriguing finding that needs to be explained and discussed in the text.

Response: Thank you for your comment. In the updated manuscript, following

the suggestions from you and other reviewers, and to further clarify the association between air pollution and depression among cancer survivors, we added analyses examining the relationship between air pollution (including NO₂, NO_x, PM_{2.5}, and PM₁₀) and depression risk. The results showed that in Model 1, higher exposure to NO₂, (HR for tertile 3 vs tertile 1 = 1.299, 95% CI: 1.142, 1.477), NO_x (HR = 1.334, 95% CI: 1.173, 1.516), and PM_{2.5} (HR = 1.298, 95% CI: 1.143, 1.473) were associated with an increased risk of depression among cancer survivors (**Table 2**). In the fully adjusted Model 3, NO₂ (HR for tertile 3 vs tertile 1 = 1.140, 95% CI: 1.001, 1.299) and NO_x (HR = 1.143, 95% CI: 1.004, 1.303) remained significantly associated with elevated depression risk (**Table 2**). Subsequently, we calculated a weighted air pollution score and similarly found a positive association between the air pollution score and depression risk among cancer survivors (HR_{per 5% increment} = 1.035, 95% CI: 1.004, 1.067, P = 0.028), with evidence of a dose–response relationship (P for trend = 0.029).

In fact, in the original manuscript, the direct and indirect effects had opposite signs, which did not indicate a negative association between air pollution and depression. Rather, it suggested that the effect of air pollution on increasing depression risk masked the protective effect of green space. This was also the reason why we originally calculated the masking effect instead of the proportion mediated. In the updated manuscript, however, for the mediation analyses involving air pollution, the direct and indirect effects mostly had the same direction. Therefore, we adopted your suggestion and quantified the proportion mediated as the ratio of the mediation effect to the total effect. Finally, we have provided a more detailed description and discussion of the revised analyses on the association between air pollution and depression among cancer survivors, along with the corresponding mediation analyses.

17. Lines 293-295 and 305-306. It would be helpful to have more information on the analysis of metabolic biomarkers. The metabolic signatures associated

with green spaces/natural environment could be influenced by social and behavioral confounding factors that are correlated with residential greenness and affect metabolism. For example, individuals who self-selected to live in greener areas might have more health-oriented behavior and healthier nutrition.

Response: Thank you for your valuable suggestion. We agree with your point that the metabolic signatures associated with green space and the natural environment could be influenced by social and behavioral confounding factors. Therefore, we carefully adjusted for these potential confounders when identifying the relevant metabolites. Specifically, in addition to baseline characteristics, we adjusted for potential social and behavioral confounders such as smoking status, drinking status, body mass index, physical activity, and diet. We have added more information on the analysis of metabolic biomarkers in the relevant section of the Discussion and have provided appropriate interpretations in light of previous literature (**Lines 349-378**).

Additionally, as you rightly emphasized, social and behavioral factors may not only influence metabolic profiles but also modify the association between environmental exposures and depression risk among cancer survivors. Interestingly, in our stratified analyses, we found that the association between NO₂ exposure and increased depression risk was more pronounced among cancer survivors with an unhealthy diet (HR for tertile 3 vs. tertile 1 = 1.256; 95% CI: 1.065, 1.481), compared to those with a healthy diet (P for interaction = 0.039) (**Supplementary Figure S9**). This finding suggests that a healthy diet may help buffer or partially mitigate the adverse effects of air pollution on depression risk in this population. These results underscore the importance of considering individual lifestyle factors when assessing environmental health risks.

Discussion, Pages 19-21, Lines 349-378: *“Metabolic signatures related to the natural environment and green space act as potential mediators in their*

associations with depression among cancer survivors. Specifically, we found that three amino acids—leucine, glutamine, and tyrosine—were consistently positively associated with green space and the natural environment. Previous studies have shown that leucine supplementation can alleviate social avoidance and depressive-like behaviors in mice ²⁷. The glutamate–glutamine cycle and glutamatergic neurotransmission play critical roles in normal brain function, highlighting the importance of maintaining glutamine homeostasis in the brain. A deficiency of neuronal glutamine in the medial prefrontal cortex of rodents has been linked to depressive behaviors and mild cognitive impairment ²⁸. Furthermore, mendelian randomization evidence supports a potential causal relationship between downregulation of glutamine and major depressive disorder, anhedonia, fatigue, and low mood ²⁹. For tyrosine, previous studies have reported significantly reduced levels in the blood and urine of patients with depression ³⁰, with concentrations showing a significant negative correlation with Hamilton Depression Rating Scale (HAM-D) scores ³¹. Additionally, 3-Hydroxybutyrate, a ketone body that was consistently positively associated with green space and the natural environment, has been shown to significantly reduce anxiety-like behaviors in rats when administered through long-term dietary supplementation ³². Interestingly, for large Very Low-Density Lipoprotein (VLDL), we found that cholesteryl esters in chylomicrons and extremely large VLDL were significantly negatively associated with exposure to natural environment and green space, whereas the phospholipids-to-total lipids ratio in large VLDL showed a significant positive association. VLDL cholesterol has previously been identified as part of the metabolomic signature of depression, and positively associated with depressive symptoms ³³. In contrast, phospholipids, as structural core components of VLDL—particularly phosphatidylcholines and phosphatidylethanolamines—are essential constituents of cell membranes in the central nervous system, playing critical roles in maintaining neurotransmitter systems, brain function, and emotional

regulation³⁴. These findings further underscore the importance of incorporating metabolic health into the understanding of the benefits of nature exposures.”

Supplementary Figure S9. Associations between air pollutants and depression risk in stratified analyses.

The analysis was conducted using Model 3, adjusted for age, sex and ethnicity, educational level, household income, employment status, body mass index, smoking status, drinking status, physical activity, diet and antidepressant use.

Abbreviations: CI, confidence interval; HR, hazard ratio; NO₂, nitrogen dioxide; NO_x, nitrogen oxides; PM₁₀, particulate matter (PM) with aerodynamic diameter ≤10 μm; PM_{2.5}, PM with aerodynamic diameter < 2.5 μm.

18. Lines 337-330. It is the other way around: air pollution mediated the effect of greenness.

Response: Thank you for your comments. Based on the updated results, we have rewritten the corresponding sentence.

19. Line 355. Two years here, three years elsewhere.

Response: Apologies for the inconsistency. In the original manuscript, we actually excluded patients who experienced depression within three years of their cancer diagnosis. In the updated manuscript, we treated this as part of the sensitivity analyses. Specifically, we conducted additional sensitivity analyses excluding depression cases occurring within one year and within three years following the cancer diagnosis. The results of these sensitivity analyses were generally consistent with those of the main analysis (**Supplementary Tables S15 and S16**). We have made corresponding revisions to the Methods section to reflect these changes (**Lines 432-438; Lines 575-587**).

Methods, Pages 23-24, Lines 432-438: *“The cohort initially included 93,677 participants with cancer. We excluded individuals diagnosed with cancer after baseline ($n = 65,195$), those with less than five years of cancer survivorship ($n = 714$)^{6,23}, participants with missing data on green space, blue space, and natural environment exposures ($n = 3,393$), those with a diagnosis of depression prior to baseline ($n = 303$), and those diagnosed with other psychiatric disorders ($n = 2,565$). After these exclusions, a total of 21,507 cancer survivors were included in the final analysis (**Supplementary Figures S11 and S12**).”*

Methods, Pages 31, Lines 575-587: *“Several sensitivity analyses were conducted to confirm the robustness of our main findings, including: (1) separate adjustment for air pollutants ($PM_{2.5}$, PM_{10} , NO_2 , and NO_x); (2) simultaneous adjustment for all four air pollutants; (3) adjustment for the APS;*

(4) adjustment for cancer type; (5) adjustment for sleep pattern; (6) exclusion of participants using antidepressants at baseline; (7) exclusion of depression cases within one year post-cancer; (8) exclusion of depression cases within three years post-cancer; (9) exclusion of death cases within ten years post-cancer; and (10) restriction to individuals living at their current address for over 10 years before baseline. Furthermore, we conducted separate analyses for cancer subtypes with more than 1,000 cases (breast cancer, melanoma skin cancer, prostate cancer, and colorectal cancer) as well as for specific cancer types (lung cancer and non-melanoma skin cancer). In the analyses of melanoma skin cancer and non-melanoma skin cancer, we additionally adjusted for daily sun exposure duration based on Model 3.”

20. Line 388. List these 249 biomarkers in Supplements.

Response: We appreciate your suggestion and have added a list of 249 biomarkers in the supplementary materials (**Supplementary Table S7**).

21. Lines 425-431. This description needs to be edited. It seems that models 2 and 3 were adjusted for age, sex, and ethnicity. Was Model 1 adjusted for these variables? The current description implies that age, sex and ethnicity were only used as stratification factors in Model 1.

Response: Apologies for the confusion. Thank you for your detailed comments. In fact, Model 1 is a Cox proportional hazards model adjusted for age, sex, and ethnicity, where age is a continuous variable, and sex and ethnicity are categorical variables. We have made revisions and standardized the description throughout the manuscript (**Lines 531-535**).

Methods, Page 29, Lines 531-535: *“We constructed three models by sequentially adding covariates: Model 1 was adjusted for age, sex and ethnicity; Model 2 was further adjusted for educational level, household income and employment status; and Model 3 was fully adjusted for all aforementioned*

variables as well as BMI, smoking status, drinking status, physical activity, diet and antidepressant use.”

22. Lines 435-436. An incomplete sentence.

Response: Apologies for the grammatical issue. We have revised the incomplete sentences and reviewed the entire manuscript for similar issues (**Lines 539-542**).

Methods, Page 29, Lines 539-542: *“To explore potential interactions, we used the ‘mediation’ package of R based on the accelerated failure time (AFT) model (via ‘survreg ()’) to examine whether air pollutants mediate the associations between green space, blue space, natural environment, and depression risk.”*

23. Line 468. Explain that PAF means Population Attributable Fraction.

Response: Thank you for your comments. Apologies for the confusion caused by the abbreviation, we have added the full name of PAF: (population attributable fraction) (**Lines 560-563**).

Methods, Page 30, Lines 560-563: *“The population attributable fraction (PAF) was calculated to estimate the proportion of depression cases among cancer survivors that could potentially be prevented if specific risk factors were eliminated.”*

A

B

C

Figure 4. Green space- and natural environment-related metabolic signatures (A), their associations with risk of incident depression among cancer survivors (B), and their mediating effects (C) in the relationship between environmental exposure and depression.

Metabolic signatures were categorized into quintiles: low (lowest quintile), moderate (quintiles 2–4), and high (highest quintile). Model 1 was adjusted

for age, sex and ethnicity; Model 2 was further adjusted for educational level, household income and employment status; Model 3 was fully adjusted for all aforementioned variables as well as body mass index, smoking status, drinking status, physical activity, diet and antidepressant use. The mediation analysis was conducted using Model 3. Abbreviations: SD, standard deviation; PAF, population attributable fraction; CI, confidence interval.

A

B

C

Figure 5. Air pollution score-related metabolic signature (A), its associations with risk of incident depression among cancer survivors (B),

and mediating effects (C) in the relationship between air pollution score and depression.

Metabolic signatures were categorized into quintiles: low (lowest quintile), moderate (quintiles 2–4), and high (highest quintile). Model 1 was adjusted for age, sex and ethnicity; Model 2 was further adjusted for educational level, household income and employment status; Model 3 was fully adjusted for all aforementioned variables as well as body mass index, smoking status, drinking status, physical activity, diet and antidepressant use. The mediation analysis was conducted using Model 3. Abbreviations: APS, air pollution score; SD, standard deviation; PAF, population attributable fraction; CI, confidence interval.

24. Lines 469-474. This is an incomplete sentence.

Response: Apologies for the grammatical issue. We have revised the incomplete sentences and reviewed the entire manuscript for similar issues.

25. Table 2. How are these regression coefficients related to HR values presented in Fig. 3 and elsewhere? Please explain why the direct effect is positive and the indirect effect is negative. According to results presented elsewhere (HR values below 1), the regression coefficients for direct effects should be negative.

Response: We apologize for the potential confusion caused by the difference between the regression coefficients and HR. In the mediation analysis in Table 2, mediation was conducted using the accelerated failure time (AFT) model fitted via 'survreg ()', which differs from the Cox proportional hazards model ('coxph ()'). We adopted 'survreg ()' for mediation analysis because, currently, there is no well-established and robust method to perform mediation within the nested Cox proportional hazards framework. In the AFT framework: (1) The regression coefficient (β) represents the effect of an exposure on the log of survival time (i.e., time to depression onset). (2) A positive β indicates a longer time to event (lower risk of depression), whereas a negative β indicates a shorter time to event (higher risk). (3) Therefore, the direction of the effect in AFT is inversely related to the interpretation of HRs in Cox models. For instance, a protective factor would yield $HR < 1$ in Cox but a positive β in AFT.

Without considering the underlying assumptions of the AFT model and the Cox proportional hazards model, $e^{(-\beta)}$ can be approximately equal to HR. However, directly using $e^{(-\beta)}$ to convert the β from the AFT model to HR in the Cox model is not appropriate. Although both models are used for survival analysis, they approach the problem from different perspectives—risk and time-to-event, respectively. The conversion between the two requires more complex

considerations and may involve different assumptions regarding survival distributions. Therefore, in the manuscript, we did not choose to convert the β into the more easily interpretable HR. To avoid potential misunderstanding, we have added detailed clarifications in the corresponding table notes and figure captions to aid readers' understanding.

26. Figure 1. Please explain why all confidence intervals collapse at the zero green space or natural environment value. Very few individuals had zero greenness values, so the confidence interval at the left end should be wide.

Response: In the RCS model, a specific point (such as the minimum value) is usually set as the reference or baseline. When plotting the RCS, we set the minimum value as the reference point, and thus, the regression coefficient for this point is set to 0. As a result, the effects of all other points (relative to the reference point) are represented through their coefficients. Therefore, the confidence interval for the minimum value is close to 0 because the regression coefficient at this point is by default set to zero.

Setting the minimum value as the reference point is primarily considered because it makes the model coefficients more intuitive. The coefficient for the reference point is set to 0, and the effects of other points are explained relative to the reference point. This allows us to clearly understand how the effects of other points change.

27. Figure 3. Explain the values of the direct effect and mediation effect and how HR values can be estimated from these values. I assume that these are regression coefficients which would need to be multiplied by 0.05 for 5% increase in green space and exponentiated. The green space=> depression direct effect values is 5.08. This positive value corresponds with HR>1 (detrimental effect), that is contrary to the text and other figures and tables.

Response: We apologize for the potential confusion caused by the mediation

analysis. In the mediation analysis, green space was treated as a continuous variable. Both the direct and mediation effects are represented as regression coefficients. However, it is important to note that the mediation analysis was conducted using the accelerated failure time (AFT) model fitted via 'survreg ()', which differs from the Cox proportional hazards model ('coxph ()'). In the AFT framework: (1) The regression coefficient (β) represents the effect of an exposure on the log of survival time (i.e., time to depression onset). (2) A positive β indicates a longer time to event (lower risk of depression), whereas a negative β indicates a shorter time to event (higher risk). (3) Therefore, the direction of the effect in AFT is inversely related to the interpretation of HRs in Cox models. For instance, a protective factor would yield an HR less than 1 in the Cox model but a positive coefficient ($\beta > 0$) in the AFT model.

Additionally, directly using $e^{-\beta}$ to convert the β from the AFT model to HR in the Cox model is not appropriate. Although both models are used for survival analysis, they approach the problem from different perspectives—risk and time-to-event, respectively. The conversion between the two requires more complex considerations and may involve different assumptions regarding survival distributions. Therefore, in the manuscript, we did not choose to convert the β into the more easily interpretable HR.

Finally, we adopted 'survreg ()' for mediation analysis because currently there is no well-established and robust method to perform mediation within the nested Cox proportional hazards framework. To avoid potential misunderstanding, we have added detailed clarifications in the corresponding table notes and figure captions.

28. Figure S1. Explain two lines that seem to be similar: ...6,447 diagnosed within 3 years... AND ...25,758 ... diagnosed withing three years... What diseases were they diagnosed with? Which one is about depression?

Response: Apologies for the confusion in the inclusion process. In accordance with reviewer 1's comments, we revised the inclusion and exclusion criteria. Specifically, we included participants diagnosed with cancer before baseline and with a cancer survivorship of ≥ 5 years. We excluded individuals with missing data on green, garden, blue, or nature space exposure, those diagnosed with depression prior to baseline, and those diagnosed with other psychiatric disorders. After applying these criteria, a total of 21,507 participants were included in the final analysis.

29. Figure S2. All values for HR (95% CI) are unreadable. Some funny characters are displayed instead of numbers. Model 3 was adjusted for age, sex, ethnicity. However, elsewhere descriptions for model 3 do not include these covariates. Please clarify which models were adjusted for age, sex and ethnicity.

Response: We apologize for the formatting issue in the original Figure S3, which rendered the HR (95% CI) values unreadable. We have revised and re-plotted Figure S3 to ensure that all elements are clearly visible (**Supplementary Figures S7-8**). Additionally, the covariates adjusted for in each model are as follows: Model 1 adjusted for age, sex and ethnicity; Model 2 adjusted for age, sex, ethnicity, educational level, household income and employment status; and Model 3 (the fully adjusted model) adjusted for age, sex, ethnicity, educational level, household income, employment status, body mass index, physical activity, smoking status, drinking status, diet and antidepressant use. All three models consistently adjusted for age, sex, and ethnicity. To ensure clarity, we have harmonized the descriptions of covariate adjustments across the manuscript.

Supplementary Figure S7. Associations between exposure to green space, blue space and natural environment at 300 m buffer and depression risk in stratified analyses.

The analysis was conducted using Model 3, adjusted for age, sex and ethnicity, educational level, household income, employment status, body mass index, smoking status, drinking status, physical activity, diet and antidepressant use.

Abbreviations: CI, confidence interval; HR, hazard ratio.

Supplementary Figure S8. Associations between exposure to green space, blue space and natural environment at 1000 m buffer and depression risk in stratified analyses.

The analysis was conducted using Model 3, adjusted for age, sex and ethnicity, educational level, household income, employment status, body mass index, smoking status, drinking status, physical activity, diet and antidepressant use.

Abbreviations: CI, confidence interval; HR, hazard ratio.

30. Table S4. Please include depression in this table and provide point biserial correlation coefficients with air pollutants and other continuous variables.

Response: Thank you for your suggestion. We have added depression to this table and provided point-biserial correlation coefficients with air pollutants and other continuous variables (**Supplementary Table S2**).

Supplementary Table S2. Correlation matrix of environmental exposures.

Exposures	Green space (300m buffer)	Green space (1000m buffer)	Blue space (300m buffer)	Blue space (1000m buffer)	Natural environment (300m buffer)	Natural environment (1000m buffer)	NO ₂	NO _x	PM ₁₀	PM _{2.5}
Green space (300m buffer)	1									
Green space (1000m buffer)	0.853 ***	1								
Blue space (300m buffer)	0.049 ***	0.020 **	1							
Blue space (1000m buffer)	0.022 **	0.003	0.722 ***	1						
Natural environment (300m buffer)	0.884 ***	0.761 ***	0.137 ***	0.086 ***	1					
Natural environment (1000m buffer)	0.824 ***	0.968 ***	0.103 ***	0.112 ***	0.786 ***	1				
NO ₂	-0.643 ***	-0.735 ***	-0.082 ***	-0.070 ***	-0.628 ***	-0.756 ***	1			
NO _x	-0.532 ***	-0.559 ***	-0.069 ***	-0.038 ***	-0.551 ***	-0.582 ***	0.922 ***	1		
PM ₁₀	-0.419 ***	-0.390 ***	-0.052 ***	-0.013	-0.446 ***	-0.406 ***	0.520 ***	0.527 ***	1	
PM _{2.5}	-0.638 ***	-0.643 ***	-0.098 ***	-0.055 ***	-0.643 ***	-0.674 ***	0.864 ***	0.847 ***	0.548 ***	1
Depression	-0.016 *	-0.023 ***	-0.009	-0.007	-0.019 **	-0.030 ***	0.028 ***	0.030 ***	0.013	0.036 ***

*, P < 0.05; **, P < 0.01; ***, P < 0.001.

Abbreviations: NO₂, nitrogen dioxide; NO_x, nitrogen oxides; PM₁₀, particulate matter (PM) with aerodynamic diameter ≤10 μm; PM_{2.5}, PM with aerodynamic diameter < 2.5 μm.

31. Table S7. I assume that these values are regression coefficients, but it is unclear how they are related to HR values presented elsewhere. All indirect effects of air pollutants are negative while most direct effects are positive suggesting that greater green space was associated with greater risk of depression after adjusting for air pollution. This contradicts protective effects of green space presented elsewhere.

Response: Thank you for your comments. In the mediation analysis presented in the original Table S7, we employed an accelerated failure time (AFT) model using the 'survreg ()' function, which differs from the Cox proportional hazards model ('coxph ()'). In the AFT framework: (1) The regression coefficient (β) represents the effect of an exposure on the logarithm of survival time (i.e., time to depression onset); (2) $\beta > 0$ indicates a prolonged time to the event, corresponding to a lower risk of depression, while $\beta < 0$ suggests a shortened time to the event, indicating a higher risk; (3) Consequently, the interpretation of β in AFT is directionally opposite to the hazard ratio (HR) in the Cox model. For example, a protective factor would yield $HR < 1$ in Cox but $\beta > 0$ in AFT. Therefore, positive direct effects ($\beta > 0$) indicate a protective influence on depression. This aligns with the protective effects of green space discussed elsewhere.

In the revised manuscript, additional analyses indicated that NO_2 , NO_x , and the air pollution score were all positively associated with increased depression risk among cancer survivors (**Table 2**). Importantly, the updated results showed that neither individual air pollutants nor the composite air pollution score mediated the association between green space, blue space, and natural environment and depression risk among cancer survivors. Moreover, as the β coefficients for most indirect and direct effects were in the same direction, we followed your suggestion and adopted the conventional method to calculate the proportion mediated—defined as the ratio of the mediation effect to the total effect

(Supplementary Tables S5 and S6).

Additionally, in the updated analysis, although the HR slightly increased after adjusting for individual air pollutants (NO₂, NO_x, PM_{2.5}, and PM₁₀), the protective associations between green space, natural environment, and depression remained statistically significant (**Supplementary Tables S11 and S12**). We also conducted further sensitivity analyses assessing the associations between environmental exposures and depression risk, adjusting separately for all air pollutants and for the composite air pollution score, and obtained similar results (**Supplementary Table S13**). These findings suggest that, despite the negative correlation between air pollutants and green/natural environment, and the opposite directions of their associations with depression risk, their effects on depression risk among cancer survivors appear to be relatively independent. This may, at least in part, be explained by the review provided by you (Zander S. Venter et al.), which estimated that increasing residential greenness by one standard deviation reduces air pollution by only about 0.8% ²⁴.

32. Tables S11 and S12. Adjusting for air pollutants increased the effects of green space (smaller HR values that correspond to greater absolute values of negative regression coefficients). These findings correspond to negative confounding or masking effects of air pollution, which is consistent with the opposite directions of direct and indirect effects in Table 2 (but not with their signs: direct effects should be negative and indirect effects should be positive to produce the patterns in Tables S11-S12). Since air pollution is negatively associated with green spaces, it must be negatively associated with depression to produce the observed patterns. This very unexpected finding needs to be explained.

Response: Thank you for your comments. In the mediation analysis presented in the original Table 2, we employed an accelerated failure time (AFT) model using the 'survreg ()' function, which differs from the Cox proportional hazards

model ('coxph ()'). In the AFT framework: (1) The regression coefficient (β) reflects the effect of an exposure on the logarithm of survival time (i.e., time to depression onset); (2) A positive β denotes a prolonged time to the event, corresponding to a lower risk of depression, whereas a negative β implies a shortened time to the event, indicating a higher risk; (3) Accordingly, the interpretation of β in AFT is directionally opposite to the hazard ratio (HR) in the Cox model—for example, a protective factor would show $HR < 1$ in Cox but $\beta > 0$ in AFT. In our mediation analysis, positive direct effects indicate a protective influence on depression, while negative indirect effects suggest increased depression risk. We chose the 'survreg ()' function for mediation because, to date, no well-established or robust method exists for conducting mediation analysis within a nested Cox proportional hazards framework.

In the original Tables S11 and S12, adjusting for air pollutants appeared to strengthen the protective effects of green space (smaller HR values). In the updated analyses, although the HRs slightly increased after adjusting for individual air pollutants (NO₂, NO_x, PM_{2.5}, and PM₁₀), the protective associations between green space, natural environment, and depression risk remained statistically significant (**Supplementary Tables S11 and S12**).

Importantly, we fully agree with you that examining the association between air pollutants and depression is of great relevance. In response, we assessed the impact of individual air pollutants on depression risk and further developed an air pollution score to evaluate its association with depression. Our results showed that higher levels of NO₂ (HR for tertile 3 vs. tertile 1 = 1.299, 95% CI: 1.142, 1.477), NO_x, (HR = 1.334, 95% CI: 1.173, 1.516), and PM_{2.5} were significantly associated with increased risk of depression. In the fully adjusted Model 3, both NO₂ (HR = 1.140, 95% CI: 1.001, 1.299) and NO_x, (HR = 1.143, 95% CI: 1.004, 1.303) remained significantly associated with elevated depression risk. Furthermore, the air pollution score was constructed by

combining NO₂, NO_x, and PM_{2.5}. Further analysis indicated that the air pollution score was significantly associated with an increased depression risk in the final Model 3, with a HR_{per 5% increment} of 1.035 (95% CI: 1.004, 1.067) and HR_{Tertile 3 vs. Tertile 1} of 1.152 (95% CI: 1.011, 1.312) (**Table 2**).

In the updated analysis, we observed protective associations of green space (at 1000 m buffer), blue space (at 300 m buffer), and natural environment (at 1000 m buffer) with depression (**Figure 1**), as well as adverse effects of air pollutants on depression risk among cancer survivors (**Table 2**). Additionally, green space, blue space, and natural environment were inversely associated with air pollutants (**Supplementary Table S2**). However, mediation analysis did not support a mediating role of air pollution in the associations between environmental exposures and depression (**Supplementary Tables S5 and S6**). These findings suggest that, despite the negative correlation between air pollutants and green/natural environment, and the opposite directions of their associations with depression risk, their effects on depression risk among cancer survivors appear to be relatively independent. This may, at least in part, be explained by the review provided by you (Zander S. Venter et al.), which estimated that increasing residential greenness by one standard deviation reduces air pollution by only about 0.8%²⁴.

Reviewer #3 (Remarks to the Author):

Response: Thank you very much for your insightful suggestion and for taking the time to co-review our manuscript as part of the Nature Communications initiative. We sincerely appreciate your thoughtful and constructive comments, which have been tremendously helpful in improving the quality and clarity of our manuscript. We commend this initiative to involve Early Career Researchers in the peer review process, which we believe is a valuable step toward fostering rigorous and collaborative scientific evaluation.

References:

- 1 Environment, E. A. The European environment — state and outlook 2020. <https://doi.org/https://www.eea.europa.eu/en/analysis/publications/soer-2020>
- 2 Soininen, P., Kangas, A. J., Wurtz, P., Suna, T. & Ala-Korpela, M. Quantitative serum nuclear magnetic resonance metabolomics in cardiovascular epidemiology and genetics. *Circ Cardiovasc Genet* **8**, 192-206 (2015). <https://doi.org/10.1161/CIRCGENETICS.114.000216>
- 3 Wurtz, P. *et al.* Quantitative Serum Nuclear Magnetic Resonance Metabolomics in Large-Scale Epidemiology: A Primer on -Omic Technologies. *Am J Epidemiol* **186**, 1084-1096 (2017). <https://doi.org/10.1093/aje/kwx016>
- 4 Sung, H. *et al.* Global Cancer Statistics 2020: GLOBOCAN Estimates of Incidence and Mortality Worldwide for 36 Cancers in 185 Countries. *CA Cancer J Clin* **71**, 209-249 (2021). <https://doi.org/10.3322/caac.21660>
- 5 Zhao, J. *et al.* Global trends in incidence, death, burden and risk factors of early-onset cancer from 1990 to 2019. *BMJ Oncology* **2**, e000049 (2023). <https://doi.org/10.1136/bmjonc-2023-000049>
- 6 Allemani, C. *et al.* Global surveillance of trends in cancer survival 2000-14 (CONCORD-3): analysis of individual records for 37 513 025 patients diagnosed with one of 18 cancers from 322 population-based registries in 71 countries. *Lancet* **391**, 1023-1075 (2018). [https://doi.org/10.1016/s0140-6736\(17\)33326-3](https://doi.org/10.1016/s0140-6736(17)33326-3)
- 7 Liu, M. *et al.* Residential green and blue spaces with nonalcoholic fatty liver disease incidence: Mediating effect of air pollutants. *Ecotoxicol Environ Saf* **264**, 115436 (2023). <https://doi.org/10.1016/j.ecoenv.2023.115436>
- 8 Krebber, A. M. *et al.* Prevalence of depression in cancer patients: a meta-analysis of diagnostic interviews and self-report instruments. *Psychooncology* **23**, 121-130 (2014). <https://doi.org/10.1002/pon.3409>
- 9 Li, M., Boquiren, V., Lo, C. & Rodin, G. in *Supportive Oncology* (eds Mellar P. Davis, Petra Ch Feyer, Petra Ortner, & Camilla Zimmermann) 528-540 (W.B. Saunders, 2011).
- 10 Walker, J. *et al.* Prevalence of depression in adults with cancer: a systematic review. *Ann Oncol* **24**, 895-900 (2013). <https://doi.org/10.1093/annonc/mds575>
- 11 Eeftens, M. *et al.* Development of Land Use Regression models for PM(2.5), PM(2.5) absorbance, PM(10) and PM(coarse) in 20 European study areas; results of the ESCAPE project. *Environ Sci Technol* **46**, 11195-11205 (2012). <https://doi.org/10.1021/es301948k>
- 12 Beelen, R. *et al.* Development of NO₂ and NO_x land use regression models for estimating air pollution exposure in 36 study areas in Europe – The ESCAPE project. *Atmospheric Environment* **72**, 10-23 (2013). <https://doi.org/https://doi.org/10.1016/j.atmosenv.2013.02.037>
- 13 Wang, M. *et al.* Joint exposure to various ambient air pollutants and incident heart failure: a prospective analysis in UK Biobank. *Eur Heart J* **42**, 1582-1591 (2021). <https://doi.org/10.1093/eurheartj/ehaa1031>
- 14 Liu, B. P. *et al.* Exposure to residential green and blue space and the natural environment is associated with a lower incidence of psychiatric disorders in middle-

- aged and older adults: findings from the UK Biobank. *BMC Med* **22**, 15 (2024). <https://doi.org:10.1186/s12916-023-03239-1>
- 15 D. Morton, C. R., C. Wood, L. Meek, C. Marston, G. Smith, R. Wadsworth, I. C. Simpson. Final Report for LCM2007 - the New UK Land Cover Map July 17th. (2011).
- 16 Alcock, I., White, M. P., Wheeler, B. W., Fleming, L. E. & Depledge, M. H. Longitudinal effects on mental health of moving to greener and less green urban areas. *Environ Sci Technol* **48**, 1247-1255 (2014). <https://doi.org:10.1021/es403688w>
- 17 White, M. P., Alcock, I., Wheeler, B. W. & Depledge, M. H. Would you be happier living in a greener urban area? A fixed-effects analysis of panel data. *Psychol Sci* **24**, 920-928 (2013). <https://doi.org:10.1177/0956797612464659>
- 18 Agay-Shay, K. *et al.* Green spaces and adverse pregnancy outcomes. *Occup Environ Med* **71**, 562-569 (2014). <https://doi.org:10.1136/oemed-2013-101961>
- 19 Maas, J. *et al.* Morbidity is related to a green living environment. *J Epidemiol Community Health* **63**, 967-973 (2009). <https://doi.org:10.1136/jech.2008.079038>
- 20 Reklaitiene, R. *et al.* The relationship of green space, depressive symptoms and perceived general health in urban population. *Scand J Public Health* **42**, 669-676 (2014). <https://doi.org:10.1177/1403494814544494>
- 21 Wan, S. *et al.* Greenspace and mortality in the U.K. Biobank: Longitudinal cohort analysis of socio-economic, environmental, and biomarker pathways. *SSM Popul Health* **19**, 101194 (2022). <https://doi.org:10.1016/j.ssmph.2022.101194>
- 22 Kim, J. *et al.* Global patterns and trends in breast cancer incidence and mortality across 185 countries. *Nat Med* **31**, 1154-1162 (2025). <https://doi.org:10.1038/s41591-025-03502-3>
- 23 Cao, X. *et al.* Association of frailty with the incidence risk of cardiovascular disease and type 2 diabetes mellitus in long-term cancer survivors: a prospective cohort study. *BMC Med* **21**, 74 (2023). <https://doi.org:10.1186/s12916-023-02774-1>
- 24 Venter, Z. S., Hassani, A., Stange, E., Schneider, P. & Castell, N. Reassessing the role of urban green space in air pollution control. *Proc Natl Acad Sci U S A* **121**, e2306200121 (2024). <https://doi.org:10.1073/pnas.2306200121>
- 25 Liu, Z. *et al.* Green space exposure on depression and anxiety outcomes: A meta-analysis. *Environ Res* **231**, 116303 (2023). <https://doi.org:10.1016/j.envres.2023.116303>
- 26 Tomita, A. *et al.* Green environment and incident depression in South Africa: a geospatial analysis and mental health implications in a resource-limited setting. *Lancet Planet Health* **1**, e152-e162 (2017). [https://doi.org:10.1016/s2542-5196\(17\)30063-3](https://doi.org:10.1016/s2542-5196(17)30063-3)
- 27 Wang, Q. *et al.* Revealing the role of leucine in improving the social avoidance behavior of depression through a combination of untargeted and targeted metabolomics. *Food Funct* **14**, 6397-6409 (2023). <https://doi.org:10.1039/d3fo01876h>
- 28 Baek, J. H. *et al.* The Role of Glutamine Homeostasis in Emotional and Cognitive Functions. *Int J Mol Sci* **25** (2024). <https://doi.org:10.3390/ijms25021302>

- 29 He, R. *et al.* Causal Association Between Obesity, Circulating Glutamine Levels, and Depression: A Mendelian Randomization Study. *J Clin Endocrinol Metab* **108**, 1432-1441 (2023). <https://doi.org/10.1210/clinem/dgac707>
- 30 Pu, J. *et al.* Characterizing metabolomic and proteomic changes in depression: a systematic analysis. *Mol Psychiatry* (2025). <https://doi.org/10.1038/s41380-025-02919-z>
- 31 Zhao, M. *et al.* Gut bacteria-driven homovanillic acid alleviates depression by modulating synaptic integrity. *Cell Metab* **36**, 1000-1012.e1006 (2024). <https://doi.org/10.1016/j.cmet.2024.03.010>
- 32 Ari, C. *et al.* Exogenous Ketone Supplements Reduce Anxiety-Related Behavior in Sprague-Dawley and Wistar Albino Glaxo/Rijswijk Rats. *Front Mol Neurosci* **9**, 137 (2016). <https://doi.org/10.3389/fnmol.2016.00137>
- 33 de Kluiver, H. *et al.* Metabolomics signatures of depression: the role of symptom profiles. *Transl Psychiatry* **13**, 198 (2023). <https://doi.org/10.1038/s41398-023-02484-5>
- 34 Vanherle, S., Loix, M., Miron, V. E., Hendriks, J. J. A. & Bogie, J. F. J. Lipid metabolism, remodelling and intercellular transfer in the CNS. *Nat Rev Neurosci* **26**, 214-231 (2025). <https://doi.org/10.1038/s41583-025-00908-3>
- 35 Yang, T., Wang, J., Huang, J., Kelly, F. J. & Li, G. Long-term Exposure to Multiple Ambient Air Pollutants and Association With Incident Depression and Anxiety. *JAMA Psychiatry* **80**, 305-313 (2023). <https://doi.org/10.1001/jamapsychiatry.2022.4812>
- 36 Gu, X. *et al.* Association Between Ambient Air Pollution and Daily Hospital Admissions for Depression in 75 Chinese Cities. *Am J Psychiatry* **177**, 735-743 (2020). <https://doi.org/10.1176/appi.ajp.2020.19070748>
- 37 Jansen, R. *et al.* The metabolome-wide signature of major depressive disorder. *Mol Psychiatry* **29**, 3722-3733 (2024). <https://doi.org/10.1038/s41380-024-02613-6>
- 38 Song, Y. *et al.* Social isolation, loneliness, and incident type 2 diabetes mellitus: results from two large prospective cohorts in Europe and East Asia and Mendelian randomization. *EClinicalMedicine* **64**, 102236 (2023). <https://doi.org/10.1016/j.eclinm.2023.102236>
- 39 Kim, H. J., Min, J. Y., Seo, Y. S. & Min, K. B. Relationship between chronic exposure to ambient air pollution and mental health in Korean adult cancer survivors and the general population. *BMC Cancer* **21**, 1298 (2021). <https://doi.org/10.1186/s12885-021-09013-x>
- 40 Radua, J. *et al.* Impact of air pollution and climate change on mental health outcomes: an umbrella review of global evidence. *World Psychiatry* **23**, 244-256 (2024). <https://doi.org/10.1002/wps.21219>

RESPONSE TO REVIEWERS

Comments from the Reviewers

Reviewer #1:

1. The authors have done substantial work to address most of the previous comments and the manuscript has improved. I applaud the authors' efforts in running additional sensitivity analyses to ensure the findings are robust and consistent. Below are some additional comments based on the study team's response:

Response: We sincerely appreciate the reviewer's recognition of our efforts and the positive feedback on the revised manuscript. We are also grateful for the insightful additional comments, which have helped us further improve the quality and clarity of the study. Please find our point-by-point responses below.

2. Air pollution estimates were derived from 2010 land-use regression models, yet the cohort was enrolled 2006–2010. The authors should explicitly acknowledge the temporal gap and potential exposure misclassification due to use of fixed-year models.

Response: Thank you for your suggestions. Although we conducted sensitivity analyses restricting the sample to individuals who had lived at their current address for more than 10 years before baseline, which confirmed the robustness of our findings, we agree that a temporal gap and potential exposure misclassification may exist due to the use of fixed-year models. Specifically, in UK Biobank, air pollution estimates were derived from 2010 land-use regression models, whereas cohort enrollment occurred during 2006–2010. We have therefore explicitly acknowledged this limitation in the revised manuscript (**lines 422-429**).

Discussion, Lines 422-429: *"Although exposure was assessed at fixed residential addresses without accounting for participants' activity patterns or residential mobility, sensitivity analyses restricted to individuals who had lived at their current address for more than 10 years suggested a consistently robust direction of the associations. However, it should be noted that the temporal gap between the air pollution estimates and the cohort recruitment period may have introduced potential exposure misclassification due to the use of fixed-year models."*

3. Regarding responses to my previous comment #1, the author provided clarification on the sample collection time on the samples undergoing the metabolomics profiling. The authors later stated that "In our study, we considered 249 available metabolic

biomarkers (excluding glucose-lactate and spectrometer-corrected alanine)". It will be best for the author to describe in the manuscript why in particular glucose-lactate and spectrometer-corrected alanine were excluded from the analysis.

Response: Thank you for the reviewer's suggestions. As we previously explained, the baseline blood samples collected at participant recruitment were used to measure NMR metabolic biomarkers in two phases. In the phase 1 release (between June 2019 and April 2020), UKB shared metabolic biomarker data from approximately 118,000 participants at baseline recruitment. The phase 2 release (between April 2020 and June 2022) includes metabolic biomarker data from an additional 157,000 participants at baseline recruitment. Specifically, both phase 1 and phase 2 releases include a total of 249 metabolic measures, comprising 168 absolute concentrations and 81 ratio measures quantified per EDTA plasma sample. However, with the phase 2 release only, two additional derived variables, glucose-lactate and spectrometer-corrected alanine, were shared in addition to the standard set of 249 biomarkers. As a result, these two metabolites are available only in a subset of samples from phase 2 and show considerable discrepancies in sample size compared to those with data on the standard set of 249 metabolic biomarkers. Moreover, since glucose-lactate and spectrometer-corrected alanine are derived from the standard set, and in line with the approach adopted by previous study ¹, we focused solely on the 249 metabolic biomarkers from the standard set. We have incorporated the relevant clarification into the Methods section of the revised manuscript (**lines 540-543**).

Methods, Lines 540-543: *"In our study, we considered 249 available metabolic biomarkers. Glucose-lactate and spectrometer-corrected alanine were excluded due to their availability only in the phase 2 release, which resulted in a substantial discrepancy in sample size compared to the other 249 biomarkers."*

4. Related to the previous comment, the authors stated in their response that "metabolomic profiling was performed in two distinct phases—Phase 1 (June 2019 to 489 April 2020) and Phase 2 (April 2020 to June 2022)". Metabolomics profiling are known to have significant batch effects and the study team should provide details in the method section how the batch effects are corrected and adjusted. Additionally, metabolomics data are known to have a lot of 0 or non-detects across samples (i.e., not all metabolites can be detected in every single bio sample), it remains unclear how the study team deal with these non-detects/0 values, and whether or not imputation is done. In general, QAQC data should be provided for the metabolomics profiling and data processing.

Response: Thank you for your valuable suggestions. In the revised manuscript, we

have added detailed descriptions regarding the quality control and batch effect adjustment strategies applied during metabolomics profiling (**lines 543-548 and Supplementary method**), as well as the approaches used to handle non-detects/zero values (**lines 597-599**).

The UK Biobank and Nightingale Health have implemented a series of quality control measures during the measurement of metabolic biomarkers to minimize batch effect-related biases, which are known to significantly affect metabolomics profiling (<https://biobank.ndph.ox.ac.uk/ukb/label.cgi?id=220>). First, the UK Biobank and Nightingale Health reached a consensus on predefined quality metrics to ensure consistency across samples, and pilot measurements were also conducted. Details of the Nightingale Health NMR biomarker platform have been described previously ^{2,3}. Additionally, the NMR platform used by Nightingale Health has been able to maintain consistent results over time and across different NMR spectrometers. Furthermore, the metabolic biomarker platform at Nightingale Health requires minimal sample preparation, eliminating all extraction steps, which contributes to highly reproducible biomarker measurements. Moreover, during the measurement of UK Biobank samples, Nightingale Health continuously monitored the measurement consistency within and between spectrometers. Two control samples provided by Nightingale Health and two blind duplicate samples provided by the UK Biobank were included in each 96-well plate to track measurement consistency across multiple spectrometers. For both the internal control samples from Nightingale Health and the blind duplicate samples from the UK Biobank, the coefficient of variation (CV) was predefined for the entire spectrum of metabolic biomarkers. These metrics were consistently met for each batch of continuous measurements, with many biomarkers exhibiting a CV below 5%. Importantly, Nightingale Health's metabolic biomarker analysis platform is renowned for its long-term high reproducibility and absence of batch effects. Metabolic biomarker data typically require no preprocessing for epidemiological analyses and can be analyzed in the same manner as the clinical biochemical data available in the UK Biobank. Additionally, throughout the quality control process, biomarkers significantly influenced by interfering substances were excluded.

To enhance the utility of the UK Biobank NMR metabolic biomarker data and promote rapid analysis, the UK Biobank has made further efforts ⁴. First, they identified and described other unnecessary technical sources of variation affecting individual biomarkers in the downloadable data from the UK Biobank. These sources of variation include sample preparation time, shipping plate well, spectrometer batch effects, drift over time within spectrometer, and outlier shipping plates. Subsequently, they developed a procedure to eliminate this unnecessary technical variation and

demonstrated that it could enhance the genetic and epidemiological research signals of the NMR metabolic biomarker data in the UK Biobank. Following this, they reported additional quality control (QC) procedures and made this procedure available to the research community as a resource via a R package, `ukbnmr`, to eliminate the impact of technical variation present in the biomarker concentrations available for download from the UK Biobank. The results showed that, overall, most biomarkers exhibit relative robustness to technical variation. Of the 249 biomarkers, at least 5% of the variation in 22 biomarkers could be explained by one or more technical factors, with the most significant influences being differences in biomarker concentrations between spectrometers and time drift between measurements of different plates within each spectrometer. To eliminate technical differences, the UK Biobank suggested a simple method of median normalization for each plate to remove inter-plate variation. However, since each plate only measures 94 samples, this method might not only eliminate technical variation but also remove biological variation that downstream analysts may be interested in. Additionally, a logarithmic transformation was applied to the original biomarker concentrations, where each unit decrease in biomarker levels corresponds to an increase of one unit, to eliminate any potential impact of sample degradation. As we mentioned in the Methods section regarding metabolite processing: the individual metabolite concentrations were natural logarithmic transformed ($\ln [x + 1]$) and standardized to z-scores.

For the non-detects/0 values in the metabolomics data, we followed the approach of Yi Fan et al ⁵. As explained in the quality control section above, it was assumed that the missing values for the 249 metabolites were attributable to the limit of detection; therefore, these non-detects/0 values were imputed using half of the minimum detectable value. Subsequently, the individual metabolite concentrations were natural logarithmic transformed ($\ln [x + 1]$) and standardized to z-scores.

Methods, Lines 543-548: *"The UK Biobank and Nightingale Health applied rigorous quality control procedures and developed statistical approaches to monitor measurement consistency, exclude unreliable biomarkers, and correct for technical variation, thereby ensuring high reproducibility and minimizing batch effect-related biases in NMR metabolomics data ⁵, with specific details provided in the **Supplementary Method.**"*

Methods, Lines 597-599: *"Missing values for the 249 metabolites, assumed to be related to the detection limit, were imputed as half of the minimum detectable value, consistent with previous study ⁵."*

Supplementary Method: *"The UK Biobank and Nightingale Health implemented a*

series of rigorous quality control (QC) procedures to minimize batch effect–related biases, which are known to significantly affect metabolomics profiling (<https://biobank.ndph.ox.ac.uk/ukb/label.cgi?id=220>). First, consensus was reached on predefined quality metrics to ensure consistency across samples, and pilot measurements were conducted. Details of the Nightingale Health NMR biomarker platform have been described previously ^{2,3}. The NMR platform maintained consistent results across different spectrometers and over time, while requiring minimal sample preparation (no extraction steps), which contributed to high reproducibility. During measurement of UK Biobank samples, Nightingale Health continuously monitored within- and between-spectrometer consistency. Each 96-well plate included two internal control samples (Nightingale Health) and two blind duplicate samples (UK Biobank). Predefined coefficients of variation (CVs) were applied across the biomarker spectrum, with most biomarkers achieving CV <5%. Biomarkers strongly influenced by interfering substances were excluded. This platform has been recognized for its long-term reproducibility and absence of major batch effects, enabling metabolomics data to be analyzed in the same way as clinical biochemistry data in the UK Biobank.

To further improve data utility, the UK Biobank identified technical sources of variation such as sample preparation time, shipping plate well, spectrometer batch, temporal drift, and outlier plates ⁴. A statistical procedure was then developed to mitigate these effects and shown to enhance genetic and epidemiological signals. Additional QC procedures were made publicly available via the R package *ukbnmr*, enabling removal of residual technical variation from biomarker concentrations. Although most biomarkers proved robust, ~22 of the 249 biomarkers showed ≥5% of variance explained by technical factors, most notably spectrometer differences and within-spectrometer temporal drift. To address this, median normalization at the plate level was suggested, although this could also remove biological variation due to the limited number of samples (94 per plate). In addition, a log transformation was applied to biomarker concentrations to reduce the impact of potential sample degradation. Individual metabolite concentrations were natural log-transformed ($\ln[x+1]$) and standardized to z-scores prior to analysis."

5. In the response letter and in the revised manuscript, the authors stated that they "conducted a sensitivity analysis restricted to individuals who had lived at their current address for more than 10 years prior to baseline". It will be best if the author can specify and describe how many individuals they included in this analysis in the main text.

Response: Thank you for your suggestion. In the revised manuscript, we have specified and described in the main text the number of individuals included in the above analysis (**Lines 256-263**). Of the 21,507 cancer survivors, 6,345 with less than 10

years at their current residence were excluded, resulting in 15,162 participants for this sensitivity analysis. To further improve the readability of the manuscript, we have added the corresponding sample sizes for each analysis in the table headers (**Supplementary Tables 11-31**).

Results, Lines 256-263: "Additionally, we conducted a series of sensitivity analyses, including: (1) further adjustment for air pollution, cancer type, sleep pattern, and daily sun exposure (for skin cancer only); (2) exclusion of antidepressant use from the covariates (N=21,507); (3) exclusion of depression cases within one year (N=21,437) or three years (N=21,279) post-cancer; (4) exclusion of deaths within ten years post-cancer (N=20,522); and (5) restriction to participants with more than ten years at their current residence prior to baseline (N=15,162). The results remained consistent with the main findings."

6. I appreciate the authors conducted additional sensitivity analysis by stratifying the main analysis by cancer type. Indeed some cancer types show stronger trends than the others-- it will be helpful for the study team to briefly describe such interesting observations in the result and discussion sections. Specifically, please move the description of methods for the sensitivity analyses by cancer type completed that are not accompanied by results (lines 230-264) to the methods section. The sensitivity analyses by cancer type yielded some interesting results. Consider including reporting on those that are meaningful briefly in the results section. Please include the N of patients by cancer type in your Supplementary Tables (e.g., in the headings).

Response: Thank you for your positive feedback and valuable suggestions. We agree that some cancer types indeed show stronger trends. In line with your recommendations, we have briefly described these noteworthy observations in the Results section (**lines 245-256**), and we have moved the description of the sensitivity analysis methods by cancer type to the Methods section (**lines 626-638**). Furthermore, we have enriched the Discussion with additional insights to better contextualize these findings (**lines 297-308**). In addition, we have added the sample sizes of the analytic subsets to the titles of all supplementary tables presenting sensitivity analyses, including those stratified by cancer type (**Supplementary Tables 11-22**).

Results, Lines 245-256: "In analyses stratified by cancer type, the results for breast cancer survivors—the largest subgroup—revealed a stronger protective association of green space and the natural environment with depression compared with the overall cancer survivor population (**Supplementary Tables 11-12**). For melanoma skin cancer patients, exposure to natural environments within a 1000 m buffer was associated with a slightly reduced risk of depression ($HR_{\text{tertile 3 vs. tertile 1}} = 0.780$, 95% CI: 0.610, 0.998)

(Supplementary Tables 13-14). For non-melanoma skin cancer patients, exposure to blue space within a 1000 m buffer was also associated with a slightly reduced risk of depression ($HR_{\text{tertile 3 vs. tertile 1}} = 0.549$, 95% CI: 0.319, 0.946) (Supplementary Tables 15-16). Interestingly, among lung cancer survivors, high exposure to blue space within a 1000 m buffer was significantly and inversely associated with depression risk (Supplementary Tables 17-18)."

Discussion, Lines 297-308: *"Interestingly, in analyses of survivors of specific cancer types, we found that among breast cancer survivors, green space and natural environment exhibited a stronger protective effect against depression than in the total cancer survivors and survivors of other cancer types. Previous studies have shown that urban green space is associated with a reduced risk of breast cancer⁶, but their impact on the risk of incident depression among breast cancer patients has not yet been explored. Our findings suggest that targeted environmental interventions, such as increasing access to green space, could be incorporated into supportive care strategies for breast cancer survivors to reduce the risk of depression. Exposure to green space and natural environment has been linked to stress reduction, improved mood, and enhanced overall well-being^{7,8}, mechanisms that may be particularly beneficial for this specific population."*

Methods, Lines 626-638: *"A series of sensitivity analyses were carried out to evaluate the robustness of these findings. First, we conducted separate analyses for the major cancer subtypes, including breast, melanoma skin, prostate, and colorectal cancers, as well as for specific cancer types such as lung cancer and non-melanoma skin cancer. Then, we further repeated the primary analysis by (1) further adjusting for air pollutants ($PM_{2.5}$, PM_{10} , NO_2 , and NO_x), both individually and simultaneously and the APS, respectively; (2) additionally adjusting for cancer type, sleep pattern, and daily sun exposure duration (only for two types of skin cancer), respectively; (3) removing antidepressant use from the covariate set; (4) excluding survivors who used antidepressants at baseline, were diagnosed with incident depression within one or three years post-cancer, to minimize the impact of reverse causality; (5) excluding death cases within ten years post-cancer; (6) restricting the analysis to individuals who had lived at their current address for over 10 years prior to baseline."*

7. Line 71-72, $PM_{2.5}$ should be called "fine particulate matter", not "particulate matter".

Response: We thank the reviewer for pointing this out. We have corrected the terminology in the manuscript, referring to $PM_{2.5}$ as "fine particulate matter" instead of "particulate matter" throughout the text.

8. The use of AFT modeling for mediation is reasonable given Cox limitations, but the interpretation of β vs HR still remains non-intuitive for many readers. It may help if the authors include a brief schematic or supplemental figure explaining AFT-based mediation interpretation for clarity.

Response: Thank you for the suggestion. To improve clarity, we have created brief schematic and supplemental figures illustrating the interpretation of AFT-based mediation (**Supplementary Figs. 16 and Supplementary Figs. 2-3**).

Supplementary Fig. 16. A brief schematic of β in the AFT model and HR in the Cox model interpretation.

The relationship between HR (from the Cox proportional hazards model) and β (from the AFT model) is generally complex and model-dependent. The schematic illustration of HR and β values provided here assumes an exponential distribution, under which HR and β can be approximately converted ($\beta \approx 1/\text{HR}$). Abbreviations: AFT, accelerated failure time; HR, hazard ratio.

Supplementary Fig. 2. Mediation analysis of air pollutants and air pollution score in the association between green space, blue space, natural environment at 300 m buffer and depression risk.

The analysis was conducted using Model 3, adjusted for age, sex and ethnicity, educational level, household income, employment status, body mass index, smoking status, drinking status, physical activity, diet and antidepressant use. The regression coefficient (β) represents the effect of exposure on log time to depression onset, where positive values indicate delayed onset (lower risk) and negative values indicate earlier onset (higher risk).

* The proportion mediated was quantified as the ratio of the mediation effect to the total effect. Abbreviations: NO₂, nitrogen dioxide; NO_x, nitrogen oxides; PM₁₀, particulate matter (PM) with aerodynamic diameter $\leq 10 \mu\text{m}$; PM_{2.5}, PM with aerodynamic diameter $< 2.5 \mu\text{m}$.

Supplementary Fig. 3. Mediation analysis of air pollutants and air pollution score in the association between green space, blue space, natural environment at 1000 m buffer and depression risk.

The analysis was conducted using Model 3, adjusted for age, sex and ethnicity, educational level, household income, employment status, body mass index, smoking status, drinking status, physical activity, diet and antidepressant use. The regression coefficient (β) represents the effect of exposure on log time to depression onset, where positive values indicate delayed onset (lower risk) and negative values indicate earlier onset (higher risk).

* The proportion mediated was quantified as the ratio of the mediation effect to the total effect. Abbreviations: NO₂, nitrogen dioxide; NO_x, nitrogen oxides; PM₁₀, particulate matter (PM) with aerodynamic diameter ≤10 μm; PM_{2.5}, PM with aerodynamic diameter < 2.5 μm.

9. The validity of baseline metabolomic profiles as long-term mediators in cancer survivors over many years remains biologically debatable. More discussion of how metabolomic stability over time may affect mediation validity are warranted.

Response: We thank you for this insightful comment. We agree that the validity of baseline metabolomic profiles as long-term mediators is biologically debatable, not only among cancer survivors but also in studies of other chronic diseases. In the revised manuscript, we have expanded the discussion to acknowledge the potential influence of temporal changes in metabolomic stability on the validity of mediation analyses (**lines 399-416**). Specifically, we noted that although baseline metabolomic profiles may capture important systemic alterations, longitudinal fluctuations could attenuate or modify mediation effects. Furthermore, we have made this limitation explicit in the revised manuscript (**lines 441-445**).

Discussion, Lines 399-416: *"It is worth mentioning that, as in most epidemiological studies, the use of single time-point measurements to determine participants' circulating metabolite levels has raised concerns about whether this approach can reasonably reflect stable metabolite levels. Several epidemiological studies have demonstrated that certain metabolites exhibit high or low within-individual reproducibility⁹⁻¹³. Specifically, a recent study evaluated the within-individual reproducibility of plasma metabolites in 61 Black breast cancer survivors from the Women's Health Circle follow-up study¹⁴. The study found that the within-individual reproducibility of plasma metabolites in breast cancer survivors over the course of one year was generally acceptable, suggesting that single time-point measurements could be useful in assessing the associations between metabolites and breast cancer outcomes. We should acknowledge that changes in cancer metabolite levels are inevitable, as the metabolic reprogramming of tumor cells plays a critical role in cancer initiation, proliferation, and progression¹⁵. However, by restricting our study population to long-term cancer survivors, whose disease status is relatively stable, we were able to mitigate this potential source of bias. Emerging research highlights the exciting role played by the unique microenvironment formed by the tumor microenvironment in shaping the tumor's metabolic landscape^{16,17}. Further research is needed on the metabolic stability/reproducibility in cancer patients."*

Limitation, Lines 441-445: *"Seventh, the reliance on baseline metabolic biomarker measurements limits the ability to explore dynamic changes in metabolites and the*

stability of metabolomic profiles over time, which may affect the validity of mediation analyses and their implications for depression risk among long-term cancer survivors."

10. The authors should revise the abstract to include sufficient details for someone to know the purpose, methods, and main results after reading the abstract. As an example, please mention where the geographic area of focus / where the cancer survivors are from.

Response: We thank you for this helpful suggestion. We have revised the Abstract to provide, as much as possible within the strict word limit of Nature Communications, additional details on the study purpose, methods, and main results, including explicit mention of the geographic area and the source of the cancer survivor population.

Abstract, Lines 27-43: *"This study aimed to investigate the longitudinal associations of green space, blue space natural environment, and air pollution with the risk of depression among cancer survivors, and to examine the potential mediating role of metabolites. 21,507 cancer survivors from the UK Biobank were included. Cox proportional hazards models were used to examine the associations of environmental exposures with depression risk among cancer survivors, and to further assess the mediating roles of relevant metabolites identified by elastic net and Lasso regression. During an average follow-up period of 12.39 years, cancer survivors with higher green space and natural environment coverage (tertile 3) within a 1000 m buffer had 15.8% (4.0%–26.1%) and 18.2% (7.0%–28.1%) lower risks of depression, respectively, compared with those in tertile 1, with the strongest protective association observed among breast cancer survivors. Adjustment for air pollution modestly reduced the estimates, but the effects remained significant. Meanwhile, exposure to higher NO₂ and NO_x levels was associated with increased depression risk. Furthermore, metabolite signatures associated with green space and the natural environment may mediate these associations. These findings highlight that residential green space, natural environment, and low air pollution may reduce depression risk among cancer survivors, possibly via metabolic pathways."*

11. While the authors stated that air pollutants were assessed as a mediator based on previous literature that green space may reduce air pollution, there are a lot of limitations to this assumption that need to be addressed and considered in the manuscript. This should be acknowledged as a limitation, as a wide variety of factors (wind speed, rainfall, etc.) can impact this.

Response: We acknowledge that the assumption that green space reduces air pollution has several limitations. Accordingly, in the revised manuscript we added a

statement in the Limitations section emphasizing that various environmental factors (e.g., wind speed, rainfall, and other meteorological conditions) may shape the relationship between green space and air pollution (**lines 438-441**), and thus our mediation findings should be interpreted with caution.

Limitations, Line 438-441: *"Sixth, various environmental and meteorological factors, such as wind speed, rainfall, and seasonal variations, may influence the dispersion and accumulation of air pollutants. Therefore, the observed mediating role of air pollution should be interpreted with caution."*

Reviewer #2:

1. The authors very thoroughly revised the manuscript and provided exceptionally detailed responses to reviewers' comments. The revised version presented interesting and convincing findings. My minor comments are below.

Response: We sincerely thank the reviewer for the positive evaluation of our work and for recognizing our revisions. We greatly appreciate your constructive feedback, and we have carefully addressed the minor comments as outlined below.

2. Abstract. Provide confidence intervals for the effect estimates, e.g., 12.2% (0.3%; 22.6%) for green space tertile 2 vs. tertile 1. It would also make sense to state that adjusting for air pollution slightly attenuated these estimates, but the effects of the natural environment and green space at the 1000 m buffer remained significant. (These results are in Table S12).

Response: Thank you for your valuable comments, which helped us make the Abstract more precise in conveying our study results. In the revised Abstract, we have added confidence intervals for the effect estimates and stated that adjustment for air pollution slightly attenuated these estimates, while the associations of natural environment and green space at the 1000 m buffer remained significant.

Abstract, Lines 27-44: *"This study aimed to investigate the longitudinal associations of green space, blue space natural environment, and air pollution with the risk of depression among cancer survivors, and to examine the potential mediating role of metabolites. 21,507 cancer survivors from the UK Biobank were included. Cox proportional hazards models were used to examine the associations of environmental exposures with depression risk among cancer survivors, and to further assess the mediating roles of relevant metabolites identified by elastic net and Lasso regression."*

During an average follow-up period of 12.39 years, cancer survivors with higher green space and natural environment coverage (tertile 3) within a 1000 m buffer had 15.8% (4.0%–26.1%) and 18.2% (7.0%–28.1%) lower risks of depression, respectively, compared with those in tertile 1, with the strongest protective association observed among breast cancer survivors. Adjustment for air pollution modestly reduced the estimates, but the effects remained significant. Meanwhile, exposure to higher NO₂ and NO_x levels was associated with increased depression risk. Furthermore, metabolite signatures associated with green space and the natural environment may mediate these associations. These findings highlight that residential green space, natural environment, and low air pollution may reduce depression risk among cancer survivors, possibly via metabolic pathways."

3. Graphical Abstract. As a graphical abstract should stand on its own, please explain what models 1, 2, and 3 are. Perhaps it makes sense to simplify and leave only one model that is presented in the Abstract.

Response: Thank you for the comment. We agree that the graphical abstract should be self-explanatory. To improve clarity, we have simplified the figure to present only Model 3.

Graphical abstract

Abbreviations: NO_2 , nitrogen dioxide; NO_x , nitrogen oxides; PM_{10} , particulate matter with aerodynamic diameter $\leq 10 \mu m$; $PM_{2.5}$, particulate matter with aerodynamic diameter $\leq 2.5 \mu m$.

4. Line 546. The Bonferroni correction is too punitive. It would be more appropriate to use an FDR (False Discovery Rate) correction.

Response: We appreciate the reviewer's valuable suggestion. In the revised analysis, we have replaced the Bonferroni correction with the False Discovery Rate (FDR) correction, which provides a less conservative and more appropriate control for multiple testing. The results have been updated accordingly in the manuscript.

5. Figures 2 and 3. There is only one model presented in each figure, but the legends

describe three models. Describe only the model presented here.

Response: We apologize for the confusion. We have revised the legends of Figures 2 and 3 to accurately describe only the models presented in each figure.

Lines 865-879: *"Fig. 2. Joint analysis of air pollution and green space, blue space, and natural environment at 300 m buffer on depression risk among cancer survivors. The HR and 95% CI were estimated using Cox regression with adjustment for covariates in Model 3 (age, sex and ethnicity, educational level, household income and employment status, body mass index, smoking status, drinking status, physical activity, diet and antidepressant use). The three-color groups represent the categories of green space, blue space, and natural environment exposure, with gray, yellow, and blue corresponding to Tertile 1, Tertile 2, and Tertile 3, respectively. The x-axis represents categories of air pollution. For a more intuitive interpretation of the joint analysis results, exposures in Tertile 3, Tertile 2, and Tertile 1 are labeled as High, Middle, and Low exposure, respectively, with the reference group defined as participants exposed to the lowest levels of green space, blue space, or natural environment and simultaneously to the highest category of air pollution. NO₂, nitrogen dioxide; NO_x, nitrogen oxides; PM_{2.5}, particulate matter with aerodynamic diameter < 2.5 μm; HR, hazard ratio; CI, confidence interval."*

Lines 882-896: *"Fig. 3. Joint analysis of air pollutants and green space, blue space, and natural environment within 1000 m buffer on depression risk among cancer survivors. The HR and 95% CI were estimated using Cox regression with adjustment for covariates in Model 3 (age, sex and ethnicity, educational level, household income and employment status, body mass index, smoking status, drinking status, physical activity, diet and antidepressant use). The three-color groups represent the categories of green space, blue space, and natural environment exposure, with gray, yellow, and blue corresponding to Tertile 1, Tertile 2, and Tertile 3, respectively. The x-axis represents categories of air pollution. For a more intuitive interpretation of the joint analysis results, exposures in Tertile 3, Tertile 2, and Tertile 1 are labeled as High, Middle, and Low exposure, respectively, with the reference group defined as participants exposed to the lowest levels of green space, blue space, or natural environment and simultaneously to the highest category of air pollution. NO₂, nitrogen dioxide; NO_x, nitrogen oxides; PM_{2.5}, particulate matter with aerodynamic diameter < 2.5 μm; HR, hazard ratio; CI, confidence interval."*

6. Fig. 2 and 3. Specify what variable is on the X axis and what variable is denoted by colors. I assume that exposure to air pollution is on the X axis, and exposure to green space is denoted by the color. Fig 2 shows that there are no associations between

green/blue/natural spaces and depression risk at low levels of air pollution. Explanations?

Response: We apologize for the lack of clarity. We have revised the figure legends to clarify the axes: the X-axis indicates the levels of air pollution exposure, while the color scale reflects the levels of exposure to green space, blue space, or the natural environment. In fact, Fig. 2 shows that the associations between green, blue, or natural spaces and depression risk were significant at low levels of air pollution. The misunderstanding arose from unclear legends and formatting. To improve the readability of the figures, we have adjusted certain aspects of the layout, as shown below (**Figs. 2-3**).

We would like to clarify that, based on our findings, the inverse associations of green, blue, and natural spaces with depression risk reached statistical significance primarily at lower levels of air pollution. In fact, the results showed that compared with participants in Tertile 1 of green space at 300 m buffer and Tertile 3 of NO₂, NO_x, PM_{2.5}, and the APS, those in Tertile 3 of green space and Tertile 1 of air pollutants had a lower risk of depression by 21.1% (HR = 0.789, 95% CI: 0.670, 0.929, P = 0.005), 15.3% (HR = 0.847, 95% CI: 0.718, 1.001, P = 0.051), 16.7% (HR = 0.833, 95% CI: 0.708, 0.980, P = 0.027), and 18.1% (HR = 0.819, 95% CI: 0.696, 0.962, P = 0.015), respectively. Compared with participants in Tertile 1 of natural environment at 300 m buffer and Tertile 3 of NO₂, NO_x, PM_{2.5}, and the APS, those in Tertile 3 of natural environment and Tertile 1 of air pollutants had a lower risk of depression by 16.0% (HR = 0.840, 95% CI: 0.714, 0.989, P = 0.036), 14.1% (HR = 0.859, 95% CI: 0.731, 1.009, P = 0.064), 14.8% (HR = 0.852, 95% CI: 0.727, 0.999, P = 0.049), and 15.2% (HR = 0.848, 95% CI: 0.723, 0.993, P = 0.041), respectively. Furthermore, compared with participants in Tertile 1 of green space at 1000 m buffer and Tertile 3 of NO₂, NO_x, PM_{2.5}, and the APS, those in Tertile 3 of green space and Tertile 1 of air pollutants had a significantly lower risk of depression by 17.9% (HR = 0.821, 95% CI: 0.705, 0.956, P = 0.011), 16.4% (HR = 0.836, 95% CI: 0.712, 0.982, P = 0.029), 16.1% (HR = 0.839, 95% CI: 0.714, 0.986, P = 0.033), and 16.4% (HR = 0.836, 95% CI: 0.716, 0.977, P = 0.024), respectively. Similarly, compared with participants in Tertile 1 of natural environment at 1000 m buffer and Tertile 3 of air pollutants, those in Tertile 3 of natural environment and Tertile 1 of NO₂, NO_x, PM_{2.5}, and the APS exhibited a reduced depression risk by 16.3% (HR = 0.837, 95% CI: 0.720, 0.973, P = 0.021), 17.7% (HR = 0.823, 95% CI: 0.703, 0.964, P = 0.016), 15.3% (HR = 0.847, 95% CI: 0.724, 0.991, P = 0.038), and 15.2% (HR = 0.834, 95% CI: 0.716, 0.972, P = 0.020), respectively. Similar results were observed in the joint analyses of blue space at 300 m and 1000 m buffers with air pollution.

Fig. 2. Joint analysis of air pollution and green space, blue space, and natural environment at 300 m buffer on depression risk among cancer survivors.

The HR and 95% CI were estimated using Cox regression with adjustment for covariates in Model 3 (age, sex and ethnicity, educational level, household income and employment status, body mass index, smoking status, drinking status, physical activity, diet and antidepressant use). The three-color groups represent the categories of green space, blue space, and natural environment exposure, with gray, yellow, and blue corresponding to Tertile 1, Tertile 2, and Tertile 3, respectively. The x-axis represents categories of air pollution. For a more intuitive interpretation of the joint analysis results, exposures in Tertile 3, Tertile 2, and Tertile 1 are labeled as High, Middle, and Low exposure, respectively, with the reference group defined as participants exposed to the lowest levels of green space, blue space, or natural environment and simultaneously to the highest category of air pollution. NO₂, nitrogen dioxide; NO_x, nitrogen oxides; PM_{2.5}, particulate matter with aerodynamic diameter < 2.5 µm; HR, hazard ratio; CI, confidence interval.

Fig. 3. Joint analysis of air pollutants and green space, blue space, and natural environment within 1000 m buffer on depression risk among cancer survivors.

The HR and 95% CI were estimated using Cox regression with adjustment for covariates in Model 3 (age, sex and ethnicity, educational level, household income and employment status, body mass index, smoking status, drinking status, physical activity, diet and antidepressant use). The three-color groups represent the categories of green space, blue space, and natural environment exposure, with gray, yellow, and blue corresponding to Tertile 1, Tertile 2, and Tertile 3, respectively. The x-axis represents categories of air pollution. For a more intuitive interpretation of the joint analysis results, exposures in Tertile 3, Tertile 2, and Tertile 1 are labeled as High, Middle, and Low exposure, respectively, with the reference group defined as participants exposed to the lowest levels of green space, blue space, or natural environment and simultaneously to the highest category of air pollution. NO₂, nitrogen dioxide; NO_x, nitrogen oxides; PM_{2.5}, particulate matter with aerodynamic diameter < 2.5 μm; HR, hazard ratio; CI, confidence interval.

7. Page 15. Briefly describe the findings presented in Supplementary Tables 17-31, focusing on the general patterns.

Response: We sincerely appreciate your suggestion. In the revised manuscript, we have included a brief description of the findings from Supplementary Tables 17–31 (lines 244-263).

Results, Lines 244-263: "We performed sensitivity analyses to analyze the

robustness of our findings (**Supplementary Tables 11-31**). In analyses stratified by cancer type, the results for breast cancer survivors—the largest subgroup—revealed a stronger protective association of green space and the natural environment with depression compared with the overall cancer survivor population (**Supplementary Tables 11-12**). For melanoma skin cancer patients, exposure to natural environments within a 1000 m buffer was associated with a slightly reduced risk of depression ($HR_{\text{tertile 3 vs. tertile 1}} = 0.780$, 95% CI: 0.610, 0.998) (**Supplementary Tables 13-14**). For non-melanoma skin cancer patients, exposure to blue space within a 1000 m buffer was also associated with a slightly reduced risk of depression ($HR_{\text{tertile 3 vs. tertile 1}} = 0.549$, 95% CI: 0.319, 0.946) (**Supplementary Tables 15-16**). Interestingly, among lung cancer survivors, high exposure to blue space within a 1000 m buffer was significantly and inversely associated with depression risk (**Supplementary Tables 17-18**). Additionally, we conducted a series of sensitivity analyses, including: (1) further adjustment for air pollution, cancer type, sleep pattern, and daily sun exposure (for skin cancer only); (2) exclusion of antidepressant use from the covariates ($N=21,507$); (3) exclusion of depression cases within one year ($N=21,437$) or three years ($N=21,279$) post-cancer; (4) exclusion of deaths within ten years post-cancer ($N=20,522$); and (5) restriction to participants with more than ten years at their current residence prior to baseline ($N=15,162$). The results remained consistent with the main findings."

8. Lines 318-320. It seems that the effects of greenness were observed at low air pollution levels only. "Stronger effects" would mean that the protective effects of greenness were observed at all levels of air pollution.

Response: Thank you for your comment. We have revised the text to clarify this point and removed the phrase "stronger effects" to avoid any misinterpretation.

Discussion, Lines 338-340: "Another key finding was that high levels of green, blue, and natural spaces, in combination with low levels of air pollution, conferred protective effects against depression risk among cancer survivors."

Reviewer #3:

Response: We sincerely thank you for your positive response and for recognizing the effort we put into our revision. We are also truly grateful for your time and effort in re-

reviewing our manuscript.

References:

- 1 Jia, X. *et al.* A Novel Metabolomic Aging Clock Predicting Health Outcomes and Its Genetic and Modifiable Factors. *Adv Sci (Weinh)* **11**, e2406670, doi:10.1002/advs.202406670 (2024).
- 2 Wurtz, P. *et al.* Quantitative Serum Nuclear Magnetic Resonance Metabolomics in Large-Scale Epidemiology: A Primer on -Omic Technologies. *Am J Epidemiol* **186**, 1084-1096, doi:10.1093/aje/kwx016 (2017).
- 3 Soininen, P., Kangas, A. J., Wurtz, P., Suna, T. & Ala-Korpela, M. Quantitative serum nuclear magnetic resonance metabolomics in cardiovascular epidemiology and genetics. *Circ Cardiovasc Genet* **8**, 192-206, doi:10.1161/CIRCGENETICS.114.000216 (2015).
- 4 Ritchie, S. C. *et al.* Quality control and removal of technical variation of NMR metabolic biomarker data in ~120,000 UK Biobank participants. *Sci Data* **10**, 64, doi:10.1038/s41597-023-01949-y (2023).
- 5 Fan, Y. *et al.* Effects of diets on risks of cancer and the mediating role of metabolites. *Nat Commun* **15**, 5903, doi:10.1038/s41467-024-50258-4 (2024).
- 6 O'Callaghan-Gordo, C. *et al.* Residential proximity to green spaces and breast cancer risk: The multicase-control study in Spain (MCC-Spain). *Int J Hyg Environ Health* **221**, 1097-1106, doi:10.1016/j.ijheh.2018.07.014 (2018).
- 7 Geneshka, M., Coventry, P., Cruz, J. & Gilbody, S. Relationship between Green and Blue Spaces with Mental and Physical Health: A Systematic Review of Longitudinal Observational Studies. *Int J Environ Res Public Health* **18**, doi:10.3390/ijerph18179010 (2021).
- 8 Nguyen, P. Y., Astell-Burt, T., Rahimi-Ardabili, H. & Feng, X. Green Space Quality and Health: A Systematic Review. *Int J Environ Res Public Health* **18**, doi:10.3390/ijerph182111028 (2021).
- 9 Goerdten, J. *et al.* Reproducibility of the Blood and Urine Exposome: A Systematic Literature Review and Meta-Analysis. *Cancer Epidemiol Biomarkers Prev* **31**, 1683-1692, doi:10.1158/1055-9965.EPI-22-0090 (2022).
- 10 Zeleznik, O. A. *et al.* Intrapersonal Stability of Plasma Metabolomic Profiles over 10 Years among Women. *Metabolites* **12**, doi:10.3390/metabo12050372 (2022).
- 11 Zheng, Y., Yu, B., Alexander, D., Couper, D. J. & Boerwinkle, E. Medium-term variability of the human serum metabolome in the Atherosclerosis Risk in Communities (ARIC) study. *OMICS* **18**, 364-373, doi:10.1089/omi.2014.0019 (2014).
- 12 Townsend, M. K. *et al.* Reproducibility of metabolomic profiles among men and women in 2 large cohort studies. *Clin Chem* **59**, 1657-1667, doi:10.1373/clinchem.2012.199133 (2013).
- 13 Sampson, J. N. *et al.* Metabolomics in epidemiology: sources of variability in metabolite measurements and implications. *Cancer Epidemiol Biomarkers Prev* **22**, 631-640, doi:10.1158/1055-9965.EPI-12-1109 (2013).
- 14 Qin, B. *et al.* Reproducibility of Plasma Metabolome over 1 Year in a Population-Based Cohort of Black Breast Cancer Survivors. *Cancer Epidemiol Biomarkers Prev* **34**, 914-921, doi:10.1158/1055-9965.EPI-24-1646 (2025).

- 15 Hanahan, D. & Weinberg, R. A. Hallmarks of cancer: the next generation. *Cell* **144**, 646-674, doi:10.1016/j.cell.2011.02.013 (2011).
- 16 Campbell, S. L. & Wellen, K. E. Metabolic Signaling to the Nucleus in Cancer. *Mol Cell* **71**, 398-408, doi:10.1016/j.molcel.2018.07.015 (2018).
- 17 Christen, S. *et al.* Breast Cancer-Derived Lung Metastases Show Increased Pyruvate Carboxylase-Dependent Anaplerosis. *Cell Rep* **17**, 837-848, doi:10.1016/j.celrep.2016.09.042 (2016).

RESPONSE TO REVIEWERS

REVIEWERS' COMMENTS

Reviewer #1 (Remarks to the Author):

The authors have addressed all my previous comments and I have no further comment to add.

Response: We sincerely appreciate the reviewer's careful assessment and constructive feedback, which have greatly improved the quality of our manuscript.